# Topic Modeling Revisited: A Document Graph-based Neural Network Perspective

**Dazhong Shen**[1,2], **Chuan Qin**[2], **Chao Wang**[1,2], **Zheng Dong**[2], **Hengshu Zhu**[2,*], **Hui Xiong**[3,*]

[1]School of Computer Science and Technology, University of Science and Technology of China
[2]Baidu Talent Intelligence Center, Baidu Inc.
[3]Artificial Intelligence Thrust, The Hong Kong University of Science and Technology
sdz@mail.ustc.edu.cn, chuanqin0426@gmail.com, wdyx2012@mail.ustc.edu.cn,
zhdong0@outlook.com, zhuhengshu@gmail.com, xionghui@ust.hk

## Abstract

Most topic modeling approaches are based on the bag-of-words assumption, where each word is required to be conditionally independent in the same document. As a result, both of the generative story and the topic formulation have totally ignored the semantic dependency among words, which is important for improving the semantic comprehension and model interpretability. To this end, in this paper, we revisit the task of topic modeling by transforming each document into a directed graph with word dependency as edges between word nodes, and develop a novel approach, namely Graph Neural Topic Model (GNTM). Specifically, in GNTM, a well-defined probabilistic generative story is designed to model both the graph structure and word sets with multinomial distributions on the vocabulary and word dependency edge set as the topics. Meanwhile, a Neural Variational Inference (NVI) approach is proposed to learn our model with graph neural networks to encode the document graphs. Besides, we theoretically demonstrate that Latent Dirichlet Allocation (LDA) can be derived from GNTM as a special case with similar objective functions. Finally, extensive experiments on four benchmark datasets have clearly demonstrated the effectiveness and interpretability of GNTM compared with state-of-the-art baselines.

## 1 Introduction

As one of the most widely-used techniques for document analysis, topic modeling aims to learn a set of latent topics from the observed document collection, each of which describes an interpretable semantic concept. In particular, Latent Dirichlet Allocation (LDA) [5] and its extensions [3, 9, 44, 60] have achieved great success in various application scenarios over the past decades. These approaches usually specify a probabilistic generative model to draw the document data with a structure of latent variables sampled from prior distributions and word distribution on the vocabulary as topics. Recently, with the development of Variational Autoencoder (VAE) [24], a Neural Variational Inference algorithm (NVI) with neural networks for topic modeling, namely Neural Topic Model, has attracted great attention [35, 34, 11, 12, 42, 49, 61] , due to its appealing flexibility and scalability.

However, it is well-known that traditional topic modeling usually has the bag-of-word assumption, where the words in a document are assumed to be "exchangeable", which also arouses two typical challenges behind most topic models. On one hand, given the topic distribution of the document, the words are conditionally independent. This implicates that the dependency relationship among words is totally ignored, while words are dependent on each other according to linguistic knowledge [30].

---

[*]*Corresponding authors. This work was accomplished when Dazhong Shen was an intern in Baidu supervised by Chuan Qin and Hengshu Zhu.

35th Conference on Neural Information Processing Systems (NeurIPS 2021).

On the other hand, the concept of topics is introduced as multinomial distributions on vocabulary without modeling the dependency relation among words, which produces more ambiguous terms in topics. For example, LDA discovers word such as "network" in a topic which does not seem to be that insightful. Instead, we can convey more interpretability to readers by mining the strong word dependency between "neural" and "network" [19].

Along this line, many extensions have been proposed to address the above two issues by dealing with the dependency relation among words. One thread of these works focus on the sequential dependency by taking the document as a sequence of words [12, 15, 16, 28, 32, 42, 56, 52, 55, 60]. However, most of them prefer to enhance language modeling with better performance on language modeling metric, such as Perplexity, rather than the state-of-the-art quality of learned topics. It may be due to the limitation that they only handle the linear dependency of words [30] and push models to capture the local syntax information instead of global semantic information. Specifically, to mitigate the first problem, most of them assume that the topic assignment or generative possibility of a word is dependent on the proceeding words [12, 15, 16, 28, 42, 56]. As for the second problem, related literature is restricted to exploring n-gram terms by integrating few neighboring words as one term with Markov Chain based generative story [52, 55, 32].

Nonetheless, words may be mutually dependent on each other in a much more complex and non-linear manner. Therefore, several studies explore to model the graph representation of documents, where nodes denote words and edges represent the relationships among words, such as syntax or semantic relation constructed by dependency parsing [30, 54] or justified by relative position among words [9, 61]. They either consider the graph structure as side information to constrain the relation of topic assignments of words [30, 54], or replacing the word sets with edge sets as the instances to generative [9, 61]. Most of them only trickle with the first problem without efforts to enhance the interpretability of topics.

To this end, in this paper, we represent documents as directed semantic graphs by introducing word dependency as edges between word nodes, and develop a Graph Neural Topic Modeling (GNTM) method. The key contributions of our model can be categorized into four parts: 1) we formulate a well-defined probabilistic generative story of the document graph with the novel generative probabilistic functions for both graph structure and word sets; 2) we propose a new concept of topics, consisting of multinomial distributions on both the vocabulary, like conventional topic models, and word dependency edges; 3) a Neural Variational Inference (NVI) approach is designed to infer our model with graph neural network to encode the document graphs; 4) we also demonstrate that LDA can be derived from GNTM as a special case with similar objective functions. Finally, extensive experiments over four benchmark datasets have clearly demonstrated the effectiveness and interpretability of GNTM compared with state-of-the-art baselines.

## 2 Preliminary

### 2.1 Topic Model

Topic modeling is one of the most important text mining techniques and has been extensively studied for a variety of applications in recent two decades [2, 21, 31, 48, 47]. Among the most representative models, Latent Dirichlet Allocation (LDA) [5] formulates each topic $k$ as the distribution $\beta_k^v$ on vocabulary $V$ and draws each word $w_{d,n}$ in a document $d$ from one of $K$ topics stochastically. The well-known generative process of LDA can be described as three steps: 1) sampling document-topic proportions $\theta_d \sim \text{Dir}(\alpha)$, where $\alpha$ is hype-parameter for the Dirichlet distribution prior; 2) drawing individual topic assignments $z_{d,n} \sim \text{Multi}(\theta_d)$; 3) generating each word $w_{d,n} \sim \text{Multi}(\beta_{z_{d,n}}^v)$. Then, the marginal likelihood for a document $d$ is:

$$p(d|\alpha, \beta) = \int_{\theta_d} p(\theta_d|\alpha) \prod_n \sum_{z_{d,n}} p(w_{d,n}|\beta_{z_{d,n}}^v) p(z_{d,n}|\theta_d) d\theta_d. \tag{1}$$

Conventional approaches for inferring topic models, such as LDA, are either Markov Chain Monte Carlo [7] or probabilistic variational inference [22], which are both widely applied [58, 14, 46]. However, they either require careful selection of distribution for each latent variable or tolerate arduous and customized mathematical derivation case by case, which limits the flexibility and scalability of model design [35]. Recently, with the successes of Variational Autoencoder (VAE) [24],

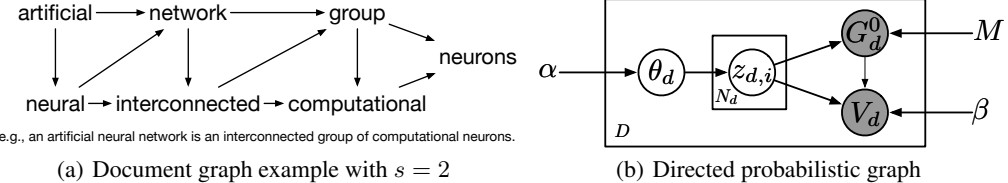



(a) Document graph example with $s = 2$       (b) Directed probabilistic graph



Figure 1: The document graph example and the directed probabilistic graph of our model.

which applies the variational distributions parameterized by neural networks to approximate the posterior of latent variables, the use of deep learning on inferring topic models has attracted more and more attention [57]. Most of those neural topic models (NTMs) [35, 34, 49, 11] can be regarded as extensions of LDA, but often integrate out the discrete latent variable $z_{d,n}$. Therefore, the variational lower bound (ELBO) for the log-likelihood of the document $d$ can be derived as:

$$\mathcal{L}_d = E_{q_\phi(\theta|d)}[\sum_n \log p(w_{d,n}|\theta_d, \beta^v)] - D_{KL}[q_\phi(\theta_d|d)||p(\theta_a)], \tag{2}$$

where $q_\phi(\theta|d)$ is the variational distribution of $\theta_d$ parametrized by a neural network $\phi$. To obtain a differentiable estimator of $\mathcal{L}_d$, the reparameterization trick (RT) has been applied for $\theta$. In particular, due to the limitation of RT on Dirichlet distribution [49], those approaches usually stand on Gaussian-based posterior $q_\phi(\theta_d|d)$.

### 2.2 Document Graph Construction

Most existing methods in topic models commonly represent a document as an unordered collection of words, known as bag-of-word (BOW) representation, which totally ignores the relationship among words in the document. To represent documents with word dependency relations, we discuss the representation of documents in a graph manner, where the graph-based models has been widely explored and applied recently extensively [25, 50, 13].

Formally, given one document $d$ with word sequences $[w_{d,t}]_{t=1}^{N_d}$, we construct a directed edge from $w_{d,t_1}$ to $w_{d,t_2}$ if $0 < t_2 - t_1 \leq s$, where $s$ is the hyper-parameter of window size. The hidden logic is that the semantic is consistent and interdependent among neighboring words, which can be traced back to n-gram [8], BTM [9] and Word2vec [36]. As the example in Figure 1(a) shows (stop words are filtered), we can denote a document $d$ as a graph $G_d = (V_d, E_d)$, where $V_d = \{w_{d,n}\}_{n=1}^{N_d}$ (words as nodes, e.g., 7 words in Figure 1(a)) and $E_d$ are sets of nodes and word edges (e.g., ("artificial", "neural")), respectively. Note that, for simplification in analysis, we distinguish the same words on different positions in $E_d$ and $V_d$. We also denote $E$ as the unique word edge sets in the whole document collection with nodes in the vocabulary $V = \{w_i\}_{i=1}^{|V|}$ (unique word set). For convention, we further denote $G_d^o = (V_d^o, E_d^o)$ as the graph structure of $G_d$ with only the placeholders $V_d^o = \{1, 2, ..., N_d\}$ (e.g., $\{1, 2, .., 7\}$ in Figure 1(a)), not the specific words $V_d$, where $|V_d^o| = |V_d|$, and the placeholder edge set $E_d^o$ is the edge set on the placeholder set $V_d^o$ (e.g., $(1, 2)$), distinguishing from the word edge set $E_d$.

## 3 Graph Neural Topic Model

With the definition of the document graph $G_d$, we turn to introduce the technical detail of GNTM. Specifically, we first formulate the generative story for both the graph structure $G_d^o$ and word sets $V_d$. Then, an NVI inference algorithm will be proposed to learn our model.

### 3.1 Generative Story

Figure 1(b) shows our directed graphical model. In contrast to three-step generative process in LDA, four steps are involved in the generative story. The main differences lie in the conditional generation for the observed data given all latent variables (steps 3 and 4 in the following). To be specific, 1) the generative story of a document $d$ begins with topic proportion $\theta$ drawn from the Dirichlet prior $\text{Dir}(\alpha)$. Then, 2) each word $w_n$ (or placeholder $n$) has corresponding topic assignment $z_n$ drawn from $\text{Multi}(\theta)$. Finally we generate the observed data with two steps: 3) drawing the graph structure

$G_d^o$ conditioning on the topic assignment set $Z_d$, i.e., $G_d^o \sim p(G_d^o|Z_d)$; 4) generating the word set $V_d$ based on both topic assignment set $Z_d$ and graph structure $G_d^o$, i.e., $V_d \sim p(V_d|G_d^o, Z_d)$. Based on the generative process, the joint model of $G_d$ and all latent variables can be decomposed as:

$$p(G_d, \theta_d, Z_d; \alpha) = p(V_d|Z_d, G_d^o)p(G_d^o|Z_d) \prod_{n=1}^{N_d} p(z_{d,n}|\theta_d)p(\theta|\alpha). \tag{3}$$

In the following, we turn to formulate the formal definition of $p(G_d^o|Z_d)$ and $p(V_d|Z_d, G_d^o)$ with additional parameters, i.e., topic dependency matrix $M$ and topic set $\beta$, which is the most important contribution of our model.

### 3.1.1 Generation for Graph Structure

Here, we aim to construct the graph structure $G_d^o$ based solely on the topic assignment $z_{d,n}$ on each placeholder $w_{d,n}^o$. The hidden motivation is to explain the word dependency as the semantic dependency among their topics, which can encourage our model to find the relation among different topics in turn. To be specific, we denote $M \in \mathbb{R}^{K \times K}$ as the dependency matrix among topics, where each element $m_{i,j} \in [0,1]$ represents the possibility that there exists a dependency edge between two placeholders with topic $i$ and $j$, respectively. Then, we can formulate $p(G_d^o|Z_d)$ as :

$$p(G_d^o|Z_d; M) = \prod_{(n,n') \in E_d^o} m_{z_{d,n}, z_{d,n'}} \prod_{(n,n') \notin E_d^o} (1 - m_{z_{d,n}, z_{d,n'}}), \tag{4}$$

where $M$ is a learnable parameter without additional assumptions.

### 3.1.2 Generation for Word Set

Contrast to LDA, once a document is represented as a graph, the word sampling should be cast as the sampling of this graph. In other words, we should consider dependency between words when generating word set $V_d$. To provide a well-defined word sampling process, we first re-define the concept of topic. Different from the conventional concept of topic in LDA, which is represented by the multinomial distribution $\beta_k^v$ on the vocabulary, we further enhance each topic $k$ by introducing one distribution on word dependency edge with parameters $\beta_k^e$, i.e., $\beta_k = (\beta_k^v, \beta_k^e)$, where:

$$\sum_{w \in V} \beta_{k,w}^v = 1, \quad \sum_{w,w' \in V} \beta_{k,w}^v \beta_{k,w'}^v \beta_{k,(w,w')}^e = 1, \quad \beta_{k,w}^v \geq 0, \quad \beta_{k,(w,w')}^e \geq 0. \tag{5}$$

where the product $\beta_{k,w}^v \beta_{k,w'}^v \beta_{k,(w,w')}^e$ denotes the possibility of the directed edge from word $w$ to $w'$ under the topic $k$. Note that, in practice, only parameters $\beta_{k,(w,w')}^e$ corresponding to edge $(w.w') \in E$ would be used in model learning. In other words, we can reduce the space complexity of $\beta_k^e$ into $O(|E|)$ easily by assuming that $\beta_{k,(w,w')}^e = 0, \forall (w,w') \notin E$.

Then, based on the new concept of the topics, we propose a well-defined probability measure for generating each word set $V_d$ with the document graph structure $G_d^o$ and topic assignment sets $Z_d$. The probability of $p(V_d|Z_d, G_d^o)$ is modeled by :

$$p(V_d|Z_d, G_d^o; \beta) = \prod_{k=1}^K p(V_{d,k}|G_{d,k}^o, \beta_k),$$

$$p(V_{d,k}|G_{d,k}^o; \beta_k) = \frac{1}{|E_{d,k}|} \prod_{w \in V_{d,k}} \beta_{k,w}^v \sum_{(w,w') \in E_{d,k}} \beta_{k,(w,w')}^e, \tag{6}$$

where $G_{d,k}^o$ is the graph structure consisting of all placeholders with the topic assignment $k$, similarly, $V_{d,k}$ is the set of words assigned by topic $k$ and $E_{d,k}$ is the word dependency edge among them. In other words, we split the document graph $G_d$ into $K$ dependent sub-graphs $G_{d,k}$ based on topic assignment set $Z_d$ and then define the generative possibility of each topical sub-graph $G_{d,k}$ with the word dependency edges $E_{d,k}$. If $E_{d,k} = \emptyset$, we set $\sum_{(w,w') \in E_{d,k}} \beta_{k,(w,w')}^e / |E_{d,k}| = 1$. Mathematically, we have the following theorem:

**Theorem 1** *Given the topic set $\beta$ in Equation 5 and the document graph structure $G_d^o$, the probability function $p(V_d|Z_d, G_d^o; \beta)$ in Equation 6 is a legal probability measure on the vocabulary $V$.*

The proof can be found in Appendix A.1 which is mainly inspired by [30].

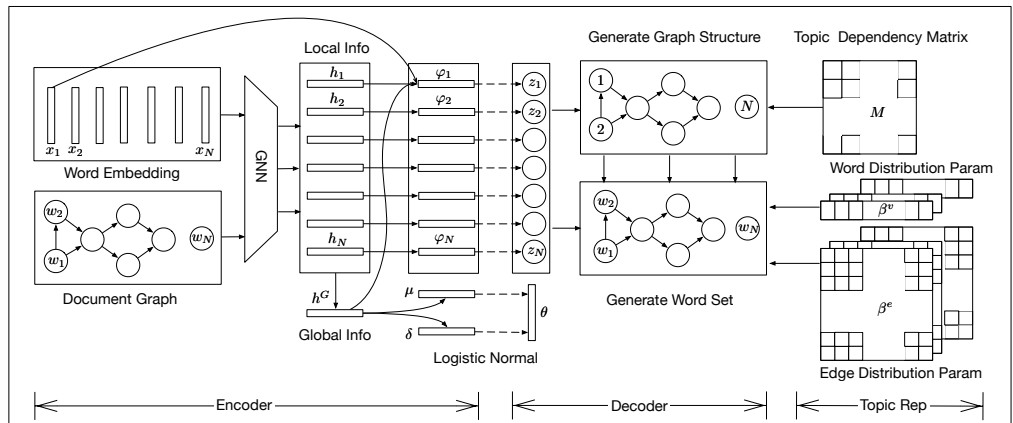

Figure 2: The overivew of network structures to infer our model.

## 3.2 NVI Inference

Here, we infer our model by applying a NVI inference algorithm, which is well-know of flexibility and scalability [24]. Based on the mean field assumption, we approximate the posterior of latent variables $\theta_d$ and $Z_d$ for each document $d$ with the following variational family:

$$q(\theta_d, Z_d | G_d) = q_\phi(\theta_d | G_d; \mu_d, \delta_d) \prod_{n=1}^{N_d} q_\phi(z_{d,n} | G_d, w_{d,n}; \varphi_{d,n}), \qquad (7)$$

where $q_\phi(z_{d,n} | G_d, w_{d,n})$ is defined as one multinomial distribution with parameters $\varphi_{d,n}$ and $q_\phi(\theta_d | G_d)$ is the approximation for the true Dirichlet posterior $p(\theta | G_d)$. Both of them are parameterized by neural networks $\phi$. However, directly approaching the Dirichlet distributions would make the reparameterization trick hard to apply. Here, we follow [49, 17] and resolve this issue by constructing a Laplace approximation to the Dirichlet posterior with the Logistic Normal distribution parameterized by $(\mu_d, \delta_d)$, that is,

$$\theta_d = \text{softmax}(\mathcal{N}(\mu_d, \delta_d^2)). \qquad (8)$$

Then, we can derive the training objective for each document $d$, i.e., ELBO:

$$\mathcal{L}_d = E_{q(Z_d | G_d)}[\log p(G_d^o | Z_d; M) + \log p(V_d | Z_d, G_d^o; \beta)]$$

$$-KL[q(\theta_d | G_d) || p(\theta_d)] - E_{q(\theta_d | G_d)}[\sum_{n=1}^{N_d} KL[q(z_{d,n} | G_d, w_{d,n}) || p(z_{d,n} | \theta_d)]], \qquad (9)$$

where the first line aims to reconstruct the observed data $G_d$, and the second line consists of two KL divergence that attempts to close the distance between the posterior and prior of latent variables $\theta_d$ and $Z_d$. Fortunately, $KL[q(\theta_d | G_d) | p(\theta_d)]$ and $KL[q(z_{d,n} | G_d, w_{d,n}) || p(z_{d,n} | \theta_d)]$ are both analytic (see Appendix A.2). Then, we approximate the expectation of the second KL term over $\theta$ with Monte Carlo samples by applying the reparameterization trick on $q(\theta_d | G_d^0)^2$. As for the deduction of the first line, we will discuss in Section 3.2.3.

After the probabilistic definition of the generative story and the corresponding variational family, we turn to introduce the detailed network structure to implement our model. As the overview shown in Figure 2, the neural network structure of GNTM can be broadly split into three parts: 1) the topic representation, including the topic set $\beta$ and the dependency relation matrix $M$; 2) an encoder, which maps the observed data to the variational parameters; 3) a decoder, which computes the possibility of the observed data given the latent variables. We describe each term as below.

---

[2]Although, the expectation of the second KL term can also be computed analytically under Dirichlet distribution, in which our model is conjugate, we still use the sampling strategy here, which is more general and can also be applied for more flexible situations without conjugate distributions, such as [4, 53]

### 3.2.1 Topic Representation

Inspired by [34], we introduce word vectors $r \in \mathbb{R}^{|V| \times H}$, topic-word vector $u^v \in \mathbb{R}^{K \times H}$, topic-edge vector $u^e \in \mathbb{R}^{K \times 2H}$, and generate $\beta_k^v$ and $\beta_k^e$ by:

$$\beta_{k,w}^v = \frac{exp(r_w \cdot (u_k^v)^T)}{\sum_{w' \in V} exp(r_{w'} \cdot (u_k^v)^T)}, \ \beta_{k,(w_1,w_2)}^e = \frac{exp(\hat{r}_{w_1,w_2} \cdot (u_k^e)^T)}{\sum_{(w,w') \in E} \beta_{k,w}^v \beta_{k,w'}^v exp(\hat{r}_{w,w'} \cdot (u_k^e)^T)}, \quad (10)$$

where, $\hat{r}_{w_1,w_2} = r_{w_1} \oplus r_{w_2}$. $(\cdot)$ and $\oplus$ denote dot product and concatenation operations, respectively.

Therefore, each topic $k$ can be represented by the vector $u_k = u_k^v \oplus u_k^e$. Meanwhile, $\beta_k^v$ and $\beta_k^e$ can be explained as the semantic similarity between the topic and words or edges (the concatenation of word vectors as the edge vector). Along this line, we further define each element of the topic dependency matrix $M$ as the semantic similarity among topics with additional learnable parameter matrix $W \in \mathbb{R}^{3H \times 2Y}$:

$$a_k \oplus b_k = u_k \cdot W, \quad m_{i,j} = \sigma(g(a_i \cdot b_j^T)), \ \forall i,j,k, \ i \neq j, \quad (11)$$

where $\sigma(\cdot)$ is the sigmod function. $a_k, b_k \in \mathbb{R}^Y$ are two transitional vectors for the topic $k$ to compute the possibility of directed edges among different topics. $g(\cdot)$ denotes a vector similarity metric and is set as the consine similarity $g(a,b) = C(a \cdot b/(||a||||b||))$ here, in which $C$ is a hyper-parameter to re-scale the range of $\sigma(g(\cdot))$ to approach $[0,1]$.

### 3.2.2 Encoder

Here, we aim to model the mapping from the observed data $G_d$ to the variational parameters for latent variables $\theta_d$ and $Z_d$. To be specific, we denote $X \in \mathbb{R}^{|V| \times L}$ as the trained and fixed word embeddings for all words in the vocabulary $V$. For the document $d$, $X_d \in \mathbb{R}^{|V_d| \times L}$ is the word embedding set for word set $V_d$. Then, we encode the local information of each node and the global information of the whole graph $G_d$ with one layer of graph neural network, such as GraphConv [38]:

$$h_d = \tanh(GNN(G_d^o, X_d)),$$
$$h_d^G = \tanh(\sum_{n=1}^{N_d} \sigma(f_1(h_{d,n} \oplus x_{d,n})) \odot \tanh(f_2(h_{d,n} \oplus x_{d,n}))), \quad (12)$$

where $h_d \in \mathbb{R}^{|V_d| \times L}$ is the node-level representation vectors, $h_d^G \in \mathbb{R}^L$ is the graph-level representation vector, and $f_*(\cdot)$ is a full connected layer. Self-loops have been added to $G_d^o$ for preserving the current word feature. The second project network is following [29].

Then, the parameters of variational marginals $q(\theta_d|G_d)$ and $q(z_{d,n}|G_d, w_{d,n})$ can be specified as:

$$\mu_d = f_\mu(h_d^G), \ \log \delta_d^2 = f_\delta(h_d^G),$$
$$\varphi_{d,n} = f_\varphi(h_d^G \oplus h_{d,n} \oplus x_{d,n}). \quad (13)$$

### 3.2.3 Decoder

We denote the reconstruction part in Equation 9 (the first line) as $\mathcal{L}_d^R$, which can be deduced as:

$$\mathcal{L}_d^R = \sum_{(n,n') \in E_d^o} \varphi_{d,n}^T \cdot (\log M) \cdot \varphi_{d,n'} + \sum_{(n,n') \notin E_d^o} \varphi_{d,n}^T \cdot \log(1 - M) \cdot \varphi_{d,n'}$$
$$+ \sum_{n=1}^{N_d} \varphi_{d,n}^T \cdot \log \beta_{\cdot,w_n}^v + \sum_{k=1}^K E_{q(Z_d|G_d)}[\log \frac{\sum_{(n,n') \in E_d^o} z_{d,n}^k z_{d,n'}^k \beta_{k,(w_{d,n},w_{d,n'})}^e}{\sum_{(n,n') \in E_d^o} z_{d,n}^k z_{d,n'}^k}], \quad (14)$$

where $z_{d,n}^k = 1$ if $z_{d,n} = k$ otherwise 0. Indeed, the first two terms equal to $E_{q(Z_d|G_d)}[\log p(G_d^o|Z_d)]$ and the other items are derived from $E_{q(Z_d|G_d)}[\log p(V_d|Z_d, G_d^o)]$ (see Appendix A.2) . In practice, the graph $G_d^o$ is often very sparse which causes the unbalance between the first two terms. Here, we follow [26] and re-weight the second terms with $|E_d^o|(\sum_{(n,n') \notin E_d^o} 1)^{-1}$.

Now, the only challenge to compute the ELBO loss $\mathcal{L}_d$ is at how to compute the last term in Eqaution 14, which is the expectation of a log-sum function and can not be computed analytically. Here, we estimate this expectation with Monte Carlo samples from $q(Z_d|G_d)$. Meanwhile, to ensure the differentiability, we use Straight-Through Gumbel-Softmax (STGS) estimator [20] to approximate

the undifferentiable multinomial sampling, which outputs the discrete samples as the approximated multinomial samples in the neural forward pass, but applies a continuous approximation in the backward pass.

We conclude this section with an observation to show that the objective function in LDA can be derived from that of GNTM as a special case. This demonstrates the generalization of our proposed model and the contribution to model the word dependency by representing documents as graphs. The proof can be found in Appendix A.3.

**Corollary 1** *If the word dependency edge $E_d$ of the document $d$ is one empty set, the objective function defined in Equation 9 would reduce into the ELBO in LDA with the Laplace approximation of the Dirichlet distribution.*

## 4 Experiments

Here, we conduct extensive experiments on four benchmark text datasets to evaluate the effectiveness and interpretability of our model. Our code and data are available at `https://github.com/SmilesDZgk/GNTM`.

### 4.1 Experimental Setup

**Datasets:** Our experiments are conducted on four benchmark datasets with varying sizes, including 20 News Groups (**20NG**) [27], Tag My News (**TMN**) [51], the British National Corpus (**BNC**) [10], **Reuters** extracted from the Reuters-21578 dataset. In particular, 20NG and TMN are single-labeled datasets. We follow [16] and only use the title text in TMN to represent the short documents. We also filter out the stop words, and words and dependency edges with low frequency among the whole collection to reduce noise. The statistics and links of these datasets are shown in Appendix A.4.

**Evaluation Metrics:** Here, we focus on measuring the quality of learned topics. Traditionally, perplexity has been used to measure the quality of topic models, but it has been repeatedly shown that perplexity is not a good metric for the qualitative evaluation of topics [39]. Therefore, we adopt another two metrics: topic coherence (**TC**) and topic diversity (**TD**). Given the reference corpus, TC measures the interpretability of a topic by computing the semantic coherence in the most significant words [37]. With the suggestion in [43], we selected two TC measurements to provide a robust evaluation, i.e, Normalized Pointwise Mutual Information (NPMI) [1] and $C_v$ [43]. Specifically, both NPMI and $C_v$ are computed on the top words of each topic with the original document corpus of each dataset as the reference documents. Higher value indicates better interpretability. TD aims to measure how diverse the discovered topics are. We follow [11] and defined TD as the percentage of unique words in the top words. TD closes to 0 indicates redundant topics. TD closes to 1 indicates more varied topics. In addition, we also evaluate our model in a document clustering task on labeled datasets, i.e., 20NG and TMN, to evaluate the quality of the learned topic propitiation. Specifically, following [40], we assign every document the topic with the highest probability as the clustering label and compute **Purity** and Normalized Mutual Information (**NMI**) [45] as metrics. Both of them always range from 0.0 to 1.0, and higher scores reflect better clustering performance.

**Benchmark Models:** We select several state-of-the art-topic model as the baselines, including: 1) **LDA** [5]; 2) **GSM** [34], an NTM model which replaces the Dirichlet-Multinomial parametrization in LDA with Gaussian Softmax; 3) **ProdLDA** [49], an NTM model which keeps the Dirichlet-Multinomial parametrization with a Laplace approximation; 4) **ETM** [11], an NTM mdoel which incorporates word embedding to model topics; 5) **GraphBTM** [61], an NTM model with graph-based word dependency; 6) **iDocNADE** and **iDocNADEe** [16], neural autoregressive topic models with sentence-based word dependency, where iDocNADEe incorporates word embeddings. Note that iDocNADEe constrains the equality of the dimension of word embedding and the number of topics. We can only set the number of topics as 50 and 100, due to the lack of pretraining word-embedding with other dimensions in public. 7)**TopicRNN** [12], a neural autoregressive topic model with word sequence-based dependency. All baselines are implemented carefully with the guidance of their official code.

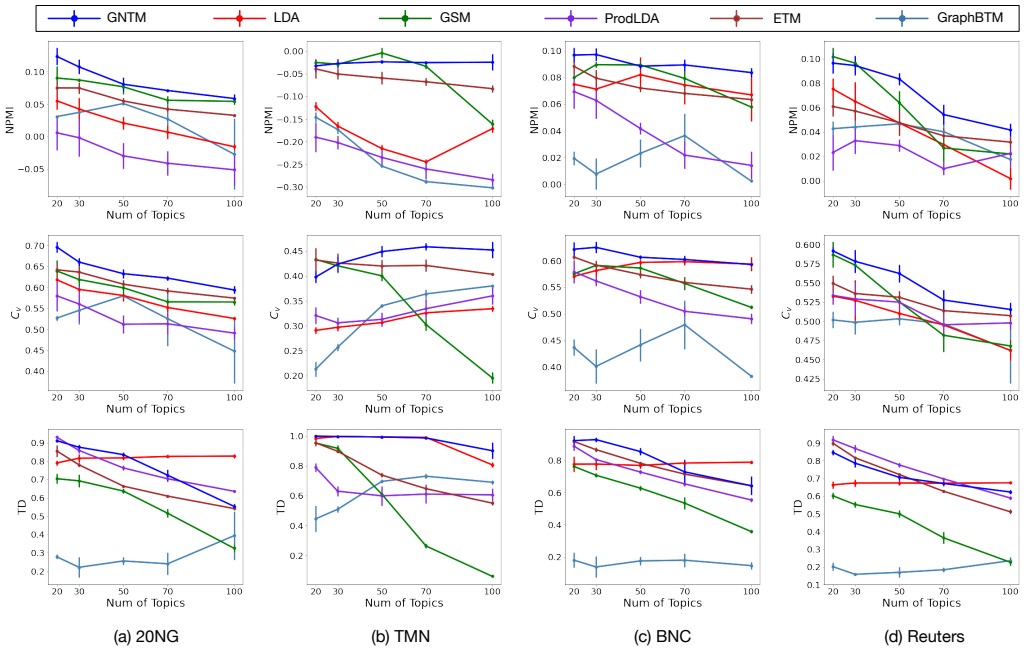

Figure 3: The performance on the topic quality. The first two rows show the topic coherence score, i.e., NPMI and $C_v$, with corresponding topic diversity in the third row, where error bars in each point denote the standard deviation of results on 5 runs. The result of iDcNADE, iDocNADEe, and TopicRNN has been omitted due to the uncompetitiveness. Completed results can be found in Appendix A.7.

**Experiment Settings:** In practice, we construct document graphs with window size $s = 5^3$. In particular, we mix together with the same words in different positions as one node with word frequency to compute the correct likelihood. In other words, we implicitly assume that the same words in one document share the topic assignment, which is widely used in [5, 34, 42, 54]. We set the Dirichlet prior parameter $\alpha = 1$. Following [49], batch-normalization [18] has been used on the inference network for the posterior parameters of $\theta$. We utilized 300-dimensional GloVe word embeddings [41] to fix $X$ (i.e., $L = 300$) in our model and word vectors in ETM. We also set the size of word vectors $\mu^v$ as 300, i.e., $H = 300$. The size of transitional vector $a_k$ and $b_k$ were set as $Y = 64$. For the optimization, We follow [34] and alternately update the decoder parameters with topic representation and the encoder parameters. Only one sample is used in neural variational inference for $\theta_d$ and $Z_d$ if needed. For the optimization, Adam [23] optimizer has been used with the initial learning rate of 0.001 and the linear learning rate decay trick to find the optimal. All quantitative results are based on 5 runs with different random seeds and the average performance is reported without special description. (More experimental details can be found in Appendix A.5.)

## 4.2 Experimental Results

We follow [11] and select the top 10 words with the highest probability in each topic as the representative word set to compute both TC and TD metrics. The results are displayed in Figure 3 with the varying number of topics $K$ from 20 to 100 (see Appendix A.7 for more results based on the different number of top words). We find that our proposed model outperforms other baselines in terms of both $C_v$ and NPMI score with high topic diversity on all datasets. In particular, GSM also achieves the competitive performance on TC metrics, but its TD score is lower than most models. It may be because that, in contrast to Dirichlet prior used in other models, the Gaussian-Softmax distribution in GSM leads to lower sparsity among topic proportions, which causes redundant topics in turn. Meanwhile, although some other baselines may have higher TD score than our model on several datasets with several topic numbers, they cannot achieve high topic coherence. In addition, despite

---

[3]When $s = 5$, the degree of one word is 10, i.e., 5 in-degrees and 5 out-degrees, which is similar to that in Skip-gram with the size of text window as 5 that performs better in empirical experiments [36]. We show the sensitivity analysis of the windows size $s$ in Appendix A.5.

Table 1: The performance on document cluster. We highlight the best and second scores in boldface and with an underline, respectively. The standard deviation can be found in Appendix A.7.

| | | Purity | | | | | NMI | | | | |
|---|---|---|---|---|---|---|---|---|---|---|---|
| | | 20 | 30 | 50 | 70 | 100 | 20 | 30 | 50 | 70 | 100 |
| 20NG | LDA | 0.2980 | 0.3340 | 0.3375 | 0.3510 | 0.3740 | 0.2908 | 0.3013 | 0.2852 | 0.2878 | 0.2858 |
| | GSM | 0.4133 | 0.4379 | 0.4629 | 0.4429 | 0.4210 | 0.4394 | 0.4369 | **0.4433** | **0.4449** | **0.4412** |
| | ProdLDA | 0.3306 | 0.3450 | 0.3641 | 0.3638 | 0.3807 | 0.3405 | 0.3345 | 0.3350 | 0.3298 | 0.3343 |
| | ETM | 0.3496 | 0.4154 | 0.4380 | 0.4510 | 0.4616 | 0.3842 | 0.4227 | 0.4296 | 0.4297 | 0.4356 |
| | GraphBTM | 0.1448 | 0.1210 | 0.1630 | 0.1068 | 0.0992 | 0.1552 | 0.1108 | 0.1807 | 0.0816 | 0.0707 |
| | iDocNADE | 0.2175 | 0.2844 | 0.3064 | 0.3187 | 0.3371 | 0.1128 | 0.1723 | 0.1802 | 0.1901 | 0.1990 |
| | iDocNADEe | - | - | 0.1300 | - | 0.1507 | - | - | 0.1185 | - | 0.1228 |
| | TopicRNN | 0.0728 | 0.0761 | 0.0865 | 0.0920 | 0.1001 | 0.0109 | 0.0141 | 0.0229 | 0.0303 | 0.0400 |
| | GNTM | **0.4500** | **0.4882** | **0.5089** | **0.5090** | **0.5021** | **0.4436** | **0.4419** | 0.4416 | 0.4362 | 0.4371 |
| TMN | LDA | 0.3509 | 0.3692 | 0.3725 | 0.4031 | 0.4228 | 0.0622 | 0.0665 | 0.0754 | 0.0901 | 0.1064 |
| | GSM | 0.5933 | 0.6054 | 0.6184 | 0.5934 | 0.2632 | 0.2848 | 0.2787 | **0.2996** | **0.3246** | 0.0204 |
| | ProdLDA | 0.3141 | 0.2808 | 0.2505 | 0.2535 | 0.2438 | 0.0508 | 0.0334 | 0.0056 | 0.0053 | 0.0000 |
| | ETM | 0.5841 | 0.5967 | 0.6347 | 0.6358 | 0.6420 | 0.2764 | 0.2705 | 0.2829 | 0.2784 | 0.2767 |
| | GraphBTM | 0.2438 | 0.2438 | 0.2438 | 0.2438 | 0.2438 | 0.0000 | 0.0000 | 0.0000 | 0.0000 | 0.0000 |
| | iDocNADE | 0.2712 | 0.3116 | 0.3314 | 0.3512 | 0.3730 | 0.1074 | 0.1436 | 0.1488 | 0.1585 | 0.1701 |
| | iDocNADEe | - | - | 0.2623 | - | 0.4171 | - | - | 0.0332 | - | 0.1728 |
| | TopicRNN | 0.2493 | 0.2537 | 0.2586 | 0.2684 | 0.2775 | 0.0048 | 0.0071 | 0.0108 | 0.0154 | 0.0221 |
| | GNTM | **0.6150** | **0.6252** | **0.6472** | **0.6656** | **0.6773** | **0.2851** | **0.2801** | 0.2789 | 0.2821 | **0.2870** |

considering word dependency, GraphBTM fails to capture the state-of-the-art performance. The reason may be that it construct each document graph on the whole vocabulary and include words that are not in the document as noise. In terms of document clustering with results in Table 1, our model performs the best with a significant gap over other models on Purity metric, while GNTM is also comparative on NMI metric. Those results demonstrates that our model captures both interpretable topics with better quality and good document representations for clustering. We also note that the GraphBTM fails to cluster documents. Especially, the NMI metric has vanished to 0 on the TMN dataset, which indicates that all documents were clustered to one category and GraphBTM cannot capture significant topics with similar semantics to the document labels.

**Visualization:** Since GNTM models topics as multinomial distributions on both the vocabulary and the word dependency edge sets, we can represent each topic $k$ as a graph on the vocabulary $V$ with the edge set $E$ and adjacency matrix $A_k = (\beta_k^v \cdot (\beta_k^v)^T) \odot \beta_k^e$. We show two topics learned by a run of GNTM on News20 with $K = 20$ as examples in the left of Figure 4. First, we can find that the top words in each topic are highly semantically consistent, where topic #12 is about space science and topic #13 is related to computer technology. Second, we observe that the top edges are also highly interpretable with semantic dependency between two word nodes, such as "space station" and "satellite orbit" in topic #12, or "floppy dives" and "floppy disk' in topic #13, which also helps us to reduce the possible ambiguity and understand the specific meaning of words in each specific topic. Third, the graph representation of topics also provides a new possibility to further separate the topic semantic by clustering the word nodes. For example, we utilize the fast unfolding algorithm [6], a widely-used community detection algorithm, to cluster nodes and color different categories. We find that, interestingly, the top words of topic #12 are split into three sub-clusters with different sub-topics related to space research (violet), celestial bodies (green), and space exploration (orange), respectively. Similarly, the top words of the topic #13 are separated into two categories: one is about software or hardware (orange); the other may be about programming development (violet).

In addition, we also show the topic dependency matrix $M$ in the right of Figure 4, where each element $m_{i,j}$ represents the possibility of existing a directed edge from topic $i$ to $j$. We have the following interesting observations. First, there always exist high possibility values between topic #0 with any other topics. It is actually reasonable because topic #0 is represented with several general words, commonly used in any scenario. Second, the diagonal tend to have a high value. It indicates that the words with the same topic should be linked, which is consistent with the widely-used assumption that interdependent (or neighboring) words prefer to have the same topic assignment [15, 30, 32, 59, 60]. Third, we note that the topic dependency matrix also mines the dependency relation between different topics. For example, in our topic sets, the top words of the topic #9 is full of sport-related words, while the topic #8 focuses on the hockey games with keywords related to team or game name, such as "bos" (Boston Bruins), "det" (Detroit Red Wings) and "nhl" (National Hockey League). Considering the strong relation between topic #8 and #9, the value $m_{8,9}$ and $m_{9,8}$ is high.

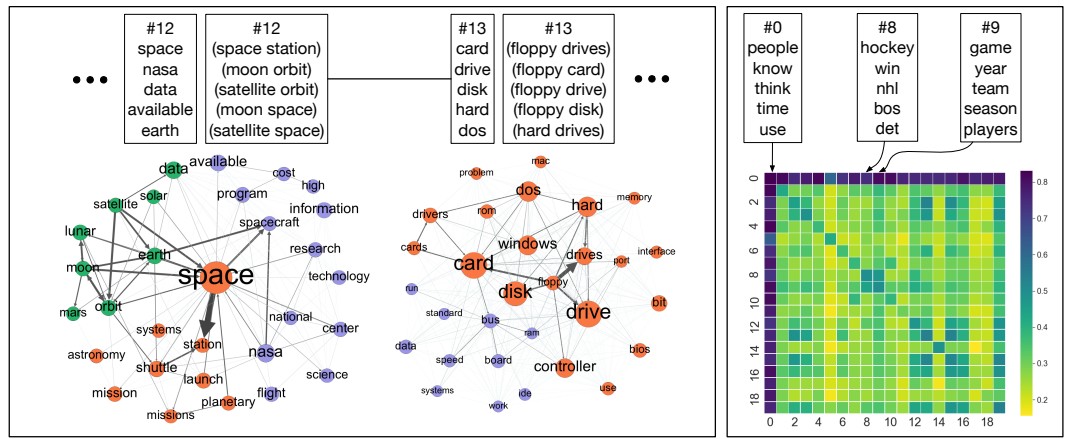

Figure 4: The case study. The left shows topics learned by GNTM with top 5 words, and top 5 edges and corresponding topic graph representation in the top 30 words, where the sizes of nodes and edges are positively correlated to $\beta_k^v$ and $A_k = (\beta_k^v \cdot (\beta_k^v)^T) \odot \beta_k^e$, respectively. Nodes are colored by their cluster labels detected by fast unfloding algorithm [6]. The right displays the heatmap of the topic dependency matrix $M$, with several topics represented by the top 5 words.

## 5 Conclusion

In this paper, we revisited the topic modeling techniques by representing documents as semantic graphs and proposed a Graph Neural Topic Model (GNTM) method. Specifically, we first formulated a well-defined probabilistic generative story to model both the graph structure and word sets with a new concept of topics, i.e., multinomial distributions on both the vocabulary and word dependency edge. Then, a Neural Variational Inference (NVI) approach was proposed to learn our model with graph neural networks to encode the document graphs. Besides, we also have demonstrated that LDA can be derived from GNTM as a special case with similar objective functions. Finally, extensive experiments on four benchmark datasets have clearly validated the effectiveness and interpretability of GNTM compared with state-of-the-art baselines.

**Limitation and Future Work:** Here, we discover word dependency only by linking words in a small sliding window on word sequence. Several other approaches can also construct document graphs, such as dependency parse [33]. An interesting direction of future work is to consider multiple types of word dependency relation together by representing documents as multi-relational graphs.

## 6 Acknowledgement

This research was supported by grants from the National Natural Science Foundation of China (Grant No. 91746301 and 61836013).

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
