# Appendix for "Topic Modeling Revisited: A Document Graph-based Neural Network Perspective"

## A  Appendix

Here, we display additional material to support our content in the main manuscript, including:

- The mathematical notations in Table S1.
- The proof of Theorem 1 in Section A.1.
- The derivation of the objective function in Section A.2.
- The proof of Corollary 1 in Section A.3.
- The dataset statistics in Section A.4.
- Detailed experimental setup with the parameter analysis of the window size $s$ in Section A.5.
- The oblation study of representing topics with a distribution over word edges in Section A.6.
- More experimental results in Section A.7.
- More examples for learned topics in Section A.8.

35th Conference on Neural Information Processing Systems (NeurIPS 2021), Sydney, Australia.

| Symbols | Descriptions |
|---|---|
| $w_{d,n}$ | The $n$-th word in the document $d$. |
| $z_{d,n}$ | The topic assignment for the word $w_{d,n}$. |
| $\theta_d$ | The topic proportion of the document $d$. |
| $N_d$ | The number of words in document $d$. |
| $V_d$ | The word set for the document $d$, i.e., $V_d = \{w_{d,n}\}_{n=1}^{N_d}$. |
| $E_d$ | The word edge set for the document $d$. |
| $Z_d$ | The topic assignment set for the document $d$ |
| $G_d$ | The graph for the document $d$, i.e., $G_d = (V_d, E_d)$. |
| $V_d^o$ | The placeholder set for the document $d$, i.e., $V_d^o = \{1, 2, ..., N_d\}$. |
| $E_d^o$ | The placeholder edge set for the document $d$. |
| $G_d^o$ | The graph structure for the document $d$, i.e., $G_d^o = (V_d^o, E_d^o)$. |
| $E$ | The word edge set on the whole document collection. |
| $V$ | The vocabulary on the whole document collection. |
| $M$ | The topic dependency matrix. |
| $m_{i,j}$ | The element of $M$ at $i$-th row and $j$-th column. |
| $\beta_k^v$ | The word distribution on $V$ for the $k$-th topic. |
| $\beta_k^e$ | The parameter to describe the distribution over word edges for $k$-th topic. |
| $\beta_k$ | The $k$-th topic, i.e., $\beta_k = (\beta_k^v, \beta_k^e)$. |
| $G_{d,k}$ | The graph among the nodes with the topic assignment $k$ in the document $d$. |
| $G_{d,k}^o$ | The graph structure among the nodes with the topic assignment $k$ in the document $d$. |
| $E_{d,k}$ | The word edge set among the nodes with the topic assignment $k$ in the document $d$. |
| $E_{d,k}^o$ | The placeholder edge set among the nodes with the topic assignment $k$ in the document $d$. |
| $\alpha$ | The parameter of the Dirichlet prior for $\theta_d$. |
| $\varphi_{d,n}$ | The paramter of the varaitionl multinomial distribution for $z_{d,n}$. |
| $\mu_d, \delta_d$ | The parameters of the varaitionl logistic normal distribution for $\theta_d$. |
| $r$ | The word vectors for computing topic set $\beta$. |
| $\hat{r}_{w_1,w_2}$ | The concatenation of word vectors, i.e., $\hat{r}_{w_1,w_2} = r_{w_1} \oplus r_{w_2}$. |
| $u^v$ | The topic-word vectors for computing topic set $\beta$. |
| $u^e$ | The topic-edge vectors for computing topic set $\beta$. |
| $W$ | The paramter matrix for computing topic dependency matrix $M$. |
| $a_k, b_k$ | The transitional vectors for the topic k to compute the possibility of directed edges among different topics. |
| $h_d$ | The node-level representation vector set for all word node on the document graph $G_d$. |
| $h_d^G$ | The graph-level representation vector for the whole document graph $G_d$. |
| $K$ | The number of topics. |
| $s$ | The window size for constructing the document graph. |
| $\text{Dir}(\cdot)$ | The Dirichlet distribution. |
| $\mathcal{N}(\cdot)$ | The Gaussian distribution. |
| $\text{Multi}(\cdot)$ | The multinomial distribution. |
| $f_*(\cdot)$ | The full connected layer. |
| $GNN(\cdot, \cdot)$ | The graph neural network. |
| $\sigma(\cdot)$ | The sigmod active function. |
| $\tanh(\cdot)$ | The $\tanh$ active function. |
| $\oplus$ | The vector concatenation operation. |
| $\cdot$ | The dot product operation. |
| $\odot$ | The Hadamard product operation. |

## A.1 Proof of Theorem 1

*Proof.* To prove Theorem 1, we introduce the following lemma:

**Lemma 1.** *Given the topic set $\beta$ defined in Equation 5 and the document graph structure $G_{d,k}^o$ under topic assignment $k$, the probability function $p(V_{d,k}|G_{d,k}^o; \beta_k)$ defined in Equation 6 is a legal probability measure on the vocabulary $V$.*

*Proof.* This proof is mainly inspired by [15]. Obsviouly, based on the definition in Equation 6, $p(V_{d,k}|G_{d,k}^o; \beta_k) > 0$. Then we need to prove the summing of $p(V_{d,k}|G_{d,k}^o; \beta_k)$ on all possible $V_{d,k}$ is equal to 1 for all possible graph structure $G_{d,k}^o$. Here, to provide a completed proof of the lemma, we cluster all situations into three categories based on the edge set $E_{d,k}^o$ of the graph structure, where 1) $E_{d,k}^o = \emptyset$, 2) $|E_{d,k}^o| = 1$, and 3) $|E_{d,k}^o| > 1$, respectively. In the following, we demonstrate that $p(V_{d,k}|G_{d,k}^o; \beta_k)$ is a legal probability measure in each category.

**Category 1:** As the definition in equation 6, we set $\sum_{(w,w') \in E_{d,k}} \beta_{k,(w,w')}^e / |E_{d,k}| = 1$ if $E_{d,k}^o = \emptyset$. Then, $p(V_{d,k}|G_{d,k}^o; \beta_k)$ will reduce to the following function:

$$p(V_{d,k}|G_{d,k}^o, E_{d,k}^o = \emptyset; \beta_k) = p(V_{d,k}|\beta_k) = \prod_{w \in V_{d,k}} \beta_{k,w}^v. \tag{S1}$$

The above function is a legal probability measure over all possible $V_{d,k}$, because we have:

$$\sum_{V_{d,k}} p(V_{d,k}|G_{d,k}^o, E_{d,k}^o = \emptyset; \beta_k) = \sum_{V_{d,k}} \prod_{w \in V_{d,k}} \beta_{k,w}^v = \prod_{n \in V_{d,k}^0} \sum_{w_n \in V} \beta_{k,w}^v = 1 \tag{S2}$$

**Category 2:** Here, we consider the graph structure $G_{d,k}^o$ with only one edge, i.e., $|E_{d,k}^o| = 1$. Without loss of generality, we let $E_{d,k}^o = \{(i,j)\}$ and $V_{d,k}^- = V_{d,k} - \{w_{d,i}, w_{d,j}\}$. Then, we have:

$$p(V_{d,k}|G_{d,k}^o, E_{d,k}^o = \{(i,j)\}; \beta_k) = (\prod_{w \in V_{d,k}^-} \beta_{k,w}^v) \times (\beta_{k,w_{d,i}}^v \beta_{k,w_{d,j}}^v \beta_{k,(w_{d,i},w_{d,j})}^e) \tag{S3}$$

Summing of the above functions over all possible $V_{d,k}$, we have

$$\sum_{V_{d,k}} p(V_{d,k}|G_{d,k}^o, E_{d,k}^o = \{(i,j)\}; \beta_k) =$$

$$(\sum_{V_{d,k}^-} \prod_{w \in V_{d,k}^-} \beta_{k,w}^v) \times (\sum_{w_i, w_j \in V} \beta_{k,w_{d,i}}^v \beta_{k,w_{d,j}}^v \beta_{k,(w_{d,i},w_{d,j})}^e) = (\prod_{w \in V_{d,k}^{o-}} \sum_{w \in V} \beta_{k,w}^v) \times 1 = 1 \tag{S4}$$

**Category 3:** When $|E_{d,k}^o| > 1$, we first have the following derivation:

$$p(V_{d,k}|G_{d,k}^o; \beta_k) = \frac{1}{|E_{d,k}|} \prod_{w \in V_{d,k}} \beta_{k,w}^v \sum_{(w',w'') \in E_{d,k}} \beta_{k,(w,w')}^e =$$

$$\frac{1}{|E_{d,k}^o|} \sum_{(i,j) \in E_{d,k}^o} (\prod_{w \in V_{d,k}} \beta_{k,w}^v) \beta_{k,(w_i,w_j)}^e = \frac{1}{|E_{d,k}^o|} \sum_{(i,j) \in E_{d,k}^o} p(V_{d,k}|G_{d,k}^o, E_{d,k}^o = \{(i,j)\}; \beta_k) \tag{S5}$$

Then, summing the above function over all possible $V_{d,k}$, we have:

$$\sum_{V_{d,k}} p(V_{d,k}|G_{d,k}^o; \beta_k) = \sum_{V_{d,k}} \frac{1}{|E_{d,k}^o|} \sum_{(i,j) \in E_{d,k}^o} p(V_{d,k}|G_{d,k}^o, E_{d,k}^o = \{(i,j)\}; \beta_k) =$$

$$\frac{1}{|E_{d,k}^o|} \sum_{(i,j) \in E_{d,k}^o} \sum_{V_{d,k}} p(V_{d,k}|G_{d,k}^o, E_{d,k}^o = \{(i,j)\}; \beta_k) = \frac{1}{|E_{d,k}^o|} \sum_{(i,j) \in E_{d,k}^o} 1 = 1. \tag{S6}$$

where the above derivation is based on the discussion in situation 2. $\square$

The above lemma tells us that each factor of $p(V_d|Z_d, G_d^o; \beta_k)$ in equation 6, i.e., $p(V_{d,k}|G_{d,k}^o; \beta_k)$, is a legal possibility measure. Meanwhile, considering those factors are independent to each other when given topic assignment set $Z_d$, we have the conclusion that $p(V_{d,k}|G_{d,k}^o; \beta_k)$ is also a legal possibility measure. $\square$

## A.2 The Derivation of the Objective Function

Here we first provide the detailed derivation of Equation 9 with the guidance of variation inference algorithm [10]. Then, we further simplify each item among it.

**Detailed Derivation of Equation 9:** Given the joint model $p(G_d, \theta_d, Z_d; \alpha)$ of $G_d$ in Equation 3 and the variational family $p(\theta_d, Z_d|G_d)$ in Equation 7, the objective of inferring GNTM is minimize the KL divergence between approximate posterior $q(\theta_d, Z_d|G_d)$ and true posterior $p(\theta_d, Z_d|G_d)$ for all latent variables:

$$
\arg \min_{Z_d, \theta_d} KL(q(\theta_d, Z_d|G_d)||p(\theta_d, Z_d|G_d))
$$
$$
= E_{q(\theta_d, Z_d|G_d)}[\log q(\theta_d, Z_d|G_d) - \log p(\theta_d, Z_d|G_d)] \quad \text{(S7)}
$$
$$
= E_{q(\theta_d, Z_d|G_d)}[\log q(\theta_d, Z_d|G_d) - \log p(G_d|\theta_d, Z_d) - \log p(\theta_d) - \log p(Z_d)] + \log p(G_d)
$$
$$
= -\mathcal{L}_d + \log p(G_d).
$$

Due to the $\log p(G_d)$ is constant, the above objective can be transformed to maximize the follow Evidence Lower Bound (ELBO) of $\log p(G_d)$:

$$
\mathcal{L}_d = E_{q(\theta_d, Z_d|G_d)}[\log p(G_d|\theta_d, Z_d)] - KL[q(\theta_d|G_d)||p(\theta)] - E_{q(\theta_d|G_d)}[KL(q(Z_d|G_d)||p(Z_d))]
$$
$$
= E_{q(Z_d|G_d)}[\log p(G_d^o|Z_d; M)] + E_{q(Z_d|G_d)}[\log p(V_d|Z_d, G_d^o; \beta)]
$$
$$
- KL[q(\theta_d|G_d)|p(\theta_d)] - E_{q(\theta_d|G_d)}[\sum_{n=1}^{N_d} KL[q(z_{d,n}|G_d, w_{d,n})||p(z_{d,n}|\theta_d)]]
$$
$$
= \mathcal{L}_d^1 + \mathcal{L}_d^2 - \mathcal{L}_d^3 - \mathcal{L}_d^4,
$$
$$\text{(S8)}$$

where $\mathcal{L}_d$ can be split into four terms , denoted as $\mathcal{L}_d^1, \mathcal{L}_d^2, \mathcal{L}_d^3$, and $\mathcal{L}_d^4$, respectively.

**Simplification for each item in Equation 9:** As for $\mathcal{L}_d^1$, we can derive:

$$
\mathcal{L}_d^1 = E_{q(Z_d|G_d)}[\sum_{(n,n')\in E_d^o} \log m_{z_{d,n}, z_{d,n'}} + \sum_{(n,n')\notin E_d^o} \log(1 - m_{z_{d,n}, z_{d,n'}})]
$$
$$
= \sum_{(n,n')\in E_d^o} E_{q(z_{d,n}, z_{d,n'}|G_d)}[\log m_{z_{d,n}, z_{d,n'}}] + \sum_{(n,n')\notin E_d^o} E_{q(z_{d,n}, z_{d,n'}|G_d)}[\log(1 - m_{z_{d,n}, z_{d,n'}})]
$$
$$
= \sum_{(n,n')\in E_d^o} \varphi_{d,n}^T \cdot (\log M) \cdot \varphi_{d,n'} + \sum_{(n,n')\notin E_d^o} \varphi_{d,n}^T \cdot \log(1 - M) \cdot \varphi_{d,n'}.
$$
$$\text{(S9)}$$

Note that, in practice, to relax the unbalance between the two terms in above equation, we follow [12] and re-weight the second term with $|E_d^o|(\sum_{(n,n')\notin E_d^o} 1)^{-1}$. That is:

$$
\mathcal{L}_d^1 = \sum_{(n,n')\in E_d^o} \varphi_{d,n}^T \cdot (\log M) \cdot \varphi_{d,n'} + \frac{|E_d^o|}{\sum_{(n,n')\notin E_d^o} 1} \sum_{(n,n')\notin E_d^o} \varphi_{d,n}^T \cdot \log(1 - M) \cdot \varphi_{d,n'}.
$$
$$\text{(S10)}$$

As for $\mathcal{L}_d^2$, we can derive:

$$
\mathcal{L}_d^2 = E_{q(Z_d|G_d)}[\sum_{k=1}^{K} (\sum_{w\in V_{d,k}} \log \beta_{k,w}^v + \log \frac{\sum_{(w,w')\in E_{d,k}} \beta_{k,(w,w')}^e}{|E_{d,k}|})]
$$
$$
= E_{q(Z_d|G_d)}[\sum_{n=1}^{N_d} \log \beta_{z_{d,n},w}^v] + E_{q(Z_d|G_d)}[\sum_{k=1}^{K} \log \frac{\sum_{(w,w')\in E_{d,k}} \beta_{k,(w,w')}^e}{|E_{d,k}|})] \quad \text{(S11)}
$$
$$
= \sum_{n=1}^{N_d} \varphi_{d,n}^T \cdot \log \beta_{\cdot,w_n}^v + \sum_{k=1}^{K} E_{q(Z_d|G_d)}[\log \frac{\sum_{(n,n')\in E_d^o} z_{d,n}^k z_{d,n'}^k \beta_{k,(w_{d,n}, w_{d,n'})}^e}{\sum_{(n,n')\in E_d^o} z_{d,n}^k z_{d,n'}^k}].
$$

As for $\mathcal{L}_d^3$, we follow the idea in Laplace approximation [18] by rewriting $p(\theta_d) = Dir(\alpha)$ as the logistic normal distribution with mean $\mu^0$ and covariance matrix $\Sigma^0$. each dimension of them can be computed by:

$$\mu_{0,k} = \log \alpha_k - \frac{1}{K}\sum_i \alpha_i, \;\; \Sigma_{0,k,k} = \frac{1}{\alpha_k}(1 - \frac{2}{K}) + \frac{1}{K^2}\sum_i \frac{1}{\alpha_i}. \tag{S12}$$

Then, the KL divergence $\mathcal{L}_d^3$ can be computed between two logistic normal distribution:

$$\mathcal{L}_d^3 = \frac{1}{2}\{Tr(\Sigma_0^{-1}\Sigma_d) + (\mu_d - \mu_0)^T \Sigma_0^{-1}(\mu_d - \mu_0) - K + \log\frac{|\Sigma_0|}{|\Sigma_d|}\}, \tag{S13}$$

where diagonal covariance $\Sigma_d = diag(\delta_d)$ returns a square diagonal matrix with the elements of vector $\delta_d$ on the main diagonal.

In term of $\mathcal{L}_d^4$, we have:

$$\mathcal{L}_d^4 = \sum_{n=1}^{N_d} \varphi_{d,n} \cdot (\log \varphi_{d,n} - E_{q(\theta_d|G_d)}[\log \theta_d]), \tag{S14}$$

where $E_{q(\theta_d|G_d)}[\log \theta_d])$ can also be computed analytically if $q(\theta_d|G_d)$ is Dirichlet distribution because $\log \theta_d$ is the sufficient statistics. However, we still use the sampling strategy to approximate its value, which is more general and can also be applied for more flexible situations without conjugate distributions, such as [1, 20].

---

**Algorithm 1:** The Generative Process of Graph Neural Topic Model

1. Draw the topic proportion of the document $d$, $\theta_d \sim Dir(\alpha)$.
2. Draw topic for $n$-th word $z_{d,n} \sim \text{Multi}(\theta_d)$, $n = 1, 2, ..., N$.
3. Draw the graph structure $G_d^o \sim p(G_d^o|Z_d)$ using Equation 4.
4. Draw the word set $p(V_d|G_d^o, Z_d, \beta)$ using Equation 6.

---

**Algorithm 2:** The Generative Process of Latent Dirichlet Allocation

1. Draw the topic proportion of the document $d$, $\theta_d \sim Dir(\alpha)$.
2. Draw topic for $n$-th word $z_{d,n} \sim \text{Multi}(\theta_d)$, $n = 1, 2, ..., N$.
3. Draw the word set $p(V_d|Z_d, \beta) = \prod_{n=1}^{N_d} p(w_{d,n}|z_{d,n}, \beta^v) = \prod_{n=1}^{N_d} \text{Multi}(\beta_{z_{d,n}}^v)$.

---

### A.3 Proof of Corollary 1

*Proof.* Here, to provide one intuitive comparison, we first show the generative processes of GNTM and Latent Dirichlet Allocation (LDA) [2] in Algorithm 1 and 2, respectively. Then, for each document $d$, we refer to the variational family $q(\theta_d, Z_d|V_d)$ and objective function, i.e., ELBO, of LDA under our mathematical notation:

$$q(\theta_d, Z_d|V_d) = q(\theta_d; \gamma_d) \prod_{n=1}^{N_d} q(z_{d,n}; \varphi_{d,n})$$

$$\mathcal{L}_d^{LDA} = E_{q(Z_d)}[\log q(V_d|Z_d; \beta^v)] - KL[q(\theta_d)||p(\theta_d)] - E_{q(\theta)}[\sum_{n=1}^{N_d} KL[q(z_{d,n})||p(z_{d,n}|\theta_d)]], \tag{S15}$$

where $q(z_{d,n}; \varphi_{d,n})$ is a multinomial distribution with parameter $\varphi_{d,n}$ and $q(\theta_d; \gamma_d)$ is a Dirichlet distribution with parameter $\gamma_d$; The first term can be further derived as $\sum_{n=1}^{N_d} \varphi_{d,n} \cdot \log \beta_{\cdot,w_n}^v$.

By comparing the Equation S8 and S15, we can find that $\mathcal{L}_d^3$ and $\mathcal{L}_d^4$ is same as the last two terms in $\mathcal{L}_d^{LDA}$ except that $q(\theta_d|G_d)$ is the Laplace approximation of the Dirichlet distribution.

In addition, we can find that if $E_{d,k} = \emptyset$, both terms in Equation S10 will vanish to 0. In other words, $\mathcal{L}_d^1$ will reduce to 0 if $E_{d,k} = \emptyset$. Meanwhile, based on the discussion about Equation S1, we can find that $\mathcal{L}_d^2$ will reduce to the following function:

$$\mathcal{L}_d^2(E_d^o = \emptyset) = E_{q(Z_d|G_d)}[\sum_{k=1}^{K} \log p(V_{d,k}|G_{d,k}^o, E_{d,k}^o = \emptyset; \beta_k)]$$

$$= E_{q(Z_d|G_d)}[\sum_{k=1}^{K} \sum_{w \in V_{d,k}} \log \beta_{k,w}^v] = E_{q(Z_d|G_d)}[\sum_{n=1}^{N_d} \log \beta_{z_{d,n},w}^v] = \sum_{n=1}^{N_d} \varphi_{d,n} \cdot \log \beta_{\cdot,w}^v,$$

(S16)

which indicates that $\mathcal{L}_d^2$ reduce to the first term in $\mathcal{L}_d^{LDA}$.

In sum, we can conclude that the objective of GNTM $\mathcal{L}_d$ in Equation 9 will be same as the ELBO $L_d^{LDA}$ of LDA if the edge set $E_d$ of the document graph $G_d$ is a empty set, except that we approach the Dirichlet distribution by the Laplace approximation. $\square$

## A.4   Dataset Statistics

The statistics of the datasets in experiments are shown in Table S2. The links of the datasets can be found in the footnote: 20 News Group (**20NG**) [13][1], Tag My News (**TMN**) [19][2], the British National Corpus (**BNC**) [4][3], **Reuters** extracted from the Reuters-21578 dataset [4]. In particular, the stop word list is same as that in genism package[5]. The frequency threshold to filter out low-frequency words is set as 20 for 20NS, 5 for TMN, 120 for BNC, and 10 for Reuters. The frequency threshold to filter out low-frequency dependency edges is set as 10 for 20NG, 3 for TMN, 40 for BNC, and 15 for Reuters.

Table S2: The statistics of the datasets.

|  | Docs | Data split (train/val/test) | Vocabulary | Word token | Edge set | Edge token | Label |
|---|---|---|---|---|---|---|---|
| 20NG | 17679 | 7105/3517/7057 | 9161 | 1265888 | 33758 | 665502 | 20 |
| TMN | 26171 | 18973/2428/4770 | 5572 | 138875 | 6799 | 44658 | 7 |
| BNC | 16963 | 14966/998/999 | 9401 | 6736890 | 28756 | 2089104 | N/A |
| Reuters | 10727 | 6757/965/3005 | 5907 | 675982 | 16668 | 857146 | N/A |

## A.5   Detailed Experimental Setup

We reproduced all baselines with the guidance of original papers and the codes provided by the original authors or other widely used sources. The code links of them can be found in the footnote: 1) **LDA** [2][6]; 2) **GSM** [16][7]; 3) **ProdLDA** [18][8]; 4) **ETM** [6][9]; 5) **GraphBTM** [21][10]; 6) **iDocNADE** and **iDocNADEe** [8][11]; 7) **TopicRNN** [7][12]; Actually, we also tried to reproduce the **GaussianLDA** [5] as a baseline, which also introduces word embedding into topic modeling, like ETM and GNTM. However, GaussianLDA is highly time-costing in practice if there is a large

---

[1]http://qwone.com/ jason/20Newsgroups/

[2]http://acube.di.unipi.it/tmn- dataset/

[3]https://www.sketchengine.eu/british-national-corpus/

[4]https://trec.nist.gov/data/reuters/reuters.html

[5]https://radimrehurek.com/gensim/

[6]https://radimrehurek.com/gensim/models/ldamodel.html

[7]https://github.com/linkstrife/NVDM-GSM

[8]https://github.com/akashgit/autoencoding_vi_for_topic_models

[9]https://github.com/adjidieng/ETM

[10]https://github.com/valdersoul/GraphBTM

[11]https://github.com/pgcool/iDocNADEe

[12]https://github.com/narratives-of-war/topic-rnn

vocabulary size, like that in our case. Without specific description, the parameters of baselines are the same as those recommended by the original paper or the official code. In particular, we set word embedding size in GSM and ETM as 300, same as that in GNTM, to provide a fair comparison.

As for GNTM, we set all main hyper-parameters as $s = 5, \alpha = 1.0, L = H = 300$, and $Y = 64$. In particular, Figure S1 shows the parameter analysis of the window size $s$, which is the most important parameter, based on 20NG dataset with the fixing number of topics $K = 20$. As the result shows, we can find that when $s = 5$, GNTM achieves best topic coherence both on NPMI and $C_v$ metrics with high topic diversity. Therefore, we set $s = 5$ in our experiments. The Glove word embeddings [17] used in GNTM and ETM can be downloaded in this link [13]. In the optimization, we followed [16] and alternately updated the decoder parameters with topic representation and the encoder parameters. Only one sample is used in neural variational inference for $\theta_d$ and $Z_d$ if needed. We use Adam [11] optimizer with the initial learning rate of 0.001 and decay it by a factor of 2 if the validation loss has not improved in 5 epochs and terminate training once the learning rate has decayed a total of 5 times. The batch size is set as 100. In particular, we pre-trained our model without the restructure of the graph structure in the first 15 epochs or 2000 iterations, which benefits for the robustness of our model empirically. Following [9], the temperature $\tau$ for the STGS estimator [9] is annealed from 1.0 to 0.3 using the schedule $\tau = max(0.3, exp(-\eta \text{iter}))$ of the global training step iter, where $\tau$ is updated every 1000 steps and $\eta = 0.00003$.

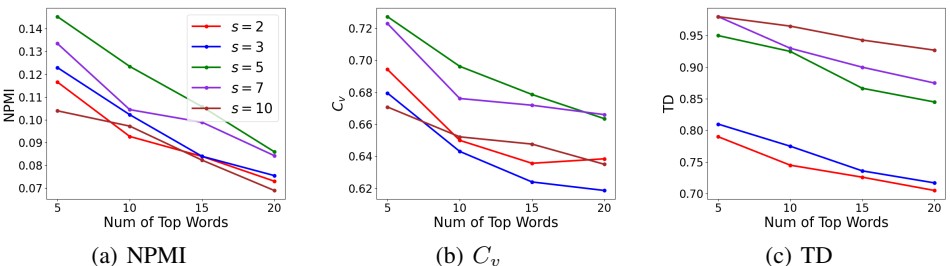

(a) NPMI          (b) $C_v$          (c) TD

Figure S1: The parameter analysis of the window size $s$ on 20NG dataset with $K = 20$. We show both topic coherence measurements, i.e., NPMI and $C_v$, and topic diversity based on varying numbers of top words of each topic as the metrics.

## A.6 The oblation study

Here, we explore the impact of representing topics with a distribution over word dependency edges on the interpretability in our model. To be specific, we construct a variant of GNTM by removing the distribution over edges from topics, namely GNTM w/o $\beta^e$. Different from the description in Section 3.1.2, we assume that words are independent form each other given the topic assignments $Z_d$. In other words, different from Equation (6), we formulate $P(V_d | Z_d, G_d^o)$ as:

$$p(V_d | Z_d, G_d^o; \beta^v) = \prod_{n=1}^{N_d} \beta_{z_{d,n}, w_{d,n}}^v. \tag{S17}$$

Then, we can re-derive the loss function based on Equation (9). The experimental results on 20NG dataset are showed in Table S3. It demonstrates that removing the distribution over edges from topics causes the decrease of interpretability of topics.

## A.7 More Experimental Results

Here, we show more experimental results on topic quality with varying top words in Table S4-S15 and detailed results on document cluster with standard deviation in Table S16 and S17. Same as the description in Section 4.2, we can find that GNTM outperforms other baselines in terms of topic coherence with high topic diversity under all settings consistently. Note that, although iDcoNADe can outperform GNTM on TMN dataset based on $C_v$ metric with top 10, 15 or 20 words, I still think

---

[13]https://nlp.stanford.edu/projects/glove/

Table S3: The oblation study on the impact of representing topics with a distribution over edges.

| | Num. of topic | 20 | 30 | 50 | 70 | 100 |
|---|---|---|---|---|---|---|
| NPMI | GNTM | 0.1235 | 0.1074 | 0.0807 | 0.0710 | 0.0587 |
| | GNTM w/o $\beta^e$ | 0.0994 | 0.0953 | 0.0791 | 0.0701 | 0.0386 |
| $C_v$ | GNTM | 0.6962 | 0.6605 | 0.6328 | 0.6222 | 0.5939 |
| | GNTM w/o $\beta^e$ | 0.6479 | 0.6445 | 0.6055 | 0.5924 | 0.5532 |
| TD | GNTM | 0.9110 | 0.8759 | 0.8352 | 0.7231 | 0.5530 |
| | GNTM w/o $\beta^e$ | 0.9000 | 0.9000 | 0.7700 | 0.4900 | 0.3700 |

iDcoNADe is uncomparative due to the unignored gap compared with most of other baselines on NPMI metric, which has been demonstrated to be highly consistent with the human evaluated the quality of the topics [14], and topic diversity, where lower value indicates redundant topics.

Table S4: The performance of topic ocherence based on NPMI metric with top 5 words.

| | 20NG | | | | | TMN | | | | |
|---|---|---|---|---|---|---|---|---|---|---|
| | 20 | 30 | 50 | 70 | 100 | 20 | 30 | 50 | 70 | 100 |
| LDA | $0.0856_{\pm0.0207}$ | $0.0617_{\pm0.0217}$ | $0.0541_{\pm0.0156}$ | $0.0503_{\pm0.0142}$ | $0.0305_{\pm0.0119}$ | $-0.0590_{\pm0.0090}$ | $-0.1045_{\pm0.0223}$ | $-0.1359_{\pm0.0087}$ | $-0.1719_{\pm0.0097}$ | $-0.1036_{\pm0.0132}$ |
| GSM | $0.1049_{\pm0.0257}$ | $0.1032_{\pm0.0208}$ | $0.1046_{\pm0.0129}$ | $0.0846_{\pm0.0114}$ | $0.0778_{\pm0.0063}$ | $0.0217_{\pm0.0346}$ | $0.0299_{\pm0.0152}$ | $0.0291_{\pm0.0178}$ | $-0.0246_{\pm0.0108}$ | $-0.1776_{\pm0.0182}$ |
| ProdLDA | $0.0544_{\pm0.0544}$ | $0.0450_{\pm0.0349}$ | $0.0167_{\pm0.0350}$ | $0.0071_{\pm0.0220}$ | $-0.0108_{\pm0.0283}$ | $-0.1292_{\pm0.0446}$ | $-0.1802_{\pm0.0196}$ | $-0.2220_{\pm0.0226}$ | $-0.2482_{\pm0.0248}$ | $-0.2656_{\pm0.0164}$ |
| ETM | $0.0912_{\pm0.0135}$ | $0.0866_{\pm0.0092}$ | $0.0730_{\pm0.0087}$ | $0.0591_{\pm0.0067}$ | $0.0555_{\pm0.0062}$ | $0.0016_{\pm0.0091}$ | $0.0099_{\pm0.0031}$ | $-0.0062_{\pm0.0121}$ | $-0.0255_{\pm0.0107}$ | $-0.0369_{\pm0.0088}$ |
| GraphBTM | $0.0383_{\pm0.0030}$ | $0.0448_{\pm0.0188}$ | $0.0646_{\pm0.0029}$ | $0.0366_{\pm0.0321}$ | $-0.0062_{\pm0.0578}$ | $-0.1534_{\pm0.0174}$ | $-0.1552_{\pm0.0073}$ | $-0.2438_{\pm0.0131}$ | $-0.2831_{\pm0.0148}$ | $-0.2922_{\pm0.0065}$ |
| iDocNADE | $-0.0608_{\pm0.0397}$ | $-0.0801_{\pm0.0410}$ | $-0.1141_{\pm0.0393}$ | $-0.1290_{\pm0.0347}$ | $-0.1550_{\pm0.0195}$ | $-0.5393_{\pm0.0372}$ | $-0.5180_{\pm0.0343}$ | $-0.4840_{\pm0.0255}$ | $-0.4588_{\pm0.0241}$ | $-0.4445_{\pm0.0199}$ |
| iDocNADEe | - | - | $-0.0236_{\pm0.0578}$ | - | $-0.0853_{\pm0.0533}$ | - | - | - | - | $-0.4569_{\pm0.0118}$ |
| TopicRNN | $-0.1403_{\pm0.0124}$ | $-0.1956_{\pm0.0236}$ | $-0.2795_{\pm0.0088}$ | $-0.3036_{\pm0.0092}$ | $-0.3159_{\pm0.0143}$ | $-0.4229_{\pm0.0354}$ | $-0.4100_{\pm0.0151}$ | $-0.4063_{\pm0.0270}$ | $-0.4066_{\pm0.0252}$ | $-0.4119_{\pm0.0096}$ |
| GNTM | $0.1453_{\pm0.0128}$ | $0.1284_{\pm0.0073}$ | $0.1084_{\pm0.0190}$ | $0.1049_{\pm0.0026}$ | $0.0888_{\pm0.0070}$ | $-0.0192_{\pm0.0143}$ | $0.0005_{\pm0.0151}$ | $0.0284_{\pm0.0074}$ | $0.0428_{\pm0.0071}$ | $0.0607_{\pm0.0178}$ |

| | BNC | | | | | Reuters | | | | |
|---|---|---|---|---|---|---|---|---|---|---|
| | 20 | 30 | 50 | 70 | 100 | 20 | 30 | 50 | 70 | 100 |
| LDA | $0.0929_{\pm0.0083}$ | $0.0886_{\pm0.0115}$ | $0.1077_{\pm0.0160}$ | $0.1087_{\pm0.0201}$ | $0.1025_{\pm0.0186}$ | $0.0916_{\pm0.0110}$ | $0.0994_{\pm0.0176}$ | $0.0866_{\pm0.0127}$ | $0.0769_{\pm0.0110}$ | $0.0518_{\pm0.0177}$ |
| GSM | $0.0989_{\pm0.0031}$ | $0.1138_{\pm0.0021}$ | $0.1179_{\pm0.0061}$ | $0.0935_{\pm0.0191}$ | $0.0750_{\pm0.0037}$ | $0.1284_{\pm0.0080}$ | $0.1305_{\pm0.0079}$ | $0.0968_{\pm0.0178}$ | $0.0637_{\pm0.0165}$ | $0.0561_{\pm0.0080}$ |
| ProdLDA | $0.0950_{\pm0.0108}$ | $0.0902_{\pm0.0184}$ | $0.0807_{\pm0.0064}$ | $0.0471_{\pm0.0128}$ | $0.0373_{\pm0.0070}$ | $0.0842_{\pm0.0230}$ | $0.0926_{\pm0.0194}$ | $0.0983_{\pm0.0126}$ | $0.0665_{\pm0.0045}$ | $0.0693_{\pm0.0214}$ |
| ETM | $0.1055_{\pm0.0071}$ | $0.0934_{\pm0.0030}$ | $0.0874_{\pm0.0030}$ | $0.0821_{\pm0.0067}$ | $0.0741_{\pm0.0046}$ | $0.1040_{\pm0.0127}$ | $0.0878_{\pm0.0125}$ | $0.0852_{\pm0.0098}$ | $0.0719_{\pm0.0043}$ | $0.0686_{\pm0.0048}$ |
| GraphBTM | $0.0173_{\pm0.0088}$ | $0.0077_{\pm0.0101}$ | $0.0291_{\pm0.0121}$ | $0.0401_{\pm0.0173}$ | $0.0057_{\pm0.0014}$ | $0.0762_{\pm0.0047}$ | $0.0711_{\pm0.0118}$ | $0.0706_{\pm0.0086}$ | $0.0748_{\pm0.0164}$ | $0.0552_{\pm0.0216}$ |
| iDocNADE | $-0.0146_{\pm0.0487}$ | $-0.0321_{\pm0.0378}$ | $-0.0921_{\pm0.0350}$ | $-0.1257_{\pm0.0159}$ | $-0.1370_{\pm0.0142}$ | $-0.0517_{\pm0.0420}$ | $-0.1184_{\pm0.0574}$ | $-0.1652_{\pm0.0445}$ | $-0.1913_{\pm0.0214}$ | $-0.2176_{\pm0.0113}$ |
| iDocNADEe | - | - | $-0.0687_{\pm0.0145}$ | - | $-0.1506_{\pm0.0115}$ | - | - | $-0.1369_{\pm0.0235}$ | - | $-0.2299_{\pm0.0039}$ |
| TopicRNN | $-0.0599_{\pm0.0326}$ | $-0.1439_{\pm0.0304}$ | $-0.1934_{\pm0.0150}$ | $-0.2318_{\pm0.0124}$ | $-0.2607_{\pm0.0124}$ | $-0.2290_{\pm0.0462}$ | $-0.2637_{\pm0.0169}$ | $-0.3218_{\pm0.0180}$ | $-0.3367_{\pm0.0186}$ | $-0.3553_{\pm0.0111}$ |
| GNTM | $0.1268_{\pm0.0126}$ | $0.1179_{\pm0.0065}$ | $0.1107_{\pm0.0059}$ | $0.1122_{\pm0.0058}$ | $0.1048_{\pm0.0040}$ | $0.1392_{\pm0.0160}$ | $0.1445_{\pm0.0173}$ | $0.1395_{\pm0.0061}$ | $0.1171_{\pm0.0075}$ | $0.0972_{\pm0.0088}$ |

Table S5: The performance of topic coherence based on $C_v$ metric with top 5 words.

| | 20NG | | | | | TMN | | | | |
|---|---|---|---|---|---|---|---|---|---|---|
| | 20 | 30 | 50 | 70 | 100 | 20 | 30 | 50 | 70 | 100 |
| LDA | $0.6825_{\pm0.0251}$ | $0.6486_{\pm0.0289}$ | $0.6437_{\pm0.0128}$ | $0.6251_{\pm0.0159}$ | $0.6087_{\pm0.0119}$ | $0.4047_{\pm0.0149}$ | $0.3841_{\pm0.0185}$ | $0.3592_{\pm0.0102}$ | $0.3419_{\pm0.0093}$ | $0.4148_{\pm0.0113}$ |
| GSM | $0.6795_{\pm0.0270}$ | $0.6634_{\pm0.0134}$ | $0.6526_{\pm0.0163}$ | $0.6276_{\pm0.0088}$ | $0.6373_{\pm0.0065}$ | $0.5152_{\pm0.0339}$ | $0.5274_{\pm0.0133}$ | $0.5060_{\pm0.0239}$ | $0.4056_{\pm0.0165}$ | $0.1770_{\pm0.0209}$ |
| ProdLDA | $0.6212_{\pm0.0611}$ | $0.6136_{\pm0.0489}$ | $0.5823_{\pm0.0313}$ | $0.5820_{\pm0.0261}$ | $0.5596_{\pm0.0180}$ | $0.3601_{\pm0.0330}$ | $0.3231_{\pm0.0126}$ | $0.2887_{\pm0.0098}$ | $0.2883_{\pm0.0108}$ | $0.2841_{\pm0.0055}$ |
| ETM | $0.6779_{\pm0.0229}$ | $0.6690_{\pm0.0165}$ | $0.6604_{\pm0.0065}$ | $0.6420_{\pm0.0086}$ | $0.6340_{\pm0.0092}$ | $0.5101_{\pm0.0244}$ | $0.5223_{\pm0.0062}$ | $0.5036_{\pm0.0157}$ | $0.4946_{\pm0.0181}$ | $0.4797_{\pm0.0081}$ |
| GraphBTM | $0.6010_{\pm0.0042}$ | $0.6073_{\pm0.0252}$ | $0.6369_{\pm0.0083}$ | $0.5909_{\pm0.0477}$ | $0.5467_{\pm0.0048}$ | $0.2237_{\pm0.0102}$ | $0.2584_{\pm0.0110}$ | $0.2840_{\pm0.0072}$ | $0.2742_{\pm0.0045}$ | $0.2762_{\pm0.0074}$ |
| iDocNADE | $0.4769_{\pm0.0288}$ | $0.4526_{\pm0.0290}$ | $0.4283_{\pm0.0286}$ | $0.4119_{\pm0.0163}$ | $0.3901_{\pm0.0074}$ | $0.3779_{\pm0.0211}$ | $0.3613_{\pm0.0164}$ | $0.3543_{\pm0.0097}$ | $0.3471_{\pm0.0078}$ | $0.3427_{\pm0.0078}$ |
| iDocNADEe | - | - | $0.4716_{\pm0.0464}$ | - | $0.4195_{\pm0.0378}$ | - | - | $0.3393_{\pm0.0078}$ | - | $0.3284_{\pm0.0096}$ |
| TopicRNN | $0.4546_{\pm0.0313}$ | $0.3939_{\pm0.0251}$ | $0.3231_{\pm0.0073}$ | $0.3040_{\pm0.0079}$ | $0.3061_{\pm0.0130}$ | $0.3467_{\pm0.0251}$ | $0.3326_{\pm0.0146}$ | $0.3181_{\pm0.0153}$ | $0.3122_{\pm0.0070}$ | $0.3097_{\pm0.0096}$ |
| GNTM | $0.7272_{\pm0.0118}$ | $0.6951_{\pm0.0054}$ | $0.6749_{\pm0.0158}$ | $0.6752_{\pm0.0081}$ | $0.6584_{\pm0.0066}$ | $0.4819_{\pm0.0069}$ | $0.5208_{\pm0.0168}$ | $0.5610_{\pm0.0121}$ | $0.5784_{\pm0.0090}$ | $0.5938_{\pm0.0173}$ |

| | BNC | | | | | Reuters | | | | |
|---|---|---|---|---|---|---|---|---|---|---|
| | 20 | 30 | 50 | 70 | 100 | 20 | 30 | 50 | 70 | 100 |
| LDA | $0.6376_{\pm0.0086}$ | $0.6394_{\pm0.0120}$ | $0.6639_{\pm0.0063}$ | $0.6669_{\pm0.0078}$ | $0.6638_{\pm0.0037}$ | $0.5862_{\pm0.0080}$ | $0.5985_{\pm0.0203}$ | $0.5803_{\pm0.0066}$ | $0.5703_{\pm0.0094}$ | $0.5494_{\pm0.0150}$ |
| GSM | $0.6375_{\pm0.0046}$ | $0.6540_{\pm0.0042}$ | $0.6573_{\pm0.0097}$ | $0.6300_{\pm0.0136}$ | $0.5966_{\pm0.0040}$ | $0.6489_{\pm0.0085}$ | $0.6461_{\pm0.0082}$ | $0.6083_{\pm0.0181}$ | $0.5807_{\pm0.0196}$ | $0.5898_{\pm0.0111}$ |
| ProdLDA | $0.6449_{\pm0.0145}$ | $0.6293_{\pm0.0071}$ | $0.6193_{\pm0.0155}$ | $0.5858_{\pm0.0130}$ | $0.5738_{\pm0.0083}$ | $0.6272_{\pm0.0343}$ | $0.6179_{\pm0.0213}$ | $0.6157_{\pm0.0153}$ | $0.5788_{\pm0.0077}$ | $0.5681_{\pm0.0142}$ |
| ETM | $0.6653_{\pm0.0083}$ | $0.6436_{\pm0.0040}$ | $0.6337_{\pm0.0059}$ | $0.6217_{\pm0.0064}$ | $0.6115_{\pm0.0091}$ | $0.6090_{\pm0.0162}$ | $0.6047_{\pm0.0095}$ | $0.6051_{\pm0.0105}$ | $0.5916_{\pm0.0059}$ | $0.5905_{\pm0.0054}$ |
| GraphBTM | $0.5218_{\pm0.0133}$ | $0.5061_{\pm0.0158}$ | $0.5317_{\pm0.0184}$ | $0.5569_{\pm0.0298}$ | $0.4954_{\pm0.0022}$ | $0.6033_{\pm0.0060}$ | $0.5925_{\pm0.0121}$ | $0.5975_{\pm0.0061}$ | $0.6015_{\pm0.0235}$ | $0.5747_{\pm0.0303}$ |
| iDocNADE | $0.5157_{\pm0.0451}$ | $0.5008_{\pm0.0276}$ | $0.4539_{\pm0.0359}$ | $0.4220_{\pm0.0121}$ | $0.4133_{\pm0.0137}$ | $0.5407_{\pm0.0334}$ | $0.5061_{\pm0.0457}$ | $0.4563_{\pm0.0351}$ | $0.4217_{\pm0.0043}$ | $0.4061_{\pm0.0023}$ |
| iDocNADEe | - | - | $0.4563_{\pm0.0164}$ | - | $0.3775_{\pm0.0130}$ | - | - | $0.4413_{\pm0.0256}$ | - | $0.3748_{\pm0.0081}$ |
| TopicRNN | $0.5164_{\pm0.0464}$ | $0.4442_{\pm0.0417}$ | $0.3804_{\pm0.0111}$ | $0.3443_{\pm0.0206}$ | $0.3186_{\pm0.0090}$ | $0.3445_{\pm0.0272}$ | $0.3316_{\pm0.0204}$ | $0.2995_{\pm0.0186}$ | $0.2957_{\pm0.0064}$ | $0.2803_{\pm0.0076}$ |
| GNTM | $0.6868_{\pm0.0204}$ | $0.6780_{\pm0.0114}$ | $0.6664_{\pm0.0056}$ | $0.6677_{\pm0.0074}$ | $0.6583_{\pm0.0038}$ | $0.6696_{\pm0.0117}$ | $0.6612_{\pm0.0175}$ | $0.6455_{\pm0.0084}$ | $0.6298_{\pm0.0116}$ | $0.6151_{\pm0.0065}$ |

Table S6: The performance of topic diversity with top 5 words.

| | 20NG | | | | | TMN | | | | |
|---|---|---|---|---|---|---|---|---|---|---|
| | 20 | 30 | 50 | 70 | 100 | 20 | 30 | 50 | 70 | 100 |
| LDA | $0.8380_{\pm0.0117}$ | $0.8450_{\pm0.0222}$ | $0.8464_{\pm0.0246}$ | $0.8640_{\pm0.0138}$ | $0.8816_{\pm0.0129}$ | $0.9900_{\pm0.0089}$ | $0.9958_{\pm0.0056}$ | $0.9944_{\pm0.0020}$ | $0.9940_{\pm0.0019}$ | $0.8816_{\pm0.0242}$ |
| GSM | $0.7380_{\pm0.0453}$ | $0.7128_{\pm0.0439}$ | $0.6920_{\pm0.0188}$ | $0.5682_{\pm0.0185}$ | $0.3628_{\pm0.0254}$ | $0.9720_{\pm0.0117}$ | $0.9586_{\pm0.0245}$ | $0.6472_{\pm0.0416}$ | $0.2874_{\pm0.0141}$ | $0.0608_{\pm0.0143}$ |
| ProdLDA | $0.9420_{\pm0.0147}$ | $0.8810_{\pm0.0271}$ | $0.7944_{\pm0.0213}$ | $0.7534_{\pm0.0272}$ | $0.7104_{\pm0.0146}$ | $0.8600_{\pm0.0374}$ | $0.6742_{\pm0.0363}$ | $0.7016_{\pm0.0531}$ | $0.7124_{\pm0.0641}$ | $0.6956_{\pm0.0397}$ |
| ETM | $0.8520_{\pm0.0319}$ | $0.7876_{\pm0.0090}$ | $0.6880_{\pm0.0152}$ | $0.6474_{\pm0.0100}$ | $0.5788_{\pm0.0086}$ | $0.9640_{\pm0.0224}$ | $0.9158_{\pm0.0209}$ | $0.7648_{\pm0.0174}$ | $0.6888_{\pm0.0132}$ | $0.5804_{\pm0.0160}$ |
| GraphBTM | $0.3080_{\pm0.0214}$ | $0.2385_{\pm0.0770}$ | $0.2776_{\pm0.0362}$ | $0.2682_{\pm0.0768}$ | $0.4380_{\pm0.1570}$ | $0.4120_{\pm0.0954}$ | $0.4785_{\pm0.0114}$ | $0.7712_{\pm0.0265}$ | $0.8185_{\pm0.0290}$ | $0.7856_{\pm0.0194}$ |
| iDocNADE | $0.7280_{\pm0.0376}$ | $0.6662_{\pm0.0311}$ | $0.6504_{\pm0.0280}$ | $0.6380_{\pm0.0217}$ | $0.6120_{\pm0.0100}$ | $0.5360_{\pm0.0242}$ | $0.4850_{\pm0.0155}$ | $0.3936_{\pm0.0169}$ | $0.3368_{\pm0.0201}$ | $0.2800_{\pm0.0066}$ |
| iDocNADEe | - | - | $0.5776_{\pm0.0635}$ | - | $0.5628_{\pm0.0659}$ | - | - | $0.4280_{\pm0.0173}$ | - | $0.3728_{\pm0.0109}$ |
| TopicRNN | $0.9800_{\pm0.0063}$ | $0.9676_{\pm0.0108}$ | $0.9768_{\pm0.0047}$ | $0.9752_{\pm0.0052}$ | $0.9536_{\pm0.0062}$ | $0.7160_{\pm0.0595}$ | $0.7316_{\pm0.0369}$ | $0.6392_{\pm0.0266}$ | $0.6206_{\pm0.0202}$ | $0.6304_{\pm0.0141}$ |
| GNTM | $0.9540_{\pm0.0102}$ | $0.9251_{\pm0.0179}$ | $0.9016_{\pm0.0190}$ | $0.8037_{\pm0.0324}$ | $0.6528_{\pm0.0217}$ | $1.0000_{\pm0.0000}$ | $1.0000_{\pm0.0000}$ | $0.9984_{\pm0.0032}$ | $0.9940_{\pm0.0019}$ | $0.9368_{\pm0.0372}$ |
| | BNC | | | | | Reuters | | | | |
| | 20 | 30 | 50 | 70 | 100 | 20 | 30 | 50 | 70 | 100 |
| LDA | $0.8080_{\pm0.0421}$ | $0.8234_{\pm0.0483}$ | $0.8368_{\pm0.0250}$ | $0.8452_{\pm0.0069}$ | $0.8516_{\pm0.0062}$ | $0.6500_{\pm0.0141}$ | $0.6690_{\pm0.0311}$ | $0.6776_{\pm0.0130}$ | $0.6882_{\pm0.0204}$ | $0.6964_{\pm0.0102}$ |
| GSM | $0.8200_{\pm0.0253}$ | $0.7744_{\pm0.0184}$ | $0.7176_{\pm0.0161}$ | $0.6018_{\pm0.0440}$ | $0.4140_{\pm0.0133}$ | $0.6060_{\pm0.0488}$ | $0.5717_{\pm0.0225}$ | $0.5240_{\pm0.0378}$ | $0.3916_{\pm0.0439}$ | $0.2376_{\pm0.0165}$ |
| ProdLDA | $0.9140_{\pm0.0287}$ | $0.8636_{\pm0.0125}$ | $0.7904_{\pm0.0169}$ | $0.7368_{\pm0.0206}$ | $0.6340_{\pm0.0252}$ | $0.9580_{\pm0.0117}$ | $0.9104_{\pm0.0150}$ | $0.8360_{\pm0.0143}$ | $0.7712_{\pm0.0130}$ | $0.6748_{\pm0.0118}$ |
| ETM | $0.9380_{\pm0.0133}$ | $0.8943_{\pm0.0108}$ | $0.8112_{\pm0.0117}$ | $0.7564_{\pm0.0148}$ | $0.6896_{\pm0.0246}$ | $0.8940_{\pm0.0427}$ | $0.8462_{\pm0.0168}$ | $0.7560_{\pm0.0181}$ | $0.6734_{\pm0.0122}$ | $0.5636_{\pm0.0189}$ |
| GraphBTM | $0.1700_{\pm0.0603}$ | $0.1211_{\pm0.0670}$ | $0.1776_{\pm0.0360}$ | $0.2008_{\pm0.0549}$ | $0.1536_{\pm0.0267}$ | $0.1940_{\pm0.0185}$ | $0.1665_{\pm0.0084}$ | $0.1776_{\pm0.0419}$ | $0.1997_{\pm0.0075}$ | $0.2444_{\pm0.0345}$ |
| iDocNADE | $0.5980_{\pm0.0279}$ | $0.6022_{\pm0.0274}$ | $0.6392_{\pm0.0452}$ | $0.6492_{\pm0.0629}$ | $0.6276_{\pm0.0414}$ | $0.5320_{\pm0.0337}$ | $0.5264_{\pm0.0250}$ | $0.4816_{\pm0.0130}$ | $0.4628_{\pm0.0103}$ | $0.4460_{\pm0.0132}$ |
| iDocNADEe | - | - | $0.6904_{\pm0.0255}$ | - | $0.6296_{\pm0.0051}$ | - | - | $0.5528_{\pm0.0223}$ | - | $0.4672_{\pm0.0129}$ |
| TopicRNN | $0.9880_{\pm0.0117}$ | $0.9810_{\pm0.0077}$ | $0.9832_{\pm0.0064}$ | $0.9746_{\pm0.0040}$ | $0.9676_{\pm0.0066}$ | $0.9640_{\pm0.0150}$ | $0.9612_{\pm0.0065}$ | $0.9624_{\pm0.0149}$ | $0.9606_{\pm0.0103}$ | $0.9508_{\pm0.0116}$ |
| GNTM | $0.9820_{\pm0.0117}$ | $0.9598_{\pm0.0164}$ | $0.9024_{\pm0.0203}$ | $0.7990_{\pm0.0762}$ | $0.7056_{\pm0.0589}$ | $0.9060_{\pm0.0150}$ | $0.8582_{\pm0.0265}$ | $0.8024_{\pm0.0235}$ | $0.7874_{\pm0.0167}$ | $0.7440_{\pm0.0102}$ |

Table S7: The performance of topic coherence based on NPMI metric with top 10 words.

| | 20NG | | | | | TMN | | | | |
|---|---|---|---|---|---|---|---|---|---|---|
| | 20 | 30 | 50 | 70 | 100 | 20 | 30 | 50 | 70 | 100 |
| LDA | $0.0552_{\pm0.0133}$ | $0.0425_{\pm0.0172}$ | $0.0209_{\pm0.0098}$ | $0.0069_{\pm0.0106}$ | $-0.0156_{\pm0.0047}$ | $-0.1222_{\pm0.0100}$ | $-0.1655_{\pm0.0096}$ | $-0.2152_{\pm0.0088}$ | $-0.2447_{\pm0.0040}$ | $-0.1714_{\pm0.0087}$ |
| GSM | $0.0902_{\pm0.0181}$ | $0.0872_{\pm0.0023}$ | $0.0767_{\pm0.0117}$ | $0.0560_{\pm0.0063}$ | $0.0543_{\pm0.0052}$ | $-0.0249_{\pm0.0048}$ | $-0.0289_{\pm0.0076}$ | $-0.0040_{\pm0.0113}$ | $-0.0335_{\pm0.0058}$ | $-0.1607_{\pm0.0093}$ |
| ProdLDA | $0.0059_{\pm0.0265}$ | $-0.0017_{\pm0.0286}$ | $-0.0296_{\pm0.0195}$ | $-0.0410_{\pm0.0190}$ | $-0.0511_{\pm0.0237}$ | $-0.1900_{\pm0.0324}$ | $-0.2019_{\pm0.0154}$ | $-0.2348_{\pm0.0176}$ | $-0.2603_{\pm0.0220}$ | $-0.2842_{\pm0.0132}$ |
| ETM | $0.0750_{\pm0.0097}$ | $0.0749_{\pm0.0088}$ | $0.0549_{\pm0.0051}$ | $0.0423_{\pm0.0043}$ | $0.0328_{\pm0.0017}$ | $-0.0391_{\pm0.0216}$ | $-0.0505_{\pm0.0122}$ | $-0.0594_{\pm0.0140}$ | $-0.0677_{\pm0.0080}$ | $-0.0829_{\pm0.0081}$ |
| GraphBTM | $0.0309_{\pm0.0017}$ | $0.0372_{\pm0.0163}$ | $0.0509_{\pm0.0019}$ | $0.0271_{\pm0.0295}$ | $-0.0271_{\pm0.0546}$ | $-0.1460_{\pm0.0113}$ | $-0.1732_{\pm0.0098}$ | $-0.2541_{\pm0.0021}$ | $-0.2883_{\pm0.0048}$ | $-0.3020_{\pm0.0041}$ |
| iDocNADE | $-0.1679_{\pm0.0179}$ | $-0.1960_{\pm0.0157}$ | $-0.2180_{\pm0.0154}$ | $-0.2208_{\pm0.0122}$ | $-0.2370_{\pm0.0147}$ | $-0.5261_{\pm0.0202}$ | $-0.5054_{\pm0.0105}$ | $-0.4744_{\pm0.0132}$ | $-0.4555_{\pm0.0091}$ | $-0.4409_{\pm0.0118}$ |
| iDocNADEe | - | - | $-0.1197_{\pm0.0592}$ | - | $-0.1680_{\pm0.0583}$ | - | - | $-0.4601_{\pm0.0145}$ | - | $-0.4492_{\pm0.0055}$ |
| TopicRNN | $-0.1959_{\pm0.0061}$ | $-0.2398_{\pm0.0217}$ | $-0.2950_{\pm0.0052}$ | $-0.3154_{\pm0.0041}$ | $-0.3332_{\pm0.0089}$ | $-0.4500_{\pm0.0197}$ | $-0.4289_{\pm0.0093}$ | $-0.4327_{\pm0.0137}$ | $-0.4334_{\pm0.0076}$ | $-0.4311_{\pm0.0037}$ |
| GNTM | $0.1235_{\pm0.0132}$ | $0.1074_{\pm0.0111}$ | $0.0807_{\pm0.0105}$ | $0.0710_{\pm0.0022}$ | $0.0587_{\pm0.0066}$ | $-0.0327_{\pm0.0137}$ | $-0.0273_{\pm0.0101}$ | $-0.0235_{\pm0.0041}$ | $-0.0253_{\pm0.0018}$ | $-0.0245_{\pm0.0179}$ |
| | BNC | | | | | Reuters | | | | |
| | 20 | 30 | 50 | 70 | 100 | 20 | 30 | 50 | 70 | 100 |
| LDA | $0.0749_{\pm0.0052}$ | $0.0712_{\pm0.0131}$ | $0.0820_{\pm0.0130}$ | $0.0743_{\pm0.0142}$ | $0.0669_{\pm0.0197}$ | $0.0753_{\pm0.0085}$ | $0.0651_{\pm0.0157}$ | $0.0474_{\pm0.0106}$ | $0.0297_{\pm0.0134}$ | $0.0019_{\pm0.0092}$ |
| GSM | $0.0798_{\pm0.0023}$ | $0.0895_{\pm0.0022}$ | $0.0893_{\pm0.0029}$ | $0.0792_{\pm0.0084}$ | $0.0578_{\pm0.0015}$ | $0.1018_{\pm0.0072}$ | $0.0965_{\pm0.0055}$ | $0.0640_{\pm0.0096}$ | $0.0269_{\pm0.0119}$ | $0.0220_{\pm0.0032}$ |
| ProdLDA | $0.0694_{\pm0.0129}$ | $0.0628_{\pm0.0137}$ | $0.0417_{\pm0.0043}$ | $0.0218_{\pm0.0101}$ | $0.0141_{\pm0.0103}$ | $0.0232_{\pm0.0146}$ | $0.0329_{\pm0.0106}$ | $0.0289_{\pm0.0050}$ | $0.0099_{\pm0.0051}$ | $0.0224_{\pm0.0106}$ |
| ETM | $0.0883_{\pm0.0066}$ | $0.0794_{\pm0.0062}$ | $0.0720_{\pm0.0027}$ | $0.0680_{\pm0.0046}$ | $0.0634_{\pm0.0034}$ | $0.0609_{\pm0.0079}$ | $0.0572_{\pm0.0057}$ | $0.0476_{\pm0.0098}$ | $0.0369_{\pm0.0031}$ | $0.0317_{\pm0.0046}$ |
| GraphBTM | $0.0195_{\pm0.0051}$ | $0.0076_{\pm0.0117}$ | $0.0231_{\pm0.0106}$ | $0.0364_{\pm0.0165}$ | $0.0024_{\pm0.0008}$ | $0.0427_{\pm0.0062}$ | $0.0441_{\pm0.0085}$ | $0.0467_{\pm0.0059}$ | $0.0402_{\pm0.0103}$ | $0.0175_{\pm0.0226}$ |
| iDocNADE | $-0.0426_{\pm0.0288}$ | $-0.0660_{\pm0.0309}$ | $-0.1228_{\pm0.0214}$ | $-0.1637_{\pm0.0246}$ | $-0.1792_{\pm0.0046}$ | $-0.1433_{\pm0.0197}$ | $-0.2049_{\pm0.0245}$ | $-0.2464_{\pm0.0162}$ | $-0.2623_{\pm0.0149}$ | $-0.2832_{\pm0.0067}$ |
| iDocNADEe | - | - | $-0.1002_{\pm0.0066}$ | - | $-0.1673_{\pm0.0069}$ | - | - | $-0.2160_{\pm0.0064}$ | - | $-0.2773_{\pm0.0049}$ |
| TopicRNN | $-0.1022_{\pm0.0159}$ | $-0.1668_{\pm0.0240}$ | $-0.2077_{\pm0.0141}$ | $-0.2440_{\pm0.0134}$ | $-0.2770_{\pm0.0103}$ | $-0.2565_{\pm0.0278}$ | $-0.2971_{\pm0.0141}$ | $-0.3311_{\pm0.0078}$ | $-0.3475_{\pm0.0023}$ | $-0.3570_{\pm0.0054}$ |
| GNTM | $0.0967_{\pm0.0054}$ | $0.0971_{\pm0.0048}$ | $0.0884_{\pm0.0021}$ | $0.0893_{\pm0.0042}$ | $0.0836_{\pm0.0033}$ | $0.0965_{\pm0.0086}$ | $0.0945_{\pm0.0080}$ | $0.0835_{\pm0.0050}$ | $0.0543_{\pm0.0081}$ | $0.0417_{\pm0.0051}$ |

Table S8: The performance of topic coherence based on $C_v$ metric with top 10 words.

| | 20NG | | | | | TMN | | | | |
|---|---|---|---|---|---|---|---|---|---|---|
| | 20 | 30 | 50 | 70 | 100 | 20 | 30 | 50 | 70 | 100 |
| LDA | $0.6182_{\pm0.0153}$ | $0.5953_{\pm0.0313}$ | $0.5807_{\pm0.0136}$ | $0.5521_{\pm0.0110}$ | $0.5259_{\pm0.0041}$ | $0.2907_{\pm0.0072}$ | $0.2969_{\pm0.0078}$ | $0.3064_{\pm0.0076}$ | $0.3260_{\pm0.0056}$ | $0.3342_{\pm0.0060}$ |
| GSM | $0.6393_{\pm0.0256}$ | $0.6189_{\pm0.0151}$ | $0.5991_{\pm0.0183}$ | $0.5656_{\pm0.0066}$ | $0.5652_{\pm0.0078}$ | $0.4332_{\pm0.0166}$ | $0.4203_{\pm0.0092}$ | $0.4002_{\pm0.0103}$ | $0.3024_{\pm0.0121}$ | $0.1955_{\pm0.0117}$ |
| ProdLDA | $0.5799_{\pm0.0372}$ | $0.5605_{\pm0.0425}$ | $0.5121_{\pm0.0218}$ | $0.5131_{\pm0.0254}$ | $0.4909_{\pm0.0145}$ | $0.3206_{\pm0.0167}$ | $0.3059_{\pm0.0109}$ | $0.3129_{\pm0.0132}$ | $0.3343_{\pm0.0196}$ | $0.3599_{\pm0.0171}$ |
| ETM | $0.6421_{\pm0.0156}$ | $0.6365_{\pm0.0164}$ | $0.6077_{\pm0.0072}$ | $0.5916_{\pm0.0079}$ | $0.5745_{\pm0.0026}$ | $0.4320_{\pm0.0238}$ | $0.4259_{\pm0.0194}$ | $0.4199_{\pm0.0124}$ | $0.4211_{\pm0.0124}$ | $0.4033_{\pm0.0024}$ |
| GraphBTM | $0.5268_{\pm0.0058}$ | $0.5452_{\pm0.0333}$ | $0.5797_{\pm0.0072}$ | $0.5253_{\pm0.0650}$ | $0.4480_{\pm0.0778}$ | $0.2128_{\pm0.0147}$ | $0.2570_{\pm0.0074}$ | $0.3400_{\pm0.0034}$ | $0.3639_{\pm0.0092}$ | $0.3799_{\pm0.0018}$ |
| iDocNADE | $0.4422_{\pm0.0282}$ | $0.4319_{\pm0.0161}$ | $0.4418_{\pm0.0123}$ | $0.4309_{\pm0.0060}$ | $0.4224_{\pm0.0085}$ | $0.5825_{\pm0.0190}$ | $0.5656_{\pm0.0114}$ | $0.5410_{\pm0.0109}$ | $0.5249_{\pm0.0085}$ | $0.5147_{\pm0.0099}$ |
| iDocNADEe | - | - | $0.3894_{\pm0.0143}$ | - | $0.3942_{\pm0.0106}$ | - | - | $0.5292_{\pm0.0109}$ | - | $0.5196_{\pm0.0057}$ |
| TopicRNN | $0.4029_{\pm0.0249}$ | $0.3947_{\pm0.0170}$ | $0.3779_{\pm0.0090}$ | $0.3943_{\pm0.0124}$ | $0.4089_{\pm0.0131}$ | $0.5169_{\pm0.0203}$ | $0.5066_{\pm0.0085}$ | $0.5083_{\pm0.0097}$ | $0.5091_{\pm0.0073}$ | $0.5086_{\pm0.0041}$ |
| GNTM | $0.6962_{\pm0.0126}$ | $0.6605_{\pm0.0094}$ | $0.6328_{\pm0.0108}$ | $0.6222_{\pm0.0055}$ | $0.5939_{\pm0.0091}$ | $0.3977_{\pm0.0117}$ | $0.4237_{\pm0.0155}$ | $0.4489_{\pm0.0114}$ | $0.4585_{\pm0.0071}$ | $0.4521_{\pm0.0163}$ |
| | BNC | | | | | Reuters | | | | |
| | 20 | 30 | 50 | 70 | 100 | 20 | 30 | 50 | 70 | 100 |
| LDA | $0.5705_{\pm0.0077}$ | $0.5815_{\pm0.0107}$ | $0.5969_{\pm0.0051}$ | $0.5983_{\pm0.0062}$ | $0.5938_{\pm0.0136}$ | $0.5327_{\pm0.0085}$ | $0.5273_{\pm0.0261}$ | $0.5102_{\pm0.0086}$ | $0.4951_{\pm0.0141}$ | $0.4622_{\pm0.0135}$ |
| GSM | $0.5755_{\pm0.0062}$ | $0.5912_{\pm0.0070}$ | $0.5863_{\pm0.0051}$ | $0.5573_{\pm0.0125}$ | $0.5122_{\pm0.0035}$ | $0.5868_{\pm0.0167}$ | $0.5732_{\pm0.0037}$ | $0.5252_{\pm0.0112}$ | $0.4817_{\pm0.0215}$ | $0.4678_{\pm0.0067}$ |
| ProdLDA | $0.5793_{\pm0.0220}$ | $0.5620_{\pm0.0098}$ | $0.5321_{\pm0.0129}$ | $0.5047_{\pm0.0096}$ | $0.4907_{\pm0.0098}$ | $0.5337_{\pm0.0118}$ | $0.5290_{\pm0.0217}$ | $0.5249_{\pm0.0107}$ | $0.4954_{\pm0.0121}$ | $0.4978_{\pm0.0090}$ |
| ETM | $0.6071_{\pm0.0044}$ | $0.5918_{\pm0.0082}$ | $0.5740_{\pm0.0073}$ | $0.5591_{\pm0.0078}$ | $0.5465_{\pm0.0081}$ | $0.5494_{\pm0.0100}$ | $0.5362_{\pm0.0039}$ | $0.5312_{\pm0.0085}$ | $0.5138_{\pm0.0046}$ | $0.5073_{\pm0.0081}$ |
| GraphBTM | $0.4368_{\pm0.0156}$ | $0.4008_{\pm0.0326}$ | $0.4413_{\pm0.0306}$ | $0.4793_{\pm0.0459}$ | $0.3825_{\pm0.0026}$ | $0.5017_{\pm0.0105}$ | $0.4986_{\pm0.0154}$ | $0.5032_{\pm0.0086}$ | $0.4966_{\pm0.0227}$ | $0.4619_{\pm0.0433}$ |
| iDocNADE | $0.4349_{\pm0.0201}$ | $0.4255_{\pm0.0237}$ | $0.3963_{\pm0.0239}$ | $0.3868_{\pm0.0103}$ | $0.3823_{\pm0.0148}$ | $0.5116_{\pm0.0142}$ | $0.4859_{\pm0.0158}$ | $0.4651_{\pm0.0140}$ | $0.4569_{\pm0.0073}$ | $0.4632_{\pm0.0041}$ |
| iDocNADEe | - | - | $0.3350_{\pm0.0099}$ | - | $0.3091_{\pm0.0082}$ | - | - | $0.3885_{\pm0.0108}$ | - | $0.4154_{\pm0.0082}$ |
| TopicRNN | $0.4139_{\pm0.0318}$ | $0.3598_{\pm0.0261}$ | $0.3176_{\pm0.0065}$ | $0.3015_{\pm0.0041}$ | $0.2943_{\pm0.0063}$ | $0.3776_{\pm0.0155}$ | $0.3900_{\pm0.0098}$ | $0.4120_{\pm0.0053}$ | $0.4345_{\pm0.0083}$ | $0.4524_{\pm0.0108}$ |
| GNTM | $0.6218_{\pm0.0137}$ | $0.6253_{\pm0.0104}$ | $0.6067_{\pm0.0037}$ | $0.6024_{\pm0.0070}$ | $0.5930_{\pm0.0032}$ | $0.5917_{\pm0.0048}$ | $0.5781_{\pm0.0150}$ | $0.5623_{\pm0.0116}$ | $0.5276_{\pm0.0133}$ | $0.5154_{\pm0.0086}$ |

Table S9: The performance of topic diversity with top 10 words.

| | 20NG | | | | | TMN | | | | |
|---|---|---|---|---|---|---|---|---|---|---|
| | 20 | 30 | 50 | 70 | 100 | 20 | 30 | 50 | 70 | 100 |
| LDA | $0.7900_{\pm0.0130}$ | $0.8158_{\pm0.0222}$ | $0.8176_{\pm0.0125}$ | $0.8256_{\pm0.0087}$ | $0.8274_{\pm0.0109}$ | $0.9840_{\pm0.0086}$ | $0.9964_{\pm0.0031}$ | $0.9936_{\pm0.0020}$ | $0.9912_{\pm0.0035}$ | $0.8066_{\pm0.0156}$ |
| GSM | $0.7040_{\pm0.0271}$ | $0.6924_{\pm0.0349}$ | $0.6368_{\pm0.0131}$ | $0.5156_{\pm0.0234}$ | $0.3246_{\pm0.0188}$ | $0.9530_{\pm0.0125}$ | $0.9176_{\pm0.0210}$ | $0.6100_{\pm0.0363}$ | $0.2632_{\pm0.0155}$ | $0.0600_{\pm0.0091}$ |
| ProdLDA | $0.9300_{\pm0.0077}$ | $0.8582_{\pm0.0152}$ | $0.7628_{\pm0.0139}$ | $0.7038_{\pm0.0162}$ | $0.6356_{\pm0.0064}$ | $0.7890_{\pm0.0278}$ | $0.6302_{\pm0.0339}$ | $0.5992_{\pm0.0647}$ | $0.6108_{\pm0.0634}$ | $0.6056_{\pm0.0415}$ |
| ETM | $0.8550_{\pm0.0327}$ | $0.7796_{\pm0.0119}$ | $0.6636_{\pm0.0054}$ | $0.6092_{\pm0.0019}$ | $0.5412_{\pm0.0084}$ | $0.9540_{\pm0.0222}$ | $0.8976_{\pm0.0120}$ | $0.7380_{\pm0.0145}$ | $0.6470_{\pm0.0091}$ | $0.5502_{\pm0.0138}$ |
| GraphBTM | $0.2780_{\pm0.0121}$ | $0.2211_{\pm0.0565}$ | $0.2556_{\pm0.0211}$ | $0.2404_{\pm0.0605}$ | $0.3936_{\pm0.1320}$ | $0.4450_{\pm0.0875}$ | $0.5091_{\pm0.0216}$ | $0.6960_{\pm0.0125}$ | $0.7306_{\pm0.0171}$ | $0.6902_{\pm0.0135}$ |
| iDocNADE | $0.7160_{\pm0.0337}$ | $0.6636_{\pm0.0181}$ | $0.6428_{\pm0.0200}$ | $0.6166_{\pm0.0134}$ | $0.5892_{\pm0.0121}$ | $0.5850_{\pm0.0235}$ | $0.4990_{\pm0.0242}$ | $0.3940_{\pm0.0109}$ | $0.3246_{\pm0.0081}$ | $0.2714_{\pm0.0059}$ |
| iDocNADEe | - | - | $0.5448_{\pm0.0416}$ | - | $0.5338_{\pm0.0538}$ | - | - | $0.3904_{\pm0.0096}$ | - | $0.3238_{\pm0.0115}$ |
| TopicRNN | $0.9590_{\pm0.0097}$ | $0.9482_{\pm0.0141}$ | $0.9504_{\pm0.0076}$ | $0.9422_{\pm0.0065}$ | $0.9262_{\pm0.0105}$ | $0.7190_{\pm0.0493}$ | $0.6944_{\pm0.0390}$ | $0.6080_{\pm0.0258}$ | $0.5834_{\pm0.0188}$ | $0.5812_{\pm0.0151}$ |
| GNTM | $0.9110_{\pm0.0086}$ | $0.8759_{\pm0.0145}$ | $0.8352_{\pm0.0115}$ | $0.7231_{\pm0.0319}$ | $0.5530_{\pm0.0139}$ | $0.9990_{\pm0.0020}$ | $0.9970_{\pm0.0027}$ | $0.9932_{\pm0.0048}$ | $0.9870_{\pm0.0045}$ | $0.9014_{\pm0.0541}$ |

| | BNC | | | | | Reuters | | | | |
|---|---|---|---|---|---|---|---|---|---|---|
| | 20 | 30 | 50 | 70 | 100 | 20 | 30 | 50 | 70 | 100 |
| LDA | $0.7770_{\pm0.0466}$ | $0.7776_{\pm0.0364}$ | $0.7692_{\pm0.0213}$ | $0.7840_{\pm0.0140}$ | $0.7890_{\pm0.0105}$ | $0.6640_{\pm0.0206}$ | $0.6742_{\pm0.0207}$ | $0.6744_{\pm0.0143}$ | $0.6730_{\pm0.0164}$ | $0.6752_{\pm0.0042}$ |
| GSM | $0.7630_{\pm0.0144}$ | $0.7082_{\pm0.0120}$ | $0.6280_{\pm0.0150}$ | $0.5348_{\pm0.0386}$ | $0.3586_{\pm0.0119}$ | $0.6020_{\pm0.0186}$ | $0.5530_{\pm0.0163}$ | $0.5004_{\pm0.0207}$ | $0.3646_{\pm0.0330}$ | $0.2274_{\pm0.0201}$ |
| ProdLDA | $0.8890_{\pm0.0285}$ | $0.8056_{\pm0.0091}$ | $0.7280_{\pm0.0125}$ | $0.6544_{\pm0.0156}$ | $0.5544_{\pm0.0126}$ | $0.9180_{\pm0.0223}$ | $0.8690_{\pm0.0224}$ | $0.7764_{\pm0.0130}$ | $0.6974_{\pm0.0080}$ | $0.5898_{\pm0.0105}$ |
| ETM | $0.9180_{\pm0.0103}$ | $0.8671_{\pm0.0141}$ | $0.7824_{\pm0.0098}$ | $0.7151_{\pm0.0090}$ | $0.6426_{\pm0.0143}$ | $0.8990_{\pm0.0150}$ | $0.8166_{\pm0.0055}$ | $0.7224_{\pm0.0085}$ | $0.6274_{\pm0.0080}$ | $0.5122_{\pm0.0130}$ |
| GraphBTM | $0.1810_{\pm0.0466}$ | $0.1383_{\pm0.0664}$ | $0.1760_{\pm0.0270}$ | $0.1804_{\pm0.0415}$ | $0.1468_{\pm0.0220}$ | $0.2020_{\pm0.0227}$ | $0.1591_{\pm0.0063}$ | $0.1700_{\pm0.0300}$ | $0.1846_{\pm0.0134}$ | $0.2356_{\pm0.0193}$ |
| iDocNADE | $0.6030_{\pm0.0474}$ | $0.5914_{\pm0.0316}$ | $0.6076_{\pm0.0336}$ | $0.6172_{\pm0.0498}$ | $0.5966_{\pm0.0393}$ | $0.5140_{\pm0.0224}$ | $0.4998_{\pm0.0155}$ | $0.4372_{\pm0.0180}$ | $0.4208_{\pm0.0107}$ | $0.3996_{\pm0.0056}$ |
| iDocNADEe | - | - | $0.6368_{\pm0.0259}$ | - | $0.6110_{\pm0.0149}$ | - | - | $0.4832_{\pm0.0077}$ | - | $0.4252_{\pm0.0035}$ |
| TopicRNN | $0.9710_{\pm0.0097}$ | $0.9602_{\pm0.0098}$ | $0.9592_{\pm0.0095}$ | $0.9430_{\pm0.0065}$ | $0.9414_{\pm0.0071}$ | $0.9560_{\pm0.0146}$ | $0.9508_{\pm0.0075}$ | $0.9360_{\pm0.0157}$ | $0.9256_{\pm0.0098}$ | $0.9092_{\pm0.0047}$ |
| GNTM | $0.9230_{\pm0.0298}$ | $0.9294_{\pm0.0142}$ | $0.8552_{\pm0.0220}$ | $0.7290_{\pm0.0752}$ | $0.6432_{\pm0.0575}$ | $0.8470_{\pm0.0163}$ | $0.7862_{\pm0.0225}$ | $0.7060_{\pm0.0178}$ | $0.6716_{\pm0.0116}$ | $0.6236_{\pm0.0112}$ |

Table S10: The performance of topic coherence based on NPMI metric with top 15 words.

| | 20NG | | | | | TMN | | | | |
|---|---|---|---|---|---|---|---|---|---|---|
| | 20 | 30 | 50 | 70 | 100 | 20 | 30 | 50 | 70 | 100 |
| LDA | $0.0357_{\pm0.0134}$ | $0.0241_{\pm0.0127}$ | $-0.0065_{\pm0.0091}$ | $-0.0195_{\pm0.0086}$ | $-0.0471_{\pm0.0052}$ | $-0.1700_{\pm0.0087}$ | $-0.2086_{\pm0.0040}$ | $-0.2637_{\pm0.0080}$ | $-0.2908_{\pm0.0034}$ | $-0.2186_{\pm0.0066}$ |
| GSM | $0.0724_{\pm0.0128}$ | $0.0634_{\pm0.0084}$ | $0.0496_{\pm0.0105}$ | $0.0312_{\pm0.0046}$ | $0.0376_{\pm0.0050}$ | $-0.0560_{\pm0.0123}$ | $-0.0667_{\pm0.0091}$ | $-0.0358_{\pm0.0106}$ | $-0.0442_{\pm0.0056}$ | $-0.1585_{\pm0.0095}$ |
| ProdLDA | $-0.0301_{\pm0.0151}$ | $-0.0367_{\pm0.0281}$ | $-0.0650_{\pm0.0128}$ | $-0.0754_{\pm0.0174}$ | $-0.0818_{\pm0.0183}$ | $-0.2062_{\pm0.0180}$ | $-0.2153_{\pm0.0124}$ | $-0.2488_{\pm0.0195}$ | $-0.2712_{\pm0.0224}$ | $-0.2973_{\pm0.0146}$ |
| ETM | $0.0611_{\pm0.0083}$ | $0.0648_{\pm0.0082}$ | $0.0425_{\pm0.0051}$ | $0.0297_{\pm0.0041}$ | $0.0173_{\pm0.0022}$ | $-0.0717_{\pm0.0174}$ | $-0.0816_{\pm0.0064}$ | $-0.0873_{\pm0.0076}$ | $-0.1002_{\pm0.0104}$ | $-0.1140_{\pm0.0065}$ |
| GraphBTM | $0.0273_{\pm0.0027}$ | $0.0287_{\pm0.0134}$ | $0.0403_{\pm0.0034}$ | $0.0196_{\pm0.0253}$ | $-0.0384_{\pm0.0523}$ | $-0.1573_{\pm0.0148}$ | $-0.1863_{\pm0.0091}$ | $-0.2663_{\pm0.0019}$ | $-0.2997_{\pm0.0034}$ | $-0.3135_{\pm0.0037}$ |
| iDocNADE | $-0.2158_{\pm0.0172}$ | $-0.2466_{\pm0.0114}$ | $-0.2605_{\pm0.0097}$ | $-0.2677_{\pm0.0084}$ | $-0.2795_{\pm0.0079}$ | $-0.5166_{\pm0.0077}$ | $-0.4966_{\pm0.0054}$ | $-0.4702_{\pm0.0047}$ | $-0.4545_{\pm0.0094}$ | $-0.4378_{\pm0.0089}$ |
| iDocNADEe | - | - | $-0.1779_{\pm0.0438}$ | - | $-0.2180_{\pm0.0477}$ | - | - | $-0.4561_{\pm0.0109}$ | - | $-0.4500_{\pm0.0009}$ |
| TopicRNN | $-0.2212_{\pm0.0051}$ | $-0.2638_{\pm0.0180}$ | $-0.3037_{\pm0.0057}$ | $-0.3264_{\pm0.0030}$ | $-0.3384_{\pm0.0069}$ | $-0.4532_{\pm0.0122}$ | $-0.4377_{\pm0.0089}$ | $-0.4421_{\pm0.0079}$ | $-0.4397_{\pm0.0026}$ | $-0.4401_{\pm0.0035}$ |
| GNTM | $0.1058_{\pm0.0096}$ | $0.0855_{\pm0.0082}$ | $0.0542_{\pm0.0070}$ | $0.0454_{\pm0.0035}$ | $0.0342_{\pm0.0039}$ | $-0.0539_{\pm0.0113}$ | $-0.0564_{\pm0.0078}$ | $-0.0700_{\pm0.0039}$ | $-0.0791_{\pm0.0056}$ | $-0.0747_{\pm0.0203}$ |

| | BNC | | | | | Reuters | | | | |
|---|---|---|---|---|---|---|---|---|---|---|
| | 20 | 30 | 50 | 70 | 100 | 20 | 30 | 50 | 70 | 100 |
| LDA | $0.0652_{\pm0.0045}$ | $0.0632_{\pm0.0111}$ | $0.0674_{\pm0.0115}$ | $0.0586_{\pm0.0121}$ | $0.0479_{\pm0.0195}$ | $0.0605_{\pm0.0058}$ | $0.0396_{\pm0.0158}$ | $0.0180_{\pm0.0102}$ | $-0.0046_{\pm0.0125}$ | $-0.0315_{\pm0.0062}$ |
| GSM | $0.0671_{\pm0.0025}$ | $0.0757_{\pm0.0018}$ | $0.0762_{\pm0.0034}$ | $0.0675_{\pm0.0052}$ | $0.0503_{\pm0.0014}$ | $0.0852_{\pm0.0078}$ | $0.0711_{\pm0.0070}$ | $0.0409_{\pm0.0091}$ | $0.0115_{\pm0.0126}$ | $0.0081_{\pm0.0035}$ |
| ProdLDA | $0.0552_{\pm0.0129}$ | $0.0490_{\pm0.0115}$ | $0.0244_{\pm0.0042}$ | $0.0080_{\pm0.0085}$ | $0.0029_{\pm0.0102}$ | $-0.0124_{\pm0.0083}$ | $-0.0039_{\pm0.0047}$ | $-0.0041_{\pm0.0092}$ | $-0.0236_{\pm0.0082}$ | $-0.0183_{\pm0.0094}$ |
| ETM | $0.0776_{\pm0.0068}$ | $0.0700_{\pm0.0056}$ | $0.0648_{\pm0.0032}$ | $0.0600_{\pm0.0034}$ | $0.0546_{\pm0.0029}$ | $0.0332_{\pm0.0110}$ | $0.0287_{\pm0.0048}$ | $0.0155_{\pm0.0095}$ | $0.0060_{\pm0.0037}$ | $0.0010_{\pm0.0038}$ |
| GraphBTM | $0.0177_{\pm0.0030}$ | $0.0067_{\pm0.0095}$ | $0.0219_{\pm0.0115}$ | $0.0336_{\pm0.0166}$ | $-0.0012_{\pm0.0003}$ | $0.0186_{\pm0.0075}$ | $0.0235_{\pm0.0083}$ | $0.0283_{\pm0.0071}$ | $0.0210_{\pm0.0086}$ | $-0.0039_{\pm0.0241}$ |
| iDocNADE | $-0.0697_{\pm0.0308}$ | $-0.0917_{\pm0.0294}$ | $-0.1470_{\pm0.0196}$ | $-0.1895_{\pm0.0176}$ | $-0.1999_{\pm0.0077}$ | $-0.2190_{\pm0.0149}$ | $-0.2576_{\pm0.0176}$ | $-0.2843_{\pm0.0076}$ | $-0.3039_{\pm0.0101}$ | $-0.3196_{\pm0.0056}$ |
| iDocNADEe | - | - | $-0.1199_{\pm0.0069}$ | - | $-0.1773_{\pm0.0051}$ | - | - | $-0.2520_{\pm0.0064}$ | - | $-0.3019_{\pm0.0039}$ |
| TopicRNN | $-0.1291_{\pm0.0189}$ | $-0.1867_{\pm0.0247}$ | $-0.2243_{\pm0.0116}$ | $-0.2557_{\pm0.0113}$ | $-0.2832_{\pm0.0089}$ | $-0.2827_{\pm0.0104}$ | $-0.3108_{\pm0.0090}$ | $-0.3409_{\pm0.0057}$ | $-0.3543_{\pm0.0031}$ | $-0.3616_{\pm0.0031}$ |
| GNTM | $0.0832_{\pm0.0041}$ | $0.0854_{\pm0.0029}$ | $0.0760_{\pm0.0033}$ | $0.0766_{\pm0.0037}$ | $0.0705_{\pm0.0032}$ | $0.0743_{\pm0.0073}$ | $0.0610_{\pm0.0097}$ | $0.0522_{\pm0.0069}$ | $0.0205_{\pm0.0056}$ | $0.0121_{\pm0.0082}$ |

Table S11: The performance of topic coherence based on $C_v$ metric with top 15 words.

| | 20NG | | | | | TMN | | | | |
|---|---|---|---|---|---|---|---|---|---|---|
| | 20 | 30 | 50 | 70 | 100 | 20 | 30 | 50 | 70 | 100 |
| LDA | $0.6047_{\pm0.0148}$ | $0.5797_{\pm0.0283}$ | $0.5521_{\pm0.0175}$ | $0.5287_{\pm0.0095}$ | $0.4992_{\pm0.0051}$ | $0.3070_{\pm0.0079}$ | $0.3403_{\pm0.0101}$ | $0.3990_{\pm0.0083}$ | $0.4302_{\pm0.0050}$ | $0.3579_{\pm0.0099}$ |
| GSM | $0.6184_{\pm0.0216}$ | $0.5925_{\pm0.0246}$ | $0.5672_{\pm0.0201}$ | $0.5346_{\pm0.0067}$ | $0.5388_{\pm0.0099}$ | $0.4035_{\pm0.0188}$ | $0.3941_{\pm0.0151}$ | $0.3450_{\pm0.0115}$ | $0.2613_{\pm0.0104}$ | $0.2488_{\pm0.0165}$ |
| ProdLDA | $0.5613_{\pm0.0176}$ | $0.5328_{\pm0.0407}$ | $0.4870_{\pm0.0114}$ | $0.4902_{\pm0.0215}$ | $0.4680_{\pm0.0129}$ | $0.3554_{\pm0.0175}$ | $0.3517_{\pm0.0100}$ | $0.3858_{\pm0.0237}$ | $0.4124_{\pm0.0297}$ | $0.4472_{\pm0.0220}$ |
| ETM | $0.6299_{\pm0.0164}$ | $0.6287_{\pm0.0148}$ | $0.5958_{\pm0.0074}$ | $0.5743_{\pm0.0086}$ | $0.5546_{\pm0.0043}$ | $0.4175_{\pm0.0089}$ | $0.4075_{\pm0.0091}$ | $0.4011_{\pm0.0101}$ | $0.3995_{\pm0.0100}$ | $0.4017_{\pm0.0043}$ |
| GraphBTM | $0.5048_{\pm0.0115}$ | $0.5135_{\pm0.0401}$ | $0.5617_{\pm0.0050}$ | $0.5038_{\pm0.0663}$ | $0.4171_{\pm0.0803}$ | $0.2642_{\pm0.0293}$ | $0.3124_{\pm0.0165}$ | $0.4162_{\pm0.0015}$ | $0.4560_{\pm0.0060}$ | $0.4698_{\pm0.0062}$ |
| iDocNADE | $0.4555_{\pm0.0181}$ | $0.4756_{\pm0.0130}$ | $0.4824_{\pm0.0121}$ | $0.4807_{\pm0.0068}$ | $0.4790_{\pm0.0052}$ | $0.6703_{\pm0.0073}$ | $0.6547_{\pm0.0059}$ | $0.6298_{\pm0.0094}$ | $0.6152_{\pm0.0097}$ | $0.6010_{\pm0.0082}$ |
| iDocNADEe | - | - | $0.3753_{\pm0.0271}$ | - | $0.4119_{\pm0.0385}$ | - | - | $0.6154_{\pm0.0115}$ | - | $0.6140_{\pm0.0019}$ |
| TopicRNN | $0.4283_{\pm0.0168}$ | $0.4342_{\pm0.0126}$ | $0.4426_{\pm0.0093}$ | $0.4749_{\pm0.0066}$ | $0.4925_{\pm0.0129}$ | $0.6121_{\pm0.0147}$ | $0.6032_{\pm0.0092}$ | $0.6096_{\pm0.0088}$ | $0.6108_{\pm0.0032}$ | $0.6150_{\pm0.0052}$ |
| GNTM | $0.6786_{\pm0.0121}$ | $0.6411_{\pm0.0091}$ | $0.6028_{\pm0.0073}$ | $0.5905_{\pm0.0039}$ | $0.5594_{\pm0.0108}$ | $0.3745_{\pm0.0109}$ | $0.3915_{\pm0.0163}$ | $0.4082_{\pm0.0077}$ | $0.4105_{\pm0.0111}$ | $0.4054_{\pm0.0042}$ |

| | BNC | | | | | Reuters | | | | |
|---|---|---|---|---|---|---|---|---|---|---|
| | 20 | 30 | 50 | 70 | 100 | 20 | 30 | 50 | 70 | 100 |
| LDA | $0.5458_{\pm0.0060}$ | $0.5629_{\pm0.0101}$ | $0.5739_{\pm0.0039}$ | $0.5717_{\pm0.0075}$ | $0.5714_{\pm0.0135}$ | $0.5183_{\pm0.0095}$ | $0.4991_{\pm0.0284}$ | $0.4819_{\pm0.0112}$ | $0.4605_{\pm0.0106}$ | $0.4407_{\pm0.0091}$ |
| GSM | $0.5445_{\pm0.0077}$ | $0.5650_{\pm0.0068}$ | $0.5607_{\pm0.0090}$ | $0.5304_{\pm0.0106}$ | $0.4814_{\pm0.0039}$ | $0.5583_{\pm0.0162}$ | $0.5310_{\pm0.0101}$ | $0.4840_{\pm0.0093}$ | $0.4382_{\pm0.0252}$ | $0.4145_{\pm0.0082}$ |
| ProdLDA | $0.5510_{\pm0.0332}$ | $0.5331_{\pm0.0121}$ | $0.5007_{\pm0.0168}$ | $0.4767_{\pm0.0112}$ | $0.4651_{\pm0.0086}$ | $0.4982_{\pm0.0061}$ | $0.4927_{\pm0.0127}$ | $0.4893_{\pm0.0131}$ | $0.4653_{\pm0.0150}$ | $0.4613_{\pm0.0088}$ |
| ETM | $0.5839_{\pm0.0078}$ | $0.5745_{\pm0.0092}$ | $0.5592_{\pm0.0077}$ | $0.5403_{\pm0.0076}$ | $0.5228_{\pm0.0068}$ | $0.5250_{\pm0.0129}$ | $0.5008_{\pm0.0084}$ | $0.4931_{\pm0.0116}$ | $0.4780_{\pm0.0052}$ | $0.4741_{\pm0.0048}$ |
| GraphBTM | $0.4061_{\pm0.0113}$ | $0.3572_{\pm0.0378}$ | $0.4113_{\pm0.0424}$ | $0.4533_{\pm0.0580}$ | $0.3305_{\pm0.0021}$ | $0.4445_{\pm0.0124}$ | $0.4473_{\pm0.0163}$ | $0.4596_{\pm0.0133}$ | $0.4502_{\pm0.0197}$ | $0.4111_{\pm0.0498}$ |
| iDocNADE | $0.3986_{\pm0.0208}$ | $0.3962_{\pm0.0213}$ | $0.3868_{\pm0.0166}$ | $0.3936_{\pm0.0111}$ | $0.3942_{\pm0.0118}$ | $0.4895_{\pm0.0045}$ | $0.4965_{\pm0.0169}$ | $0.4898_{\pm0.0179}$ | $0.5023_{\pm0.0057}$ | $0.5126_{\pm0.0031}$ |
| iDocNADEe | - | - | $0.2964_{\pm0.0099}$ | - | $0.3068_{\pm0.0040}$ | - | - | $0.4121_{\pm0.0151}$ | - | $0.4672_{\pm0.0059}$ |
| TopicRNN | $0.3809_{\pm0.0232}$ | $0.3433_{\pm0.0208}$ | $0.3250_{\pm0.0065}$ | $0.3260_{\pm0.0052}$ | $0.3300_{\pm0.0030}$ | $0.4354_{\pm0.0096}$ | $0.4696_{\pm0.0093}$ | $0.5038_{\pm0.0060}$ | $0.5296_{\pm0.0057}$ | $0.5514_{\pm0.0104}$ |
| GNTM | $0.6015_{\pm0.0120}$ | $0.6041_{\pm0.0069}$ | $0.5856_{\pm0.0069}$ | $0.5807_{\pm0.0079}$ | $0.5664_{\pm0.0040}$ | $0.5526_{\pm0.0046}$ | $0.5226_{\pm0.0185}$ | $0.5127_{\pm0.0136}$ | $0.4825_{\pm0.0101}$ | $0.4755_{\pm0.0098}$ |

Table S12: The performance of topic diversity with top 15 words.

| | 20NG | | | | | TMN | | | | |
|---|---|---|---|---|---|---|---|---|---|---|
| | 20 | 30 | 50 | 70 | 100 | 20 | 30 | 50 | 70 | 100 |
| LDA | $0.7770_{\pm0.0147}$ | $0.7950_{\pm0.0226}$ | $0.7952_{\pm0.0196}$ | $0.7974_{\pm0.0076}$ | $0.7934_{\pm0.0094}$ | $0.9808_{\pm0.0053}$ | $0.9886_{\pm0.0033}$ | $0.9890_{\pm0.0026}$ | $0.9882_{\pm0.0013}$ | $0.6820_{\pm0.0141}$ |
| GSM | $0.6890_{\pm0.0161}$ | $0.6628_{\pm0.0330}$ | $0.6070_{\pm0.0149}$ | $0.4872_{\pm0.0227}$ | $0.3028_{\pm0.0155}$ | $0.9228_{\pm0.0105}$ | $0.8828_{\pm0.0238}$ | $0.5732_{\pm0.0333}$ | $0.2452_{\pm0.0172}$ | $0.0552_{\pm0.0103}$ |
| ProdLDA | $0.9114_{\pm0.0170}$ | $0.8330_{\pm0.0137}$ | $0.7290_{\pm0.0095}$ | $0.6672_{\pm0.0178}$ | $0.5926_{\pm0.0079}$ | $0.7430_{\pm0.0338}$ | $0.5772_{\pm0.0419}$ | $0.5376_{\pm0.0674}$ | $0.5374_{\pm0.0609}$ | $0.5436_{\pm0.0487}$ |
| ETM | $0.8378_{\pm0.0331}$ | $0.7556_{\pm0.0156}$ | $0.6454_{\pm0.0097}$ | $0.5802_{\pm0.0032}$ | $0.5116_{\pm0.0087}$ | $0.9390_{\pm0.0213}$ | $0.8910_{\pm0.0124}$ | $0.7164_{\pm0.0136}$ | $0.6276_{\pm0.0124}$ | $0.5264_{\pm0.0096}$ |
| GraphBTM | $0.2745_{\pm0.0213}$ | $0.2195_{\pm0.0424}$ | $0.2419_{\pm0.0175}$ | $0.2260_{\pm0.0581}$ | $0.3626_{\pm0.1139}$ | $0.4423_{\pm0.0776}$ | $0.5228_{\pm0.0146}$ | $0.6503_{\pm0.0077}$ | $0.6672_{\pm0.0093}$ | $0.6203_{\pm0.0133}$ |
| iDocNADE | $0.7146_{\pm0.0284}$ | $0.6746_{\pm0.0210}$ | $0.6448_{\pm0.0122}$ | $0.6072_{\pm0.0199}$ | $0.5786_{\pm0.0119}$ | $0.5890_{\pm0.0133}$ | $0.5058_{\pm0.0155}$ | $0.3956_{\pm0.0143}$ | $0.3274_{\pm0.0088}$ | $0.2698_{\pm0.0077}$ |
| iDocNADEe | - | - | $0.5285_{\pm0.0440}$ | - | $0.5225_{\pm0.0497}$ | - | - | $0.3668_{\pm0.0122}$ | - | $0.3044_{\pm0.0068}$ |
| TopicRNN | $0.9478_{\pm0.0075}$ | $0.9372_{\pm0.0105}$ | $0.9288_{\pm0.0073}$ | $0.9188_{\pm0.0024}$ | $0.8924_{\pm0.0117}$ | $0.7216_{\pm0.0428}$ | $0.6844_{\pm0.0345}$ | $0.5974_{\pm0.0205}$ | $0.5742_{\pm0.0177}$ | $0.5460_{\pm0.0196}$ |
| GNTM | $0.8691_{\pm0.0153}$ | $0.8392_{\pm0.0190}$ | $0.7842_{\pm0.0193}$ | $0.6622_{\pm0.0319}$ | $0.5019_{\pm0.0133}$ | $0.9992_{\pm0.0016}$ | $0.9942_{\pm0.0010}$ | $0.9882_{\pm0.0077}$ | $0.9710_{\pm0.0099}$ | $0.8648_{\pm0.0649}$ |

| | BNC | | | | | Reuters | | | | |
|---|---|---|---|---|---|---|---|---|---|---|
| | 20 | 30 | 50 | 70 | 100 | 20 | 30 | 50 | 70 | 100 |
| LDA | $0.7470_{\pm0.0341}$ | $0.7422_{\pm0.0319}$ | $0.7312_{\pm0.0302}$ | $0.7384_{\pm0.0147}$ | $0.7408_{\pm0.0113}$ | $0.6824_{\pm0.0162}$ | $0.6778_{\pm0.0161}$ | $0.6562_{\pm0.0128}$ | $0.6510_{\pm0.0155}$ | $0.6518_{\pm0.0060}$ |
| GSM | $0.7284_{\pm0.0072}$ | $0.6570_{\pm0.0112}$ | $0.5838_{\pm0.0115}$ | $0.4938_{\pm0.0313}$ | $0.3306_{\pm0.0101}$ | $0.5958_{\pm0.0192}$ | $0.5415_{\pm0.0089}$ | $0.4775_{\pm0.0193}$ | $0.3439_{\pm0.0296}$ | $0.2172_{\pm0.0169}$ |
| ProdLDA | $0.8590_{\pm0.0304}$ | $0.7698_{\pm0.0171}$ | $0.6794_{\pm0.0180}$ | $0.6070_{\pm0.0112}$ | $0.5010_{\pm0.0115}$ | $0.9016_{\pm0.0140}$ | $0.8482_{\pm0.0118}$ | $0.7314_{\pm0.0135}$ | $0.6448_{\pm0.0078}$ | $0.5364_{\pm0.0087}$ |
| ETM | $0.8785_{\pm0.0062}$ | $0.8312_{\pm0.0080}$ | $0.7549_{\pm0.0093}$ | $0.6834_{\pm0.0082}$ | $0.6091_{\pm0.0132}$ | $0.8758_{\pm0.0128}$ | $0.7930_{\pm0.0070}$ | $0.6880_{\pm0.0072}$ | $0.5934_{\pm0.0050}$ | $0.4828_{\pm0.0121}$ |
| GraphBTM | $0.1965_{\pm0.0437}$ | $0.1445_{\pm0.0534}$ | $0.1747_{\pm0.0202}$ | $0.1696_{\pm0.0367}$ | $0.1383_{\pm0.0210}$ | $0.2150_{\pm0.0156}$ | $0.1726_{\pm0.0052}$ | $0.1749_{\pm0.0276}$ | $0.1852_{\pm0.0128}$ | $0.2300_{\pm0.0222}$ |
| iDocNADE | $0.5890_{\pm0.0467}$ | $0.5710_{\pm0.0411}$ | $0.5854_{\pm0.0252}$ | $0.6012_{\pm0.0417}$ | $0.5788_{\pm0.0333}$ | $0.5106_{\pm0.0249}$ | $0.4790_{\pm0.0177}$ | $0.4266_{\pm0.0123}$ | $0.4038_{\pm0.0109}$ | $0.3860_{\pm0.0083}$ |
| iDocNADEe | - | - | $0.6240_{\pm0.0232}$ | - | $0.5884_{\pm0.0129}$ | - | - | $0.4494_{\pm0.0104}$ | - | $0.3980_{\pm0.0062}$ |
| TopicRNN | $0.9562_{\pm0.0022}$ | $0.9440_{\pm0.0133}$ | $0.9354_{\pm0.0070}$ | $0.9246_{\pm0.0075}$ | $0.9148_{\pm0.0079}$ | $0.9416_{\pm0.0123}$ | $0.9348_{\pm0.0126}$ | $0.9188_{\pm0.0165}$ | $0.9022_{\pm0.0075}$ | $0.8822_{\pm0.0022}$ |
| GNTM | $0.8890_{\pm0.0367}$ | $0.9034_{\pm0.0170}$ | $0.8224_{\pm0.0264}$ | $0.6834_{\pm0.0732}$ | $0.5944_{\pm0.0544}$ | $0.7890_{\pm0.0253}$ | $0.7128_{\pm0.0236}$ | $0.6384_{\pm0.0131}$ | $0.5976_{\pm0.0108}$ | $0.5438_{\pm0.0077}$ |

Table S13: The performance of topic coherence based on NPMI metric with top 20 words.

| | 20NG | | | | | TMN | | | | |
|---|---|---|---|---|---|---|---|---|---|---|
| | 20 | 30 | 50 | 70 | 100 | 20 | 30 | 50 | 70 | 100 |
| LDA | $0.0214_{\pm0.0150}$ | $0.0075_{\pm0.0121}$ | $-0.0257_{\pm0.0030}$ | $-0.0437_{\pm0.0053}$ | $-0.0691_{\pm0.0028}$ | $-0.1973_{\pm0.0115}$ | $-0.2329_{\pm0.0061}$ | $-0.2955_{\pm0.0035}$ | $-0.3221_{\pm0.0017}$ | $-0.2529_{\pm0.0049}$ |
| GSM | $0.0590_{\pm0.0108}$ | $0.0487_{\pm0.0085}$ | $0.0313_{\pm0.0071}$ | $0.0167_{\pm0.0046}$ | $0.0266_{\pm0.0033}$ | $-0.0784_{\pm0.0101}$ | $-0.0928_{\pm0.0109}$ | $-0.0554_{\pm0.0092}$ | $-0.0531_{\pm0.0058}$ | $-0.1534_{\pm0.0061}$ |
| ProdLDA | $-0.0531_{\pm0.0151}$ | $-0.0700_{\pm0.0212}$ | $-0.0900_{\pm0.0131}$ | $-0.0960_{\pm0.0132}$ | $-0.1010_{\pm0.0149}$ | $-0.2195_{\pm0.0198}$ | $-0.2283_{\pm0.0121}$ | $-0.2581_{\pm0.0219}$ | $-0.2804_{\pm0.0234}$ | $-0.3075_{\pm0.0152}$ |
| ETM | $0.0482_{\pm0.0081}$ | $0.0489_{\pm0.0093}$ | $0.0319_{\pm0.0052}$ | $0.0182_{\pm0.0024}$ | $0.0041_{\pm0.0031}$ | $-0.0966_{\pm0.0126}$ | $-0.1026_{\pm0.0028}$ | $-0.1103_{\pm0.0065}$ | $-0.1247_{\pm0.0085}$ | $-0.1368_{\pm0.0058}$ |
| GraphBTM | $0.0244_{\pm0.0022}$ | $0.0266_{\pm0.0132}$ | $0.0327_{\pm0.0025}$ | $0.0137_{\pm0.0242}$ | $-0.0463_{\pm0.0508}$ | $-0.1661_{\pm0.0201}$ | $-0.2056_{\pm0.0065}$ | $-0.2757_{\pm0.0042}$ | $-0.3064_{\pm0.0034}$ | $-0.3200_{\pm0.0039}$ |
| iDocNADE | $-0.2435_{\pm0.0155}$ | $-0.2698_{\pm0.0086}$ | $-0.2826_{\pm0.0065}$ | $-0.2913_{\pm0.0069}$ | $-0.3002_{\pm0.0045}$ | $-0.5105_{\pm0.0058}$ | $-0.4895_{\pm0.0045}$ | $-0.4659_{\pm0.0075}$ | $-0.4536_{\pm0.0071}$ | $-0.4399_{\pm0.0054}$ |
| iDocNADEe | - | - | $-0.2072_{\pm0.0450}$ | - | $-0.2440_{\pm0.0419}$ | - | - | $-0.4530_{\pm0.0091}$ | - | $-0.4513_{\pm0.0023}$ |
| TopicRNN | $-0.2394_{\pm0.0085}$ | $-0.2783_{\pm0.0112}$ | $-0.3125_{\pm0.0054}$ | $-0.3303_{\pm0.0023}$ | $-0.3423_{\pm0.0063}$ | $-0.4584_{\pm0.0092}$ | $-0.4443_{\pm0.0072}$ | $-0.4426_{\pm0.0060}$ | $-0.4412_{\pm0.0015}$ | $-0.4442_{\pm0.0028}$ |
| GNTM | $0.0860_{\pm0.0064}$ | $0.0643_{\pm0.0072}$ | $0.0356_{\pm0.0064}$ | $0.0263_{\pm0.0046}$ | $0.0172_{\pm0.0039}$ | $-0.0739_{\pm0.0114}$ | $-0.0862_{\pm0.0108}$ | $-0.1051_{\pm0.0038}$ | $-0.1158_{\pm0.0072}$ | $-0.1139_{\pm0.0228}$ |

| | BNC | | | | | Reuters | | | | |
|---|---|---|---|---|---|---|---|---|---|---|
| | 20 | 30 | 50 | 70 | 100 | 20 | 30 | 50 | 70 | 100 |
| LDA | $0.0572_{\pm0.0050}$ | $0.0571_{\pm0.0098}$ | $0.0580_{\pm0.0111}$ | $0.0468_{\pm0.0105}$ | $0.0344_{\pm0.0165}$ | $0.0442_{\pm0.0069}$ | $0.0184_{\pm0.0141}$ | $-0.0058_{\pm0.0125}$ | $-0.0284_{\pm0.0093}$ | $-0.0572_{\pm0.0045}$ |
| GSM | $0.0612_{\pm0.0032}$ | $0.0678_{\pm0.0024}$ | $0.0679_{\pm0.0027}$ | $0.0589_{\pm0.0048}$ | $0.0431_{\pm0.0017}$ | $0.0684_{\pm0.0044}$ | $0.0525_{\pm0.0057}$ | $0.0222_{\pm0.0100}$ | $0.0006_{\pm0.0124}$ | $-0.0010_{\pm0.0038}$ |
| ProdLDA | $0.0478_{\pm0.0116}$ | $0.0359_{\pm0.0125}$ | $0.0160_{\pm0.0054}$ | $-0.0009_{\pm0.0067}$ | $-0.0065_{\pm0.0087}$ | $-0.0382_{\pm0.0064}$ | $-0.0402_{\pm0.0075}$ | $-0.0374_{\pm0.0076}$ | $-0.0528_{\pm0.0079}$ | $-0.0489_{\pm0.0075}$ |
| ETM | $0.0687_{\pm0.0072}$ | $0.0634_{\pm0.0062}$ | $0.0591_{\pm0.0024}$ | $0.0537_{\pm0.0029}$ | $0.0472_{\pm0.0029}$ | $0.0133_{\pm0.0096}$ | $0.0054_{\pm0.0036}$ | $-0.0110_{\pm0.0081}$ | $-0.0178_{\pm0.0038}$ | $-0.0232_{\pm0.0033}$ |
| GraphBTM | $0.0170_{\pm0.0039}$ | $0.0050_{\pm0.0101}$ | $0.0204_{\pm0.0120}$ | $0.0316_{\pm0.0166}$ | $-0.0035_{\pm0.0001}$ | $0.0069_{\pm0.0098}$ | $0.0130_{\pm0.0100}$ | $0.0165_{\pm0.0066}$ | $0.0101_{\pm0.0097}$ | $-0.0196_{\pm0.0222}$ |
| iDocNADE | $-0.0889_{\pm0.0412}$ | $-0.1130_{\pm0.0316}$ | $-0.1648_{\pm0.0184}$ | $-0.2056_{\pm0.0132}$ | $-0.2161_{\pm0.0048}$ | $-0.2554_{\pm0.0119}$ | $-0.2863_{\pm0.0128}$ | $-0.3089_{\pm0.0058}$ | $-0.3251_{\pm0.0070}$ | $-0.3358_{\pm0.0036}$ |
| iDocNADEe | - | - | $-0.1323_{\pm0.0073}$ | - | $-0.1847_{\pm0.0055}$ | - | - | $-0.2742_{\pm0.0053}$ | - | $-0.3176_{\pm0.0008}$ |
| TopicRNN | $-0.1469_{\pm0.0178}$ | $-0.1981_{\pm0.0220}$ | $-0.2341_{\pm0.0087}$ | $-0.2627_{\pm0.0091}$ | $-0.2870_{\pm0.0087}$ | $-0.2974_{\pm0.0102}$ | $-0.3219_{\pm0.0080}$ | $-0.3466_{\pm0.0060}$ | $-0.3566_{\pm0.0033}$ | $-0.3632_{\pm0.0022}$ |
| GNTM | $0.0752_{\pm0.0025}$ | $0.0754_{\pm0.0024}$ | $0.0671_{\pm0.0022}$ | $0.0676_{\pm0.0031}$ | $0.0609_{\pm0.0024}$ | $0.0542_{\pm0.0103}$ | $0.0396_{\pm0.0082}$ | $0.0296_{\pm0.0042}$ | $-0.0012_{\pm0.0040}$ | $-0.0084_{\pm0.0058}$ |

Table S14: The performance of topic coherence based on $C_v$ metric with top 20 words.

| | 20NG | | | | | TMN | | | | |
|---|---|---|---|---|---|---|---|---|---|---|
| | 20 | 30 | 50 | 70 | 100 | 20 | 30 | 50 | 70 | 100 |
| LDA | $0.5959_{\pm0.0139}$ | $0.5662_{\pm0.0266}$ | $0.5347_{\pm0.0130}$ | $0.5106_{\pm0.0055}$ | $0.4856_{\pm0.0055}$ | $0.3638_{\pm0.0103}$ | $0.3983_{\pm0.0080}$ | $0.4821_{\pm0.0042}$ | $0.5200_{\pm0.0028}$ | $0.4334_{\pm0.0061}$ |
| GSM | $0.6065_{\pm0.0202}$ | $0.5789_{\pm0.0224}$ | $0.5522_{\pm0.0191}$ | $0.5208_{\pm0.0068}$ | $0.5260_{\pm0.0119}$ | $0.3898_{\pm0.0271}$ | $0.3890_{\pm0.0198}$ | $0.3228_{\pm0.0125}$ | $0.2461_{\pm0.0063}$ | $0.2805_{\pm0.0116}$ |
| ProdLDA | $0.5535_{\pm0.0149}$ | $0.5124_{\pm0.0280}$ | $0.4729_{\pm0.0143}$ | $0.4770_{\pm0.0144}$ | $0.4573_{\pm0.0115}$ | $0.4000_{\pm0.0226}$ | $0.4042_{\pm0.0150}$ | $0.4394_{\pm0.0286}$ | $0.4692_{\pm0.0316}$ | $0.5083_{\pm0.0245}$ |
| ETM | $0.6258_{\pm0.0173}$ | $0.6199_{\pm0.0157}$ | $0.5934_{\pm0.0085}$ | $0.5670_{\pm0.0084}$ | $0.5471_{\pm0.0082}$ | $0.4077_{\pm0.0075}$ | $0.4011_{\pm0.0103}$ | $0.4021_{\pm0.0104}$ | $0.4024_{\pm0.0094}$ | $0.4061_{\pm0.0093}$ |
| GraphBTM | $0.4994_{\pm0.0153}$ | $0.5121_{\pm0.0491}$ | $0.5469_{\pm0.0078}$ | $0.4891_{\pm0.0692}$ | $0.4003_{\pm0.0773}$ | $0.3120_{\pm0.0380}$ | $0.3774_{\pm0.0140}$ | $0.4749_{\pm0.0050}$ | $0.5119_{\pm0.0053}$ | $0.5254_{\pm0.0062}$ |
| iDocNADE | $0.4917_{\pm0.0278}$ | $0.5188_{\pm0.0096}$ | $0.5267_{\pm0.0079}$ | $0.5267_{\pm0.0074}$ | $0.5279_{\pm0.0043}$ | $0.7202_{\pm0.0059}$ | $0.7032_{\pm0.0058}$ | $0.6807_{\pm0.0091}$ | $0.6687_{\pm0.0072}$ | $0.6558_{\pm0.0055}$ |
| iDocNADEe | - | - | $0.3998_{\pm0.0410}$ | - | $0.4480_{\pm0.0515}$ | - | - | $0.6642_{\pm0.0101}$ | - | $0.6702_{\pm0.0030}$ |
| TopicRNN | $0.4508_{\pm0.0207}$ | $0.4708_{\pm0.0141}$ | $0.4914_{\pm0.0055}$ | $0.5248_{\pm0.0044}$ | $0.5448_{\pm0.0122}$ | $0.6723_{\pm0.0136}$ | $0.6641_{\pm0.0097}$ | $0.6659_{\pm0.0079}$ | $0.6697_{\pm0.0029}$ | $0.6768_{\pm0.0053}$ |
| GNTM | $0.6635_{\pm0.0139}$ | $0.6253_{\pm0.0049}$ | $0.5870_{\pm0.0061}$ | $0.5693_{\pm0.0086}$ | $0.5386_{\pm0.0101}$ | $0.3606_{\pm0.0155}$ | $0.3765_{\pm0.0085}$ | $0.4009_{\pm0.0066}$ | $0.4046_{\pm0.0114}$ | $0.3972_{\pm0.0105}$ |

| | BNC | | | | | Reuters | | | | |
|---|---|---|---|---|---|---|---|---|---|---|
| | 20 | 30 | 50 | 70 | 100 | 20 | 30 | 50 | 70 | 100 |
| LDA | $0.5305_{\pm0.0080}$ | $0.5557_{\pm0.0071}$ | $0.5639_{\pm0.0040}$ | $0.5581_{\pm0.0062}$ | $0.5543_{\pm0.0146}$ | $0.4979_{\pm0.0099}$ | $0.4809_{\pm0.0219}$ | $0.4635_{\pm0.0159}$ | $0.4466_{\pm0.0068}$ | $0.4327_{\pm0.0092}$ |
| GSM | $0.5351_{\pm0.0096}$ | $0.5525_{\pm0.0076}$ | $0.5487_{\pm0.0080}$ | $0.5135_{\pm0.0125}$ | $0.4611_{\pm0.0049}$ | $0.5363_{\pm0.0111}$ | $0.5031_{\pm0.0073}$ | $0.4547_{\pm0.0080}$ | $0.4133_{\pm0.0234}$ | $0.3822_{\pm0.0081}$ |
| ProdLDA | $0.5403_{\pm0.0365}$ | $0.5163_{\pm0.0096}$ | $0.4891_{\pm0.0185}$ | $0.4632_{\pm0.0115}$ | $0.4504_{\pm0.0084}$ | $0.4744_{\pm0.0069}$ | $0.4594_{\pm0.0085}$ | $0.4628_{\pm0.0094}$ | $0.4424_{\pm0.0130}$ | $0.4397_{\pm0.0102}$ |
| ETM | $0.5707_{\pm0.0101}$ | $0.5669_{\pm0.0108}$ | $0.5535_{\pm0.0081}$ | $0.5314_{\pm0.0083}$ | $0.5103_{\pm0.0073}$ | $0.5073_{\pm0.0107}$ | $0.4749_{\pm0.0061}$ | $0.4678_{\pm0.0066}$ | $0.4548_{\pm0.0066}$ | $0.4538_{\pm0.0038}$ |
| GraphBTM | $0.3955_{\pm0.0163}$ | $0.3314_{\pm0.0462}$ | $0.3985_{\pm0.0521}$ | $0.4428_{\pm0.0678}$ | $0.2995_{\pm0.0021}$ | $0.4096_{\pm0.0209}$ | $0.4160_{\pm0.0244}$ | $0.4290_{\pm0.0143}$ | $0.4270_{\pm0.0237}$ | $0.3786_{\pm0.0509}$ |
| iDocNADE | $0.3866_{\pm0.0171}$ | $0.3819_{\pm0.0124}$ | $0.3822_{\pm0.0093}$ | $0.4054_{\pm0.0172}$ | $0.4092_{\pm0.0181}$ | $0.5001_{\pm0.0087}$ | $0.5136_{\pm0.0141}$ | $0.5222_{\pm0.0119}$ | $0.5368_{\pm0.0068}$ | $0.5510_{\pm0.0060}$ |
| iDocNADEe | - | - | $0.2846_{\pm0.0068}$ | - | $0.3162_{\pm0.0043}$ | - | - | $0.4480_{\pm0.0078}$ | - | $0.5090_{\pm0.0052}$ |
| TopicRNN | $0.3727_{\pm0.0190}$ | $0.3510_{\pm0.0174}$ | $0.3485_{\pm0.0133}$ | $0.3546_{\pm0.0094}$ | $0.3715_{\pm0.0083}$ | $0.4826_{\pm0.0090}$ | $0.5239_{\pm0.0070}$ | $0.5609_{\pm0.0072}$ | $0.5898_{\pm0.0070}$ | $0.6113_{\pm0.0092}$ |
| GNTM | $0.5889_{\pm0.0087}$ | $0.5917_{\pm0.0070}$ | $0.5724_{\pm0.0038}$ | $0.5683_{\pm0.0074}$ | $0.5517_{\pm0.0045}$ | $0.5220_{\pm0.0140}$ | $0.4920_{\pm0.0188}$ | $0.4825_{\pm0.0104}$ | $0.4569_{\pm0.0073}$ | $0.4513_{\pm0.0129}$ |

Table S15: The performance of topic diversity with top 20 words.

| | 20NG | | | | | TMN | | | | |
|---|---|---|---|---|---|---|---|---|---|---|
| | 20 | 30 | 50 | 70 | 100 | 20 | 30 | 50 | 70 | 100 |
| LDA | $0.7688_{\pm0.0152}$ | $0.7810_{\pm0.0226}$ | $0.7736_{\pm0.0099}$ | $0.7698_{\pm0.0051}$ | $0.7648_{\pm0.0092}$ | $0.9784_{\pm0.0045}$ | $0.9784_{\pm0.0059}$ | $0.9842_{\pm0.0013}$ | $0.9866_{\pm0.0010}$ | $0.5504_{\pm0.0150}$ |
| GSM | $0.6696_{\pm0.0170}$ | $0.6372_{\pm0.0282}$ | $0.5770_{\pm0.0110}$ | $0.4586_{\pm0.0214}$ | $0.2862_{\pm0.0131}$ | $0.9036_{\pm0.0087}$ | $0.8568_{\pm0.0168}$ | $0.5454_{\pm0.0301}$ | $0.2374_{\pm0.0128}$ | $0.0520_{\pm0.0089}$ |
| ProdLDA | $0.8988_{\pm0.0118}$ | $0.8054_{\pm0.0119}$ | $0.7070_{\pm0.0083}$ | $0.6388_{\pm0.0160}$ | $0.5578_{\pm0.0050}$ | $0.7146_{\pm0.0284}$ | $0.5320_{\pm0.0373}$ | $0.5022_{\pm0.0675}$ | $0.4988_{\pm0.0598}$ | $0.4970_{\pm0.0459}$ |
| ETM | $0.8142_{\pm0.0321}$ | $0.7416_{\pm0.0113}$ | $0.6316_{\pm0.0111}$ | $0.5622_{\pm0.0098}$ | $0.4966_{\pm0.0072}$ | $0.9332_{\pm0.0253}$ | $0.8742_{\pm0.0112}$ | $0.7020_{\pm0.0123}$ | $0.6124_{\pm0.0096}$ | $0.5090_{\pm0.0155}$ |
| GraphBTM | $0.2723_{\pm0.0166}$ | $0.2118_{\pm0.0408}$ | $0.2312_{\pm0.0171}$ | $0.2143_{\pm0.0545}$ | $0.3404_{\pm0.1057}$ | $0.4523_{\pm0.0797}$ | $0.5217_{\pm0.0156}$ | $0.6180_{\pm0.0064}$ | $0.6235_{\pm0.0075}$ | $0.5622_{\pm0.0075}$ |
| iDocNADE | $0.7272_{\pm0.0306}$ | $0.6706_{\pm0.0122}$ | $0.6376_{\pm0.0021}$ | $0.5982_{\pm0.0154}$ | $0.5638_{\pm0.0117}$ | $0.5842_{\pm0.0149}$ | $0.5060_{\pm0.0129}$ | $0.3966_{\pm0.0096}$ | $0.3264_{\pm0.0029}$ | $0.2668_{\pm0.0061}$ |
| iDocNADEe | - | - | $0.5274_{\pm0.0417}$ | - | $0.5098_{\pm0.0443}$ | - | - | $0.3552_{\pm0.0100}$ | - | $0.2922_{\pm0.0039}$ |
| TopicRNN | $0.9342_{\pm0.0117}$ | $0.9228_{\pm0.0063}$ | $0.9182_{\pm0.0079}$ | $0.9042_{\pm0.0016}$ | $0.8686_{\pm0.0069}$ | $0.7248_{\pm0.0399}$ | $0.6748_{\pm0.0330}$ | $0.5914_{\pm0.0174}$ | $0.5586_{\pm0.0153}$ | $0.5264_{\pm0.0199}$ |
| GNTM | $0.8468_{\pm0.0099}$ | $0.7991_{\pm0.0262}$ | $0.7394_{\pm0.0199}$ | $0.6169_{\pm0.0307}$ | $0.4632_{\pm0.0125}$ | $0.9962_{\pm0.0026}$ | $0.9926_{\pm0.0024}$ | $0.9814_{\pm0.0099}$ | $0.9512_{\pm0.0122}$ | $0.8278_{\pm0.0720}$ |

| | BNC | | | | | Reuters | | | | |
|---|---|---|---|---|---|---|---|---|---|---|
| | 20 | 30 | 50 | 70 | 100 | 20 | 30 | 50 | 70 | 100 |
| LDA | $0.7222_{\pm0.0314}$ | $0.7234_{\pm0.0324}$ | $0.6932_{\pm0.0284}$ | $0.7028_{\pm0.0183}$ | $0.7034_{\pm0.0126}$ | $0.6748_{\pm0.0143}$ | $0.6666_{\pm0.0217}$ | $0.6394_{\pm0.0060}$ | $0.6286_{\pm0.0123}$ | $0.6284_{\pm0.0068}$ |
| GSM | $0.7076_{\pm0.0056}$ | $0.6220_{\pm0.0122}$ | $0.5506_{\pm0.0048}$ | $0.4620_{\pm0.0266}$ | $0.3114_{\pm0.0118}$ | $0.5867_{\pm0.0152}$ | $0.5253_{\pm0.0101}$ | $0.4564_{\pm0.0130}$ | $0.3243_{\pm0.0256}$ | $0.2027_{\pm0.0136}$ |
| ProdLDA | $0.8458_{\pm0.0362}$ | $0.7530_{\pm0.0108}$ | $0.6506_{\pm0.0205}$ | $0.5736_{\pm0.0107}$ | $0.4618_{\pm0.0136}$ | $0.8888_{\pm0.0085}$ | $0.8268_{\pm0.0113}$ | $0.7058_{\pm0.0078}$ | $0.6076_{\pm0.0137}$ | $0.5062_{\pm0.0120}$ |
| ETM | $0.8497_{\pm0.0074}$ | $0.8093_{\pm0.0088}$ | $0.7264_{\pm0.0056}$ | $0.6526_{\pm0.0067}$ | $0.5799_{\pm0.0123}$ | $0.8560_{\pm0.0080}$ | $0.7686_{\pm0.0117}$ | $0.6634_{\pm0.0077}$ | $0.5696_{\pm0.0057}$ | $0.4640_{\pm0.0092}$ |
| GraphBTM | $0.2019_{\pm0.0346}$ | $0.1431_{\pm0.0514}$ | $0.1664_{\pm0.0171}$ | $0.1616_{\pm0.0326}$ | $0.1301_{\pm0.0200}$ | $0.2223_{\pm0.0197}$ | $0.1797_{\pm0.0081}$ | $0.1716_{\pm0.0275}$ | $0.1779_{\pm0.0146}$ | $0.2210_{\pm0.0248}$ |
| iDocNADE | $0.5764_{\pm0.0570}$ | $0.5698_{\pm0.0460}$ | $0.5824_{\pm0.0240}$ | $0.5910_{\pm0.0359}$ | $0.5660_{\pm0.0322}$ | $0.5118_{\pm0.0204}$ | $0.4772_{\pm0.0128}$ | $0.4144_{\pm0.0149}$ | $0.3950_{\pm0.0070}$ | $0.3764_{\pm0.0039}$ |
| iDocNADEe | - | - | $0.6106_{\pm0.0172}$ | - | $0.5720_{\pm0.0128}$ | - | - | $0.4320_{\pm0.0099}$ | - | $0.3850_{\pm0.0088}$ |
| TopicRNN | $0.9452_{\pm0.0103}$ | $0.9270_{\pm0.0129}$ | $0.9152_{\pm0.0043}$ | $0.9054_{\pm0.0059}$ | $0.8910_{\pm0.0066}$ | $0.9278_{\pm0.0108}$ | $0.9198_{\pm0.0083}$ | $0.9052_{\pm0.0138}$ | $0.8786_{\pm0.0079}$ | $0.8532_{\pm0.0059}$ |
| GNTM | $0.8724_{\pm0.0411}$ | $0.8782_{\pm0.0173}$ | $0.7874_{\pm0.0252}$ | $0.6450_{\pm0.0717}$ | $0.5526_{\pm0.0496}$ | $0.7402_{\pm0.0240}$ | $0.6676_{\pm0.0203}$ | $0.5892_{\pm0.0079}$ | $0.5442_{\pm0.0137}$ | $0.4918_{\pm0.0110}$ |

Table S16: The performance of the document cluster on 20NG.

| | Purity | | | | | NMI | | | | |
|---|---|---|---|---|---|---|---|---|---|---|
| | 20 | 30 | 50 | 70 | 100 | 20 | 30 | 50 | 70 | 100 |
| LDA | $0.2980_{\pm0.0131}$ | $0.3340_{\pm0.0106}$ | $0.3375_{\pm0.0086}$ | $0.3510_{\pm0.0098}$ | $0.3740_{\pm0.0138}$ | $0.2908_{\pm0.0097}$ | $0.3013_{\pm0.0064}$ | $0.2852_{\pm0.0033}$ | $0.2878_{\pm0.0085}$ | $0.2858_{\pm0.0092}$ |
| GSM | $0.4133_{\pm0.0268}$ | $0.4379_{\pm0.0133}$ | $0.4629_{\pm0.0180}$ | $0.4429_{\pm0.0137}$ | $0.4210_{\pm0.0290}$ | $0.4394_{\pm0.0083}$ | $0.4369_{\pm0.0158}$ | $0.4433_{\pm0.0059}$ | $0.4449_{\pm0.0064}$ | $0.4412_{\pm0.0202}$ |
| ProdLDA | $0.3306_{\pm0.0168}$ | $0.3450_{\pm0.0221}$ | $0.3641_{\pm0.0108}$ | $0.3638_{\pm0.0210}$ | $0.3807_{\pm0.0244}$ | $0.3405_{\pm0.0178}$ | $0.3345_{\pm0.0133}$ | $0.3350_{\pm0.0035}$ | $0.3298_{\pm0.0090}$ | $0.3343_{\pm0.0170}$ |
| ETM | $0.3496_{\pm0.0173}$ | $0.4154_{\pm0.0134}$ | $0.4380_{\pm0.0136}$ | $0.4510_{\pm0.0174}$ | $0.4616_{\pm0.0079}$ | $0.3842_{\pm0.0097}$ | $0.4227_{\pm0.0063}$ | $0.4296_{\pm0.0017}$ | $0.4297_{\pm0.0090}$ | $0.4356_{\pm0.0033}$ |
| GraphBTM | $0.1448_{\pm0.0084}$ | $0.1210_{\pm0.0454}$ | $0.1630_{\pm0.0125}$ | $0.1068_{\pm0.0609}$ | $0.0992_{\pm0.0497}$ | $0.1552_{\pm0.0185}$ | $0.1108_{\pm0.0825}$ | $0.1807_{\pm0.0135}$ | $0.0816_{\pm0.0956}$ | $0.0707_{\pm0.0848}$ |
| iDocNADE | $0.2175_{\pm0.1087}$ | $0.2844_{\pm0.0796}$ | $0.3064_{\pm0.0758}$ | $0.3187_{\pm0.0758}$ | $0.3371_{\pm0.0756}$ | $0.1128_{\pm0.0444}$ | $0.1723_{\pm0.0771}$ | $0.1802_{\pm0.0728}$ | $0.1901_{\pm0.0737}$ | $0.1990_{\pm0.0684}$ |
| iDocNADEe | - | - | $0.1300_{\pm0.0005}$ | - | $0.1507_{\pm0.0035}$ | - | - | $0.1185_{\pm0.0016}$ | - | $0.1228_{\pm0.0017}$ |
| TopicRNN | $0.0728_{\pm0.0022}$ | $0.0761_{\pm0.0020}$ | $0.0865_{\pm0.0022}$ | $0.0920_{\pm0.0010}$ | $0.1001_{\pm0.0002}$ | $0.0109_{\pm0.0007}$ | $0.0141_{\pm0.0006}$ | $0.0229_{\pm0.0006}$ | $0.0303_{\pm0.0011}$ | $0.0400_{\pm0.0004}$ |
| GNTM | $0.4500_{\pm0.0222}$ | $0.4882_{\pm0.0328}$ | $0.5089_{\pm0.0136}$ | $0.5090_{\pm0.0156}$ | $0.5021_{\pm0.0241}$ | $0.4436_{\pm0.0084}$ | $0.4419_{\pm0.0102}$ | $0.4416_{\pm0.0060}$ | $0.4362_{\pm0.0084}$ | $0.4371_{\pm0.0044}$ |

Table S17: The performance of the document cluster on TMN.

| | Purity | | | | | NMI | | | | |
|---|---|---|---|---|---|---|---|---|---|---|
| | 20 | 30 | 50 | 70 | 100 | 20 | 30 | 50 | 70 | 100 |
| LDA | $0.3509_{\pm0.0093}$ | $0.3692_{\pm0.0026}$ | $0.3725_{\pm0.0072}$ | $0.4031_{\pm0.0033}$ | $0.4228_{\pm0.0108}$ | $0.0622_{\pm0.0055}$ | $0.0665_{\pm0.0016}$ | $0.0754_{\pm0.0040}$ | $0.0901_{\pm0.0016}$ | $0.1064_{\pm0.0069}$ |
| GSM | $0.5933_{\pm0.0170}$ | $0.6054_{\pm0.0182}$ | $0.6184_{\pm0.0096}$ | $0.5934_{\pm0.0233}$ | $0.2632_{\pm0.0205}$ | $0.2848_{\pm0.0127}$ | $0.2787_{\pm0.0138}$ | $0.2996_{\pm0.0099}$ | $0.3246_{\pm0.0084}$ | $0.0204_{\pm0.0149}$ |
| ProdLDA | $0.3141_{\pm0.0159}$ | $0.2808_{\pm0.0208}$ | $0.2505_{\pm0.0065}$ | $0.2535_{\pm0.0194}$ | $0.2438_{\pm0.0000}$ | $0.0508_{\pm0.0198}$ | $0.0334_{\pm0.0108}$ | $0.0056_{\pm0.0047}$ | $0.0053_{\pm0.0107}$ | $0.0000_{\pm0.0000}$ |
| ETM | $0.5841_{\pm0.0160}$ | $0.5967_{\pm0.0109}$ | $0.6347_{\pm0.0141}$ | $0.6358_{\pm0.0065}$ | $0.6420_{\pm0.0086}$ | $0.2764_{\pm0.0095}$ | $0.2705_{\pm0.0059}$ | $0.2829_{\pm0.0088}$ | $0.2784_{\pm0.0052}$ | $0.2767_{\pm0.0066}$ |
| GraphBTM | $0.2438_{\pm0.0000}$ | $0.2438_{\pm0.0000}$ | $0.2438_{\pm0.0000}$ | $0.2438_{\pm0.0000}$ | $0.2438_{\pm0.0000}$ | $0.0000_{\pm0.0000}$ | $0.0000_{\pm0.0000}$ | $0.0000_{\pm0.0000}$ | $0.0000_{\pm0.0000}$ | $0.0000_{\pm0.0000}$ |
| iDocNADE | $0.2712_{\pm0.0962}$ | $0.3116_{\pm0.0766}$ | $0.3314_{\pm0.0766}$ | $0.3512_{\pm0.0751}$ | $0.3730_{\pm0.0727}$ | $0.1074_{\pm0.0643}$ | $0.1436_{\pm0.0728}$ | $0.1488_{\pm0.0681}$ | $0.1585_{\pm0.0654}$ | $0.1701_{\pm0.0647}$ |
| iDocNADEe | - | - | $0.2623_{\pm0.0094}$ | - | $0.4171_{\pm0.0067}$ | - | - | $0.0332_{\pm0.0054}$ | - | $0.1728_{\pm0.0022}$ |
| TopicRNN | $0.2493_{\pm0.0014}$ | $0.2537_{\pm0.0007}$ | $0.2586_{\pm0.0017}$ | $0.2684_{\pm0.0024}$ | $0.2775_{\pm0.0053}$ | $0.0048_{\pm0.0006}$ | $0.0071_{\pm0.0005}$ | $0.0108_{\pm0.0014}$ | $0.0154_{\pm0.0004}$ | $0.0221_{\pm0.0014}$ |
| GNTM | $0.6150_{\pm0.0161}$ | $0.6252_{\pm0.0140}$ | $0.6472_{\pm0.0040}$ | $0.6656_{\pm0.0054}$ | $0.6773_{\pm0.0069}$ | $0.2851_{\pm0.0067}$ | $0.2801_{\pm0.0099}$ | $0.2789_{\pm0.0039}$ | $0.2821_{\pm0.0037}$ | $0.2870_{\pm0.0043}$ |

## A.8 More Examples for Learned Topics

Figure S2-S4 show the two randomly selected learned topics on TMN, BNC, and Reuters datasets, respectively, with $K = 20$. We report the top 8 words, and top 8 edges and corresponding topic graph representation in the top 30 words, where the sizes of nodes and edges are positively correlated to $\beta_k^v$ and $A_k = (\beta_k^v \cdot (\beta_k^v)^T) \odot \beta_k^e$, respectively. Nodes are colored by their cluster labels detected by fast unfloding algorithm [3], where isolated nodes are coloured in gray.

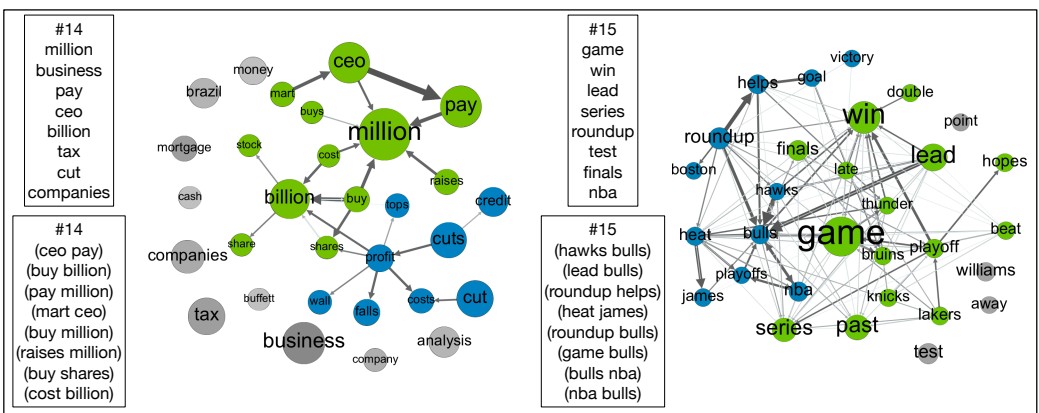

Figure S2: The illustration of learned topics on TMN dataset.

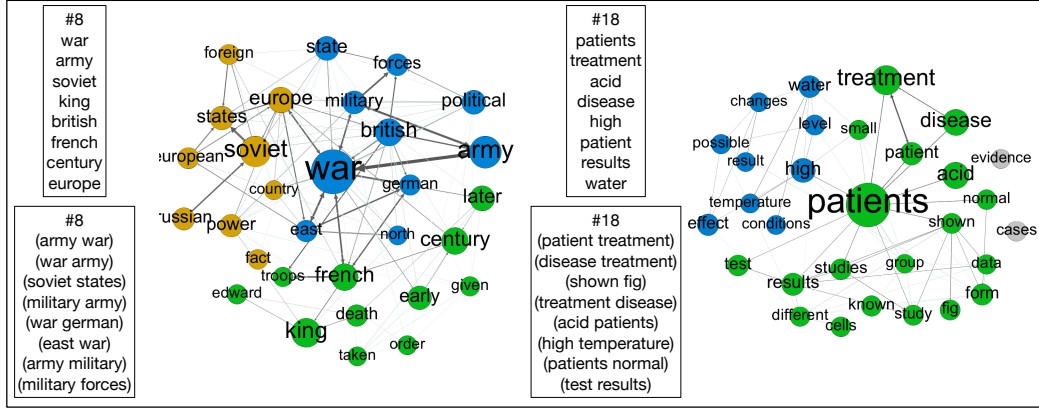

Figure S3: The illustration of learned topics on BNC dataset.

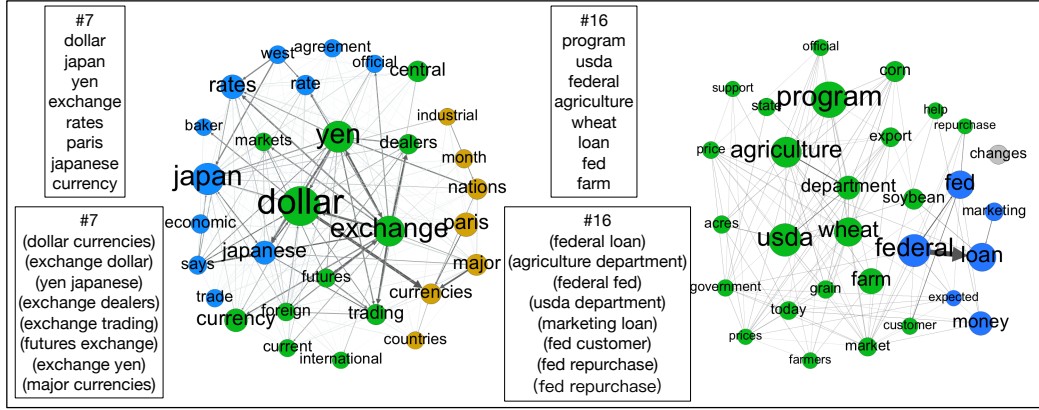

Figure S4: The illustration of learned topics on Reuters dataset.