# OpenReview forum: "Topic Modeling Revisited: A Document Graph-based Neural Network Perspective"
_NeurIPS.cc/2021/Conference — NeurIPS 2021 Poster_

### Official Review · Reviewer_EBBL · 2021-06-25

**Rating:** 5
**Confidence:** 4

**Summary:**

The authors propose a topic model that explicitly captures the semantic dependency among words in the same document, by representing each word as a node in a graph and connecting words appearing closely in the document through an edge in the graph.
This approach overcome the limitation of standard topic models as they can only consider word as independent. in this way, inferred topics may be more interpretable and semantically more meaningful. This is validated in the experiments as the proposed model achieves better topic coherence.

**Limitations And Societal Impact:**

Yes

**Main Review:**

One limitation of the approach is that the documents are split into windows to construct the graphs. This means that the model would disregard the co-occurrences of terms in the same document but in different positions of the document. This means that the model may link together words close-by, but lose more distant co-occurrences. It would be nice to see an experiment where this trade off is compared.

Also, the experiments are tested with a window of 5. I believe this is to limit the complexity of the method, as considering a bigger window size would require building a large number of subgraph. How much is the model scalable in terms of the window size? One limitation of standard topic models is that they require limited size of the vocabulary. By adding the complexity of creating subgraphs for each part of a document, I believe the model would require a limited vocabulary, . However, I could not find a comment on the practical scalability in the paper.

In terms of topic diversity, the results show that the proposed model performs worse when increasing the number of topics. Vice versa, a simple LDA is able to consistency achieving good or even better results with the increase of the number of topics. Additional comment on the results would be beneficial to understand this.


**Time Spent Reviewing:**

3

---

> ### Author Response · Authors · 2021-08-10
> **Responses to Reviewer EBBL**
>
> Thanks for your valuable suggestions and comments. Here, we provide detailed responses to your comments.
>
> ***Comment 1**: One limitation of the approach is that the documents are split into windows to construct the graphs … … It would be nice to see an experiment where this trade off is compared.*
>
> **Response 1**: Actually, our model has involved both the local information (extracted from the neighbor words) and global information (extracted from the whole document), to capture co-occurrences of near words and distant words. As the description shows in Section 3.2.2, we apply graph neural networks to encode each node as the local information and produce the graph-level representation as to the global information by integrating each node following [1]. However, there indeed exists a trade-off about the selection of the window size when constructing document graphs. As the window size increasing, both more semantic correlation and noise would be introduced into our model. Therefore, in Appendix A.5, we have discussed the sensitivity of this hyper-parameter based on one dataset and found that a 5-size window is the best choice. We would add more description about this in the revision.
>
>
> ***Comment 2**: Also, the experiments are tested with a window of 5. … … However, I could not find a comment on the practical scalability in the paper.*
>
> **Response 2**: Sorry for the confusion. Here, we make it clear that the selection of the window size as 5 is based on the sensitivity analysis in Appendix A.5. Actually, according to our experiments, the window size is not the direct factor to impact the complexity of our model. Instead, the sizes of vocabulary and the word dependency edge set mostly impact the complexity of our model. In recent days, we have run our model on 20NS dataset with different window sizes $s$ by fixing the size of both vocabulary and the word dependency edge set like that in Appendix A.4. The cost time for 100 iterations is 9.1571 seconds for $s=3$, 9.3214 seconds for $s=5$, 9.5142 seconds for $s=7$, and 9.5214 seconds for $s=10$.
>
> In practice, in order to limit the complexity of our model, we followed other topic models, such as LDA and ProdLDA, and filtered out words and word dependency edges with few frequencies. We have summarized the important dataset statistic after those data preprocessing in Appendix A.4, including the sizes of vocabulary and word dependency edge set used in our practical experiments. In particular, we found that our model can achieve a better trade-off between effectiveness and efficiency in most datasets, when constraining the size of the dependency edge set is about three times that of vocabulary.
>
> ***Comment 3**: In terms of topic diversity, the results show that … … on the results would be beneficial to understand this.*
>
> **Response 3**: Sorry for the confusion. Indeed, as the number of topics increases, topic models prefer to capture similar topics. As a result, the topic diversity of most models in our experiments is reduced when increasing the number of topics. As for the special phenomenon about LDA, it may be because LDA prefers to capture several different but meaningless topics. To be specific, we follow [2] and define topic diversity (TD) as the percentage of unique words in the top words. However, TD cannot measure the interpretability of topics independently. Sometimes, topic models can learn several meaningless topics with top words semantically uncorrelated, but top words of different topics are different from each other. In this case, we cannot claim that the model is equipped with high interpretability, although the TD score is high. In contrast, we must consider both topic coherence metrics and topic diversity metric to measure the interpretability of topic models, where TC measure the average interpretability in each topic and TD measures the diversity among different topics. In our experiments, LDA has captured a great TD score. However, the TC score is lower than our model and some other baselines, especially based on NPMI metric, which matches human judgment most closely [3]. In other words, LDA prefers to capture several different but meaningless topics. We would provide more explanation in the revision about this interesting phenomenon.
>
> [1] Li, Yujia, et al. Gated graph sequence neural networks. arXiv. 2015.
>
> [2] Dieng, Adji B., et al. Topic modeling in embedding spaces. TACL. 2020.
>
> [3] Newman, David, et al. "Automatic evaluation of topic coherence." NAACL. 2010.

---

### Official Review · Reviewer_zRWm · 2021-07-15

**Rating:** 7
**Confidence:** 4

**Summary:**

In this paper, the authors provide a novel generative model to learn topics using graph modeling based on the word network, following a (now classic) variational encoding approach. The documents are encoded as graphs by modeling an edge between two words if they appear in the same window, which is similar to the embedding methods with the distributional hypothesis. The graph structure is then part of the generative process. They demonstrate that the well-known LDA model is a special case of their approach if we ignore edges. Most of the time, the model outperforms existing topic models (classic such as LDA and neural TM) on 4 English written datasets in term of Cv, NPMI and Topic Diversity. It also outperforms prior works in term of clustering accuracy.

**Limitations And Societal Impact:**

We can read "N/A" in the form. I guess developing a new topic modeling approach let the authors ignore this point.

**Main Review:**

Pros:

The model is quite clearly defined. It is novel and interesting. All modeling choices are motivated by previous works, and the additional construction of a document graph using word neighborhood is highly relevant: the BOW hypothesis is one of the main weaknesses of previous topic models. Even if incorporating the sequential information was already studied (e.g., HTMM), there is still room for improvement.
The graph that one can construct at the end when the parameters are learned (Figure 4, left) is particularly interesting for interpretability. It seems that it can really improve the understanding of the topic content.
The approach seems to outperform (recent) prior works in term of topic coherence, diversity, and clustering.

Cons:

Not a rejection criterion per se, but your approach is beaten in few configurations, e.g. NMI on TMN and 20NG with a high number of topic for clustering and TD score for 20NG, BNC and Reuters, again with high number of topics. Do you have an intuition on why your method performs slightly worst when the number of topics grows?
Even if you provide the effect of the window size in supp.mat. on the 20NG dataset, it would be interesting to provide further experiments regarding this parameter, on more datasets. Similarly, you should provide more experiments to test the effect of the word embedding used on the performances of your model.
L35: “For example, LDA discovers word such as “station” in a topic which does not seem to be that insightful. » You do not introduce this example. Where does it come from? You cannot drop examples like that and expect that the reader will understand your point.
L 108: “We also filter out the stop words, and words and dependency edges with low frequency among the whole collection to reduce noise.”: it should be in the experimental section, and you should add the parameters you use (which stop words, the threshold for selection, etc…). All your experiments must be reproduceable.
The paper is not well written. There is a lot of grammatical errors, typos, and some sentences are not clear.
Additional remarks:
2.2 : there should be a way to simplify the notation, this subsection is hard to follow.
In the abstract and in the experiment section, you should use present tense, not past tense.
Why using AVI rather than the usual expression (VAE)?
L 100: “ we can donate a document” this is not clear, donate does not fit here
L 104: “with nodes on the vocabulary” : should be “in the vocabulary”
L 128:  just based  based solely on the topic assignment ?
L 136 : should be cast into  cast as
L 210 : “Now, the only challenge to compute the ELBO loss Ld is at the last term in Eqaution: “ this sentence is not grammatically correct and Eqaution should be Equation
L 58 : Graphic Neural Topic Modeling  Graph Neural Topic Modeling. This is the name of your method!

---

I've read the authors' response and the other reviews. The authors answered most of my remarks and I choose to keep my score of 7. I'd be interested to read the reactions of the other reviewers before, but they never came...


**Time Spent Reviewing:**

2h

---

> ### Author Response · Authors · 2021-08-10
> **Responses to Reviewer zRWm**
>
> Thanks very much for your positive comments about our paper and valuable suggestions. First, as per your suggestion, we would like to explore the sensitivity of the window size parameter based on more datasets and the impact of word embedding used on the performances of our model in the future work. We believe that those explorations can help us to enhance the evaluation experiments and inspire us to improve the design of our model. Meanwhile, we are sorry for our unclear description, grammatical error, and some typos. We would improve our presentation and written carefully. To be specific, other detailed responses can be found as follows.
>
> ***Comment 1**: L35: “For example, LDA discovers word such as “station” in a topic which does not seem to be that insightful. » You do not introduce this example. Where does it come from? You cannot drop examples like that and expect that the reader will understand your point.*
>
> **Response1**： We will modify this description and the example in Figure 1(a), and re-describe Line 35 based on the example.
>
> ***Comment 2**: L 108: “We also filter out the stop words, and words and dependency edges with low frequency among the whole collection to reduce noise.”: it should be in the experimental section, and you should add the parameters you use (which stop words, the threshold for selection, etc…). All your experiments must be reproduceable."*
>
> **Response 2**: We will move Line 108 into Section 4.1, and the related setting we used will be provided in Appendix A.4. Specifically, we use the stop word list same as that in genism package, which is a well-known python package for topic modeling and text processing. The frequency threshold to filter out low-frequency words is set as 20 for 20NS, 5 for TMN, 120 for BNC, and 10 for Reuters. The frequency threshold to filter out low-frequency dependency edges is set as 10 for 20NG, 3 for TMN, 40 for BNC, and 15 for Reuters.
>
>
> ***Comment 3**: 2.2 : there should be a way to simplify the notation, this subsection is hard to follow. In the abstract and in the experiment section, you should use present tense, not past tense. “*
>
> **Response 3**: We will simplify our notation and check the tense in our paper carefully to ensure correctness in the revision.
>
> ***Comment 4**: Why using AVI rather than the usual expression (VAE)?” *
>
> **Response 4**: Sorry for the confusion. We will change this expression back to (VAE) to reduce unnecessary misunderstandings.
>
> ***Comment 5**: L100: “we can donate a document” this is not clear, donate does not fit here” L 104: “with nodes on the vocabulary” : should be “in the vocabulary” and “L 136 : should be cast into  cast as”“ L 58 : Graphic Neural Topic Modeling  Graph Neural Topic Modeling. This is the name of your method!”, “ L 128: just based  based solely on the topic assignment ?”, “L 210 : “Now, the only challenge to compute the ELBO loss Ld is at the last term in Eqaution: “ this sentence is not grammatically correct and Eqaution should be Equation”*
>
> **Response 5**: Sorry for those typos and mistakes. The word “donate” in Line 100 should be “denote”. The phrase in Line 128 should be "based solely on the topic assignment". The correct expression in Line 210 is “Now, the only challenge to compute the ELBO loss $L_d$ is how to compute the last term in Equation 14.”. We will correct those typos and other mistakes in the revision.
>
> [1] Kingma, Diederik P., et al. Auto-encoding variational bayes. arXiv. 2013.

---

> > ### Comment · Reviewer_zRWm · 2021-08-31
> > **thank you**
> >
> > Thank you for the various clarifications on most of my remarks. I will keep my score of 7.
> > Regards

---

### Official Review · Reviewer_hog6 · 2021-07-17

**Rating:** 6
**Confidence:** 4

**Summary:**

Going beyond the bag-of-word assumption in topic modelling is an interesting problem, which has been studied since the introduction of the vanilla LDA, for instance, the syntactic topic model. This paper proposes to represent each document as a directed word graph, which is constructed via moving a window along the word sequence. The main technical contributions of this paper can be 1) the graph neural topic model, where words are generated conditioned on both the topic assignments and the reconstructed word graph; 2) the VAE framework used to learn the posteriors.


**Limitations And Societal Impact:**

The authors have briefly discussed the limitation of the proposed work.

**Main Review:**

* Representing each document as a directed word graph is interesting, which could potentially capture complex and nonlinear relationships among words. The idea of integrating GCN with the topic model can also be found in [1-3]. The difference is that the reconstruction of the directed graph given a set of sampled latent topic assignments and the word generation given the reconstructed graph and topic assignment, which can be the strength and could be also the weakness. The strength resides in that it could potentially learn the semantic correlation between words. However, my concern is also about the graph reconstruction. I am a bit confused by G_d and G^o_d (the placeholder) in regards to the generative process. It would be good to explain it with the example in figure 1 (a). For instance, is it assumed that the relative position of words is fixed in G^o_d, the reconstruction is just about deciding if there is a directed edge between two words. However, eq (4) seems to say E^o_d is observed, and line 107 indicates it is different from E_d. Further clarity is needed here.
* In the proposed model, each topic is also a distribution over edges between pairs of words. Presumably, there is a one to one making between a topic as a distribution over words and that as a distribution over edges.   I wonder if the topic assignment to edges is independent of the topic assigned to a word, or they are dependent.
* Both topic coherence scores and downstream clustering tasks are used in the experiment, and the proposed GNTM is compared with several neural topic models, showing promising results.  Here are my comments:
   * In regards to the topic coherence scores, it is not unknown that consistent document preprocessing can significantly affect the topic coherence scores, reporting how documents are preprocessed is necessary.
   * There is a lack of oblation study on the impact of representing topics as a distribution over edges.
   * Besides GraphBTM, it might also be good to consider those GCN based topic models, like [1-3].
   * it is good to show how sensitive the model is to the window size used in construction G_d.

References
[1] Xie et al 2021 Graph Topic Neural Network for Document Representation
[2] Long et al 2020, Graph Structural-topic Neural Network
[3] Yang et al 2020, Graph Attention Topic Modeling Network

**Time Spent Reviewing:**

>5

---

> ### Author Response · Authors · 2021-08-10
> **Responses to Reviewer hog6**
>
> Thanks for your valuable comments. Here, we provide detailed responses to each of your comments.
>
> ***Comment 1**: Representing each document as a directed word graph is interesting, … … Further clarity is needed here.*
>
> **Response 1**: Thanks for your positive comments. Sorry for the confusion about the description of notations. In our paper, for each document $d$, we denote the document graph as $G_d=(V_d, E_d)$, where $V_d=[w_{d,1}, …, w_{d,N_d}]$ is the word set and $E_d$ is the set of word edges. As for the example in Figure 1(a), $V_d$ is a set of 7 words, i.e, $w_{d,1}=$latent, $w_{d,2}=$dirichlet, …, $w_{d,7}=$documents. $E_d$ is a set of 11 word edges, i.e., $\\{$(latent, dirichlet), (latent, allocation),…, (topics, documents) $\\}$. We denote $G^o_d=(V^o_d,E^o_d)$ as the graph structure of $G_d$ with anonymous nodes. The placeholder set $V^o_d$ only includes the position assignments of each word, such as $\\{1,2,….,7\\}$ for the same example in Figure 1(a). The placeholder edge set $E_d^o$ is the edge set between position assignments, such as $\\{$(1,2), (1,3), …, (6,7)$\\}$ for the same example. In other words, $G^o_d$ can be specified solely by the edge set $E^o_d$ when giving the number of words. However, $G_d$ is dominated by the tuple of both $G^o_d$ and $V_d$.
>
> Along this line, as the description shows in section 3.1 and Algorithm 2 in Appendix, we need to generate those two observed data successively, $G^o_d$ and $V_d$, to construct the document graph $G_d$ in the generative story. Equation (4) aims to model the conditional probability of the graph structure, i.e, $G^o_d$, given the topic assignment set $z_{d,n}$ in each position $n$. This process is about deciding if there is a directed edge between two placeholders, with specific topic assignments, not words. To be specific, we denote the adjacent matrix of $G_d^o$ as $A^o_d$. Inspired by [4], we formulate $p(G^o_d|Z_d;M)$ as following:
>
> $p(G^o_d|Z_d;M) = p(E_d^o|Z_d;M)= p(A^o_d|Z_d;M) = \prod_{i=1}^N\prod_{j=1}^N p(A^o_{d,ij}|z_{d,i},z_{d,j};M),$
>
> where $p(A^o_{d,ij}=1|z_i,z_j) = m_{z_i,z_j}$, $M\in \mathbb{R}^{K\times K}$ is the dependency matrix among topics and each element $m_{i,j} \in [0,1]$ represents the possibility that there exists a dependency edge between two placeholders with topic $i$ and $j$. Then, we can obtain Equation (4) after simple derivations. In the revision, we would provide more descriptions to explain our notations.
>
> ***Comment 2**: In the proposed model, each topic is also a distribution over edges between pairs of words … … if the topic assignment to edges is independent of the topic assigned to a word, or they are dependent.*
>
> **Response 2**: Sorry for the confusion. Our model infers the topic assignment only for each word, not for word dependency edges. The hidden idea is that the topic assignments of all placeholders in the document graph should indicate both the dependency relation between different placeholders and the specific words on those positions. Along this line, as the description shows in section 3.1 and Algorithm 2 in Appendix, we need to generate those two observed data successively, $G^o_d$ and $V_d$, to construct the document graph $G_d$ based on the topic assignments $Z_d$ of all placeholders. In particular, as the description shows in Section 3.1.2, when generating word set $V_d$ given both topic assignments $Z_d$ and graph structure $G^o_d$, we assume that the words on the placeholders with the same topics are dependent mutually if there exist edges among those placeholders. Based on this assumption, we introduce the distribution on word edges into topics to model those dependency relations and design a legal probabilistic measure on the vocabulary $V$ to model the generation of $V_d$, i.e., Equation (5).
>
> ***Comment 3**: In regards to the topic coherence scores, … … reporting how documents are preprocessed is necessary.*
>
> **Response 3**: Thanks for your valuable comments. For all datasets, we filtered out the stop words and words with low frequency. Appendix A.4 has shown the important data statistics after data preprocessing. When computing the topic coherence score, similar to [5], we regarded all documents after data preprocessing in the corresponding original dataset as the reference corpus.
>
> ***Comment 4**: There is a lack of oblation study on the impact of representing topics as a distribution over edges.*
>
> **Response 4**: As your suggestion, we have conducted the oblation study by removing the distribution over edges from topics, namely GNTM w/o $\beta^e$. To be specific, different from the description in Section 3.1.2, we assume that words are independent form each other given the topic assignments $Z_d$. In other words, different from Equation (6), we formulate $P(V_d|Z_d, G_d^o)$ as:
>
> $p(V_d|Z_d,G_d^o;\beta^v) = \prod_{n=1}^{N_d} \beta^v_{z_{d,n}, w_{d,n}}. $
>
> Then, we can re-derive the loss function based on Equation (9). The experiment results on 20NG dataset in the following table show that removing the distribution over edges from topics causes the decrease of interpretability of topics.
>
> |Num. of topics|20|30|50|70|100|
> |  ----  | ----  |----  |----  |----  |----  |
> |*The performance of NPMI with top 10 words*|
> |GNTM|0.1235|0.1074|0.0807|0.0710|0.0587|
> |GNTM w/o $\beta^e$|0.0994|0.0953|0.0791|0.0701|0.0386|
> |*The performance of $C_v$ with top 10 words*|
> |GNTM|0.6962|0.6605|0.6328|0.6222|0.5939|
> |GNTM w/o $\beta^e$|0.6479 |0.6445 |0.6055 |0.5924 |0.5532 |
> |*The performance of topic diversity with top 10 words*|
> |GNTM|0.9110|0.8759|0.8352|0.7231|0.5530|
> |GNTM w/o $\beta^e$|0.9000 |0.9000 |0.7700 |0.4900 |0.3700|
>
> ***Comment 5**: Besides GraphBTM, it might also be good to consider those GCN based topic models, like [1-3].*
>
> **Response 5**: Thanks for your valuable comments. We would like to introduce more GCN based topic models in the revision to improve our model evaluation.
>
>
> ***Comment 6**: it is good to show how sensitive the model is to the window size used in construction G_d.*
>
> **Response 6**: Thanks for your comments. Indeed, in Appendix A.5, we have conducted several analyses about the sensitivity of the window size. The results show 5-size window is the best choice. We will provide more descriptions in the revision.
>
> [1] Xie, Qianqian, et al. Graph Topic Neural Network for Document Representation. WWW. 2021
>
> [2] Long, Qingqing, et al. Graph Structural-topic Neural Network. SIGKDD. 2020
>
> [3] Yang, Liang, et al. Graph Attention Topic Modeling Network. WWW. 2020.
>
> [4] Kipf, Thomas N. et al. Variational graph auto-encoders. arXiv. 2016.
>
> [5] Mimno, David et al. Optimizing semantic coherence in topic models. EMNLP. 2011.

---

### Official Review · Reviewer_7TJJ · 2021-07-18

**Rating:** 6
**Confidence:** 4

**Summary:**

This work is based on neural topic modeling where a document is represented by a graph of word notes with edges representing the semantic dependency and the generative process of learning is a function of both the document-graph structure (i.e., word-dependency edges) as well as word sets (i.e., document vocabulary). On top, the AVI approach is introduced to perform learning over the proposed Graph Neural Topic Model (GNTM).

Several past works in topic modeling have investigated dependency relations among words: modeling sequential dependency and syntactic relationships [1] using dependency parse trees. Another thread of research [2, 3] incorporates both: BoW and contextualized information in the generative process of topic modeling, where the contextualized representations introduced modeling word-dependency relations in the generative process of neural topic models. This line of research is missing in quantitative comparison with the proposed topic model.

**Limitations And Societal Impact:**

Yes

**Main Review:**

Strengths:
----------------------------------------
+ paper is well formulated and easy to read/follow
+ the proposed methodology is well detailed and described
+ novel approach of introducing document-graph structure in the generative process of topic modeling
+ novel AVI approach of learning over GNTM model


Weaknesses:
----------------------------------------
- Unclear contribution since several existing works in topic modeling have already investigated incorporating word-dependency relations in the generative process of neural topic modeling
- comparison to the baselines methods used is not fair and incomplete
- the baseline methods do not use pretrained word embeddings (e.g. word2vec), however the proposed model GNTM utilizes external knowledge via pre-trained embeddings word2vec. Therefore, comparison is unfair. The topic models with word embeddings such as [4 , 5, 6] that are based on BoW and have also incorporated pretrained word embeddings should be used in quantitative analysis for fair comparison.
- missing comparison with the topic models that incorporates word dependency relations such as [1, 2, 3]
- experimental setup is unclear; how are the scores of baseline methods obtained?
- Additional experiments required to evaluate the quality of document representations such as text classification and unsupervised IR.
- DocNADE and its variants such as iDocNADE/iDocNADEe have shown to outperform [3] ProdLDA; however the score of iDocNADE are low. Please provide experimental details and reasoning on the deviation of scores from original works?
- The proposed GNTM model incorporates word dependency relations, while the baseline methods do not and only assumes bow. Therefore, additional comparisons/experiments with the topic models variants such as [1, 2, 3] that incorporate (sequential/syntactic/contextualized) word-dependency relations are required for completeness.

References:

[1] Extended Topic Model for Word Dependency. ACL 2015.

[2] TopicRNN: A Recurrent Neural Network with Long-Range Semantic Dependency. ICLR 2017.

[3] Texttovec: Deep Contextualized Neural autoregressive Topic Models of Language with Distributed Compositional Prior. ICLR 2019.

[4] Gaussian LDA for Topic Models with Word Embeddings. ACL 2015.

[5] Improving topic models with latent feature word representations. TACL 2015.

[6] Document Informed Neural Autoregressive Topic Models with Distributional Prior. AAAI 2019.


<< --------------------------------------- Update -------------------------------------- >>

Thanks for clarification and additional evaluation in the baseline table.

Based on the response, I increase my score by +1.


**Time Spent Reviewing:**

6

---

> ### Author Response · Authors · 2021-08-10
> **Responses to Reviewer 7TJJ**
>
> Thanks for your valuable comments! Here, we provide detailed responses to each of your review comments. In particular, we have added some comparative experiments on 20NS dataset shown in the following table.
>
> |Num. of topics|20|30|50|70|100|
> |  ----  | ----  |----  |----  |----  |----  |
> |The performance of NPMI with top 10 words|
> |GNTM|0.1235|0.1074|0.0807|0.0710|0.0587|
> |TopicRNN|-0.1895 |-0.2606 |-0.2894 |-0.3124 |-0.3288 |
> |iDocNADEe|N/A|N/A|-0.0944|N/A|-0.1115|
> |The performance of $C_v$ with top 10 words|
> |GNTM|0.6962|0.6605|0.6328|0.6222|0.5939|
> |TopicRNN|0.3738 |0.3965 |0.3689 |0.3843 |0.4032 |
> |iDocNADEe|N/A|N/A|0.3744|N/A|0.3998|
> |The performance of topic diversity with top 10 words|
> |GNTM|0.9110|0.8759|0.8352|0.7231|0.5530|
> |TopicRNN|0.9700 |0.9560 |0.9420 |0.9410 |0.9190 |
> |iDocNADEe|N/A|N/A|0.5180|N/A|0.4960|
>
> ***Comment 1**: Unclear contribution since several existing works in topic modeling have already investigated incorporating word-dependency relations in the generative process of neural topic modeling*
>
> **Response 1**: As you pointed out, several existing works have studied word dependency, such as [1][2][3]. As the description in Introduction, most of them focus on enhancing language modeling with topic models. However, different from these works, we focus on improving the interpretability of topic models. To our best knowledge, we are the first to present a topic as a graph and introduce the distribution over word dependency edges to the concept of topics, which helps to improve the interpretability of topics.
>
> ***Comment2**: comparison to the baselines methods used is not fair and incomplete … … for fair comparison.*
>
> **Response 2**: Actually, in our experiments, the state-of-the-art baseline, ETM [7] (TACL 2020), is a representation of topic models based on pre-trained word embedding. In recent days, we also reproduced iDocNADEe [6] (AAAI 2019) on 20NS dataset, and the results in the above table show that our model also outperforms this baseline. Note that iDocNADEe constrains the equality of the dimension of word embedding and the number of topics. We can only set the number of topics as 50 and 100, due to the lack of pretraining word-embedding with other dimensions in public. We also tried to reproduce GaussianLDA [4]. However, the model is highly time-costing if there is a large vocabulary size. We cannot report the result due to the limitation of time. In the revision, we would explore this baseline and the models in [2], or other comparative baselines.
>
> ***Comment 3**: missing comparison with the topic models that incorporates word dependency relations such as [1, 2, 3]*
>
> **Response 3**: In recent days, we reproduced TopicRNN [2] as the representation of models you proposed incorporating word dependency relations. The results in the above table show that it is not competitive. Moreover, in this paper, we have implemented the GraphBTM [9] model as another comparison, which models word dependency in a graph manner.
>
> ***Comment 4**: experimental setup is unclear; how are the scores of baseline methods obtained?*
>
> **Response 4**: Sorry for the confusion. In our experiments, all baselines are implemented carefully with the guidance of their official code, and the code links can be found in Appendix A.5. Our code will be publicly available after the acceptance of this paper. Without specific description, the parameters of baselines are the same as those recommended by the original paper. To ensure a fair comparison, the vocabulary of our model and all baselines are the same, and the metric scores, i.e., NPMI, $C_v$, TD, Purity, and NMI, are computed based on the same strategies for all models. We would provide more descriptions on the experimental setup in the revision.
>
> ***Comment 5**: Additional experiments are required to evaluate the quality of document representations such as text classification and unsupervised IR.*
>
> **Response 5**: In this paper, the main goal is to improve the interpretability of topic models with topic coherence and topic diversity as our main metrics. We would further extend our models for other downstream tasks in the future.
>
> ***Comment 6**: DocNADE and its variants such as iDocNADE/iDocNADEe have shown to outperform [3] ProdLDA; however the score of iDocNADE are low. Please provide experimental details and reasoning on the deviation of scores from original works?*
>
> **Response 6**: Sorry for the confusion. In [3], the authors showed that DocNADE and its variants could achieve better performance based on $C_v$ metric. However, they conducted experiments based on full documents for language modeling without reduced vocabulary setting, i.e., removing the stop words and few frequency words. In contrast, we followed LDA and ProdLDA [10] and conducted our experiments with the reduced vocabulary setting, where our model and all baselines require the reduced vocabulary to improve interpretability. I have checked our code to ensure the correctness of our results. Our code will also be publicly available after possible publication. Meanwhile, we found that the sentence “NPMI over (50/200) topics learned on 20NS by: ProdLDA (.24/.19) and DocNADE (.15/.12) in RV (reduced vocabulary) setting” at the footnote of Page 7 in [3]. In other words, DocNADE cannot outperform ProdLDA based on NPMI metrics with reduced vocabulary, same as the results in our experiments.
>
> ***Comment 7**: The proposed GNTM model incorporates word dependency relations… … required for completeness.*
>
> **Response 7**: Thanks for your valuable comments. In recent days, we have reproduced TopicRNN [2] as the representation of the models you proposed. The results in the above table show that this model is not competitive. Actually, in our experiments, the baseline, GraphBTM [9], is one representation with graph-based word dependency. We would explore other types of word dependency, such as syntactic or contextualized word dependency, in future work.
>
> [1] Wang, Tong, et al. Extended topic model for word dependency. ACL. 2015.
>
> [2] Dieng, Adji B., et al. Topicrnn: A recurrent neural network with long-range semantic dependency.  ICLR. 2017.
>
> [3] Gupta, Pankaj, et al. Texttovec: Deep contextualized neural autoregressive topic models of language with distributed compositional prior. ICLR. 2019.
>
> [4] Das, Rajarshi, et al. Gaussian lda for topic models with word embeddings. ACL 2015.
>
> [5] Nguyen, et al. Improving topic models with latent feature word representations. TACL. 2015.
>
> [6] Gupta, Pankaj, et al. Document informed neural autoregressive topic models with distributional prior. AAAI. 2019.
>
> [7] Adji B Dieng, et al.. Topic modeling in embedding spaces. TACL, 2020.
>
> [8] Larochelle, H, et al. A neural autoregressive topic model. NIPS. 2012
>
> [9] Zhu, Qile, et al. GraphBTM: Graph enhanced autoencoded variational inference for biterm topic model. EMNLP. 2018.
>
> [10] Akash Srivastava, et al. Autoencoding variational inference for topic models. ICLR. 2017.

---

### Decision · Program_Chairs · 2021-09-27

**Decision:**

Accept (Poster)

**Comment:**

Reviews have positive comments about the family of topic models explored here, which uses graph-neural-networks to remove the word independence assumptions in classical models like LDA.  Some reviewers raised questions about experimental comparisons to related work, that were reasonably addressed by the author response; please be sure to include these comparisons in future revisions.  The manuscript text also needs to be polished to improve readability, and clarify several details raised in reviews.